# Ceilometers as planetary boundary layer height detectors and a corrective tool for COSMO and IFS models

Leenes Uzan[1,2], Smadar Egert[1], Pavel Khain[2], Yoav Levi[2], Elyakom Vladislavsky[2], Pinhas Alpert[1]

[1] Porter School of the Environment and Earth Sciences, Raymond and Beverly Sackler Faculty of Exact Sciences, Dept. of Geophysics, Tel-Aviv University, Tel Aviv, 6997801, Israel.
[2] The Israeli Meteorological Service, Bet Dagan, Israel.

Correspondence to: Leenes Uzan (Leenesu@gmail.com)

**Abstract**

The significance of the planetary boundary layer (PBL) height detection is apparent in various fields, especially in air pollution dispersion assessments. Numerical weather models produce a high spatial and temporal resolution of PBL heights albeit, their performance requires validation. This necessity is addressed here by an array of 8 ceilometers, a radiosonde, and two models - IFS global model and COSMO regional model. The ceilometers were analyzed by the wavelet covariance transform method, and the radiosonde and models by the parcel method and the bulk Richardson method. Good agreement for PBL height was found between the ceilometer and the adjacent Bet Dagan radiosonde (33 m a.s.l) at 11 UTC launching time (N = 91 days, ME = 4 m, RMSE=143 m, R=0.83). The models' estimations were then compared to the ceilometers' results in an additional five diverse regions where only ceilometers operate. A correction tool was established based on the altitude (h) and distance from shoreline (d) of eight ceilometer sites in various climate regions, from the shoreline of Tel Aviv (h = 5 m a.s.l, d = 0.05 km), to eastern elevated Jerusalem (h = 830 m a.s.l, d = 53 km), and southern arid Hazerim (h = 200 m a.s.l, d = 44 km). The tool examined the COSMO PBL height approximations based on the parcel method. Results for August 14, 2015 case-study, between 9-14 UTC showed the tool decreased the PBL height in the shoreline and inner strip of Israel by ~ 100 m and increased the elevated sites of Jerusalem up to ~ 400 m, and Hazerim up to ~ 600 m. Cross-validation revealed good results without Bet Dagan. However, without measurements from Jerusalem, the tool underestimated Jerusalem's PBL height up to ~600 m difference.

## 1. Introduction

In the era of substantial industrial development, the need to mitigate the detrimental effects of air pollution exposure is unquestionable (Anenberg et al., 2019, WHO, 2016, Héroux et al., 2015, Dockery et al.,1993). However, to regulate and establish environmental thresholds, a comprehensive understanding of the air pollution dispersion processes is necessary (Luo et al., 2014, Seidel et al., 2012, Seidel et al., 2010, Ogawa et al., 1986, Lyons, 1975). One of the critical meteorological parameters governing air pollution dispersion is the planetary boundary layer (PBL) height (Sharf et al., 1993, Garratt, 1992, Ludwing, 1983, Dayan et al., 1988). The PBL height is classified as the first level of the atmosphere that dictates the vertical dispersion extent of air pollution (Stull, 1988). Hence, the quality of meteorological data provided to these models is of great importance (Urbanski et al., 2010, Scarino et al., 2014, Su et al., 2018).

Numerical weather prediction (NWP) models provide a high temporal and spatial resolution of PBL height based on mathematical equations with initial assumptions, and boundary conditioned set beforehand. However, the models display difficulty to accurately simulate the PBL creation and evolution (Luo et al., 2014, Seidal et al., 2010), and validation against actual measurements is advised. In situ atmospheric measurements by radiosondes are most efficient but costly as successive measurements. Remote sensing measurements such as wind profilers and sophisticated lidars are mostly designated for specific campaigns limited in location and operational time (Manninen et al., 2018, Mamouri et al., 2016). Ceilometers, on the other hand, are ubiquitous in airports and meteorological service centers worldwide (TOPROF of COST Action ES1303 and E-PROFILE of the EUMETNET Profiling Program), thus provide an advantage over the relatively scarce deployment of sophisticated lidars.

Ceilometers are single wavelength micro-lidars intended for cloud base height detection. Vaisala ceilometers produce backscatter profiles every ~15 s with a vertical resolution of 10 m and a height range up to 8 or 15 km, depending on the ceilometer type and the atmospheric conditions (Uzan et al., 2018). Unlike sophisticated lidars, ceilometers are not equipped to provide aerosol properties such as size distribution, scattering, and absorption coefficients (Ansmann et al., 2011, Papayannis et al., 2008, Ansmann et al., 2003). Nevertheless, their advantages have been recognized as low cost, easy to maintain, and continuous unattended operation under diverse meteorological conditions (Kotthaus and Grimmond, 2018). Over the years, several studies have assigned ceilometers as PBL height detectors (Eresmaa et al., 2006, Van der Kamp and McKendry, 2010, Haeffelin and Angelini, 2012, Wiegner et al., 2014).

Previous research employed ceilometers as PBL height detectors and compared them to NWP models (Collaud et al., 2014, Ketterer et al., 2014, Gierens et al., 2018). However, scarce attention has been paid to designate ceilometers for a correction tool for NWP PBL heights. The main goal of this study is to create this tool and improve the input data for air pollution dispersion evaluations. A description of the models and instruments applied is given in Sect. 2 and Sect. 3, respectively. Sect. 4 presents the PBL height detection methods. Spatial and temporal analysis of the PBL heights generated by the models and instruments in six sites are shown in Sect 5.1. The PBL height correction tool is explained in 5.2 and demonstrated by a case-study employing eight ceilometer sites.  Summary and conclusions are drawn in Sect. 6.

## 1.1 Study time and region

Located in the East Mediterranean, Israel obtains various climate measurement sites in comparatively short distances (Fig. 1). The ceilometer array (Fig. 1, Table 1) is comprised of two coastline sites, 40 km apart, in Hadera (10 m a.s.l), and Tel-Aviv (5 m a.s.l). Further inland, 12 km and 23 km southeast to Tel Aviv, are Bet Dagan (33 m a.s.l) and Weizmann (60 m a.s.l), respectively. About 70 km southwest to the elevated Jerusalem site (830 m a.s.l) are Hazerim (200 m a.s.l) and Nevatim (400 m a.s.l).  Ramat David (50 m a.s.l) represents the northern region 24 km inland.

Various institutions operate the ceilometers. In several cases, the ceilometers' output files were not methodically saved. In others, the ceilometers worked for limited periods. Following Kotthaus & Grimmond (2018), the analysis concentrated on the dry summer season due to the difficulty of evaluating the PBL height from backscatter signals during precipitation episodes. The database narrowed down by removing dates with partial data or during dust storm events such as the unprecedented extreme dust storm in September 2015 (Uzan et al., 2018). In general, summer dust outbreaks in the eastern Mediterranean are quite rare at the low altitudes (~ 1-2 km) of the PBL height (Alpert and Ziv 1989, Alpert et al., 2000, Alpert et al., 2002). Eventually, the analysis focused the data available from each ceilometer within six summer months: July-September 2015, and June-August 2016.

A characteristic Israeli summer has no precipitation and mainly sporadic shallow cumulus clouds (Ziv et al., 2004, Goldreich, 2003, Saaroni and Ziv, 2000). The dominant synoptic system is the persistent Persian Trough (either deep, shallow, or medium) followed by a Subtropical High aloft (Alpert et al., 1990, Feliks Y., 1994, Dayan et al., 2002, Alpert et al.,

2004). The average summer PBL height is under 2 km a.s.l  (Dayan et al., 1988, Feliks 2004)

Since backscatter signals decline with height, the conditions of low PBL heights comes as an advantage.

## 1.2 The summer PBL height

The formation and evolution of the summer PBL height are as follows: After sunrise, ~ 4-5 local standard time (LST = UTC+2), clouds initially formed over the Mediterranean Sea,

advect eastward to the shoreline. As the ground warms up, the nocturnal surface boundary layer dissipates, and buoyancy induced convective updrafts instigate the formation of the sea breeze

circulation (Stull, 1988). Previous research of the PBL height in Bet Dagan (33 m a.s.l and 7.5 km east from the shoreline) revealed an average height of ~900 m a.g.l after sunrise (Koch and

Dayan, 1988, Feliks Y.,1994, Dayan and Rodinzki, 1999, Uzan et al., 2016, Yuval et al., 2019). The sea breeze front enters between 7-9 LST (Feliks Y., 1993, Alpert and Rabinovich-Hadar,

2003, Uzan and Alpert, 2012), depending on the time of sunrise and the different synoptic modes of the prevailing system - the Persian Trough (Alpert et al., 2004). Cool and humid

marine air hinder the convective updrafts. Clouds dissolve, and the height of the shoreline boundary layer lowers by ~250 m (Feliks Y., 1993, Feliks Y., 1994, Levi et al., 2011, Uzan

and Alpert, 2012). Further inland, the convective thermals continue to inflate the boundary layer (Hashmonay et al., 1991, Feliks, 1993, Lieman, R. and Alpert, 1993). West-north-west

synoptic winds enhance the sea breeze wind as it steers north-west (Neumann, 1952, Neumann, 1977, Uzan and Alpert, 2012). By noontime (~11-13 LST), maximum wind speeds further

suppress the boundary layer (Uzan and Alpert, 2012). In the afternoon (~13-14 LST), the sea breeze front reaches ~30-50 km inland to the eastern elevated complex terrain (Hashmonay et

al., 1991, Lieman, R. and Alpert, 1993). At sunset (~18-19 LST), as the insolation diminishes, the potential energy of the convective updrafts weakens, and the boundary layer height drops

(Dayan and Rodnizki, 1999). After sunset, as ground temperature cools down, the boundary layer collapses, and a residual layer is formed above a surface boundary layer (Stull, 1988).

High humidity and a low residual layer create low condensation levels, and shallow evening clouds are produced.

## 2. IFS and COSMO Models

IMS capitalizes two operational models: The European Centre for Medium-range Weather Forecasts (ECMWF) Integrated Forecast System (IFS) global model and the COnsortium for

Small-scale MOdeling (COSMO) regional model (Table 2).

IFS consists of 137 vertical levels. In the years 2015 and 2016 relevant to this study, the grid

resolution was ~13 km and ~10 km, respectively. It applies a turbulent diffusion scheme representing the vertical exchange of heat, momentum, and moisture through the sub-grid

turbulence scale. A first-order K-diffusion closure based on the Monin-Obukhov (MO) similarity theory represents the surface layer turbulent fluxes. The Eddy-Diffusivity Mass-Flux

(EDMF) framework (Koehler et al. 2011) describes the unstable conditions above the surface layer.

IMS runs COSMO over the Eastern Mediterranean domain (25-39 E/26-36 N) since 2013 with boundary and initial conditions from IFS. It consists of 60 vertical levels up to 23.5 km and a

horizontal grid spacing of 2.5 km (Table 2). Primitive thermo-hydrodynamic equations represent the non-hydrostatic compressible flow in a moist atmosphere (Steppeler et al., 2003,

Doms et al., 2011, Baldauf et al., 2011). The model runs a two-time level integration scheme, based on a third-order of the Runge–Kutta method, and a fifth-order of the upwind scheme for

horizontal advection. Unlike in the IFS model, in the COSMO model, only shallow convection is parameterized, and the deep convection is switched off (Tiedtke, 1989). The turbulence

scheme of Mellor and Yamada (1982) at level 2.5, uses a reduced second-order closure with a prognostic equation for the turbulent kinetic energy. Transport and local time tendency terms

in the second-order momentum equations are neglected, and the vertical turbulent fluxes are derived diagnostically (Cerenzia I., 2017).

Both models estimate the PBL height by The bulk Richardson number method (described in Sect. 4.1).  IFS produced hourly results while COSMO generated profiles every 15 min. A

series of trials disclosed the COSMO profiles of the last 15 min within an hour, best represent the hourly values of the IFS model.

## 3. Instruments

### 3.1 Ceilometers

Vaisala ceilometers type CL31 is the primary research tool in this study (Fig.1, Table 1). CL31 is a pulsed, elastic micro-lidar, employing an Indium Gallium Arsenide (InGaAs) laser diode transmitter of 910 nm ±10 nm near-infrared wavelength at 25˚C and a high pulse repetition rate of 10 kHz, every two seconds (Vaisala ceilometer CL31 user's guide: http://www.vaisala.com). The backscatter signals are collected by an avalanche photodiode (APD) receiver and designed as attenuated backscatter profiles at intervals of 2-120 s (determined by the user). This study applied CL31 ceilometers except for ceilometer CL51 stationed in Weizmann Institute (Fig.1, Table 1). CL51 consists of a higher signal and signal-to-noise ratio. Hence the backscatter profile measurement reaches ~15 km compared to ~ 8 km of CL31. The ceilometers produce 10 m vertical resolution profiles every 15 or 16 sec. Half hourly backscatter profiles improved the signal to noise ratio. The second half-hour profile within each hour defined the hourly profiles.

One drawback is that calibration procedures were nonexistent in all sites. In most cases, maintenance procedures (cleaning of the ceilometer window), were not regularly carried out, except for the IMS Bet Dagan ceilometer. In the case of the backscatter coefficients detection, signal calibrations, and water vapor corrections are necessary (Wiegner and Gasteiger, 2015). However, the PBL height detection method employed here (Sect. 4.3), locates the height of a pronounced change in the attenuated backscatter profile rather than a specific value. Therefore, calibration procedures are not mandatory (Weigner et al., 2014, Gierens et al., 2018).

### 3.2 Radiosonde

IMS obtains systematic radiosonde atmospheric observations twice daily, at 23 UTC and 11 UTC. The radiosonde launching site is adjacent to the Bet Dagan ceilometer (32.0 ° long, 34.8 ° lat, 33 m a.s.l, 7.5 km east from the shoreline, 12 km southeast to Tel Aviv, 45 km north-west to Jerusalem, see Fig.1 and Table 1). The radiosonde, type Vaisala RS41-SG, retrieves profiles of relative humidity, temperature, pressure, wind speed, and wind direction, every 10 seconds, (~ every 45 m), rising to ~25 km. Here, we refer to the first 2 km for the detection of the midday summer PBL height. At this height, the average wind speed at 11 UTC is ~5 m/s (Uzan et al., 2012). Therefore, the horizontal displacement is relatively low ~ 2.5 km and neglected.

Moreover, previous research showed the midday PBL height in Bet Dagan is below 1 km (Dayan and Rodinzki, 1999, Uzan et al., 2016, Yuval et al., 2019), corresponding to horizontal
displacement of ~ 0.01° which is well under the grid resolution of the IFS and the COSMO models.

## 4. Methods

**4.1 The bulk Richardson number method**

The COSMO and IFS schemes calculate the PBL height by the bulk Richardson number ($R_b$)
method (Hanna R. Steven,1969, Zhang et al., 2014) given in the formula below:

$$R_b = \frac{\frac{g}{\theta_v}(\theta_{vz} - \theta_{v0})(Z - Z_0)}{U^2 + V^2} \tag{1}$$

where g is the gravitational force, $\theta_{vz}$ is the virtual potential temperature at height Z, $\theta_{v0}$ is the virtual potential temperature at ground level ($Z_0$). U and V are the horizontal wind speed
components at height Z (assuming U and V at surface height are insignificant, therefore negligible).

The $R_b$ threshold determines the PBL height. The IFS model has a single limit of 0.25 (Seidel et al., 2012). The COSMO model refers to 0.33 for stable atmospheric conditions (Wetzel,
1982), and 0.22 for unstable conditions by 0.22 (Vogelezang and Holtslag, 1996) in the first four levels of the model (10, 34.2, 67.9, 112.3 m a.g.l.). Linear interpolation determines the
height if the detection is between two model levels. The height is assigned with a missing value if the thresholds were not reached. The models' PBL heights ( given as m a.g.l.) are adjusted to
the actual altitude of the ceilometer sites (Table 1). The radiosonde 11 UTC PBL heights were defined where the $R_b$ profile values (derived every 10 sec correspond ding to ~ 45 m)  altered
from negative to positive. In all the dates studied, the first positive value was well above the thresholds for unstable conditions by both models (0.25 and 0.33). Therefore the PBL height
was defined at the height point of the last negative value.

## 4.2 The parcel method

The parcel method defines the PBL height where the virtual potential temperature aloft reaches the value evaluated at the surface level (Holzworth 1964, Stull, 1988, Seidel et al., 2010). The description of the virtual potential temperature is as follows:

$$\theta_v = T_v \left(\frac{P_0}{P}\right)^{\frac{Rd}{Cp}} \tag{2}$$

where $P_0$ is the ground level pressure, $P$ is the pressure at height Z, $R_d$ is the dry air gas constant, $C_p$ is the heat capacity of dry air. The virtual temperature ($T_v$) is obtained by:

$$T_v = \frac{T}{1-\frac{e}{P}(1-\varepsilon)} \tag{3}$$

where T is the temperature at height Z, $e$ is the actual vapor pressure, and $\varepsilon$ is the ratio of molecular weight of water vapor and dry air ($\varepsilon$=0.622).

The virtual potential temperature profiles were computed based on the available meteorological parameters from the models and radiosonde: mixing ratio, pressure, and temperature profiles from the IFS model. Relative humidity, pressure, and temperature profiles from the COSMO model and the radiosonde. The virtual potential temperature profiles of the models at ground level were obtained by the temperature and dew point temperature at 2 m a.g.l. These parameters were derived from the models by the similarity theory. Finally, the PBL heights (given in m. a.s.l) were adjusted to the actual altitude of the ceilometer sites (Table 1).

## 4.3 The wavelet covariance transform method

The wavelet covariance transform (WCT) method (Baars et al., 2008, Brooks Ian, 2003) is implemented on backscatter profiles by the formula given in Eq. 4:

$$W_{f(a,b)} = \frac{1}{a} \int_{Zb}^{Zt} f(z) h\left(\frac{z-b}{a}\right) dz \tag{4}$$

where $W_{f(a,b)}$ is the local maximum of the backscatter profile ($f(z)$) determined within the range of step (a) by the Haar step function ($h$). The length of the step is the number of height levels (n) multiplied by the profile height resolution ($\Delta z$) from ground level (Zb) and up (Zt).

246 In this study, Zb was defined as the height above the perturbation of the overlap function (~ 100 m), and Zt as the height with the most significant signal variance or, the first appearance

248 of negative values. Both thresholds indicate a low signal-to-noise ratio. Zb is the lowest height among the two options. These thresholds apply under clear sky conditions. When clouds exist

250 in the summer, they are mainly shallow cumulus clouds (Sect. 1.1). The PBL height is the height within the cloud, above the cloud base height  (Wang et al. 2012, Stull 1988).

252 The Haar step function given in Eq.(4) is equivalent to a derivative at height $z$, representing the value difference of each step ($a$) above and beneath a point of interest ($b$). In this study, $b$ is

254 the measurement heights of the ceilometer backscatter profile. The value of the step ($a$ ) varied for each ceilometer, depending on the site location.

$$
\qquad h(\frac{z-b}{a}) = \begin{cases} +1, & b - \frac{a}{2} \le z \le b, \\ -1, & b \le z \le b + \frac{a}{2} \\ 0, & elsewhere \end{cases} \qquad\qquad (5)
$$

In arid and dusty areas such as Nevatim and Hazerim, specifically on clear days, the WCT

258 method failed to distinguish the PBL height (Van der Kamp and McKendry, 2010, Gierens et al., 2018). The analysis excluded these cases. The last stage consisted of manual inspection of

260 the WCT results.

262 **5. Results**

In the Israeli summer season, stable PBL conditions are generated from sunset to an hour after

264 sunrise (Stull, 1988). At this period the models'  $R_b$ profiles do not accede the relevant thresholds, and a missing value is assigned (Sect. 4.1). Additionally, the difficulty to estimate

266 the surface boundary layer by ceilometers (Gierens et al., 2018) was associated with a constant perturbation within the first range gates due to the overlap of the emitted laser beam and the

268 receiver's field of view (Weigner et al., 2014). Hence, the analysis focused on the midday summer PBL heights.

270 **5.1 Spatial and temporal analysis**

The analysis was performed based on six ceilometers with available data of at least 50 days

272 within the study period: Bet Dagan, Tel Aviv, Ramat David, Weizmann, Jerusalem, and Nevatim. In Bet Dagan, the results were compared to the radiosonde, thereupon, the analysis

fixated at 11 UTC launching time. In the remaining five sites, the models compared to the ceilometers. Statistical analysis for each site presents the mean error (ME), root mean square

(RMSE), correlation (R), Mean and standard deviation (STD) given in tables and plots.

Good agreement was found between the ceilometer and the radiosonde (RS) in Bet Dagan (Fig.

2 and Table 3, ME = 4, RMSE = 143, R = 0.83). The IFS by the parcel method (IFS-pm) appears to overestimate the PBL height (ME = 346, RMSE = 494, R = 0.14), as well as by the

Richardson method (IFS-ri, ME = 366, RMSE = 579, R = -0.13). Among the models and methods, the COSMO model by the parcel method derived the best results (COSMO-pm, ME

= -52, RMSE = 146, R = 0.84).

In the shoreline site of Tel Aviv (Fig. 3, Table 4), COSMO-pm displayed good agreement with

284 the ceilometer measurements (ME = 17, RMSE = 183, R = 0.74), similar to COSMO-ri  (ME = 18, RMSE = 187, R = 0.7). IFS-ri produced the highest overestimations (ME = 436, RMSE

= 616, R = -0.03).

In Ramat David, stationed in the northern inner plain of Israel, the parcel method derived better

results than the Richardson method in both models (Fig. 4, Table 5). Among the models, COSMO displayed better results (ME = 40, RMSE = 245, R = 0.55). IFS-ri generated the

poorest correlation (ME = 446, RMSE = 745, R = -0.08).

In Weizmann (Fig. 5 Table 6), 11 km southeast to Bet Dagan, IFS-ri produced the poor results

(ME = 430, RMSE = 604, R = -0.01), conversley to the good results by the parcel method (ME = 67, RMSE = 162, R = 0.85). The COSMO model derived similar results by both methods

(COSMO-pm: ME = -106, RMSE = 207, R = 0.76, COSMO-ri: ME = 21, RMSE = 192, R = 0.72).

In the mountainous site of Jerusalem, the bulk Richardson method produced better results than the parcel method in both models (Fig. 6, Table 7). COSMO-pm derived good results (ME = -

44, RMSE = 239, R = 0.70) and IFS-ri the poorest (ME = 366, RMSE = 498, R = 0.18).

In the elevated and arid site of Nevatim, overall correlations were weak (0.1-0.3) and high

RMSE (369 - 488).

Main conclusions derived from Fig.2-7 are summarized below:

-   Low correlation in Nevatim (0.1-0.3) demonstrates the difficulty of the models to assess the PBL height over complex terrain. Evaluation of PBL heights in complex terrain was

studies by Ketterer et al. (2014) in the Swiss Alps by a ceilometer, wind profiler, and

in-situ continuous aerosol measurements. The ceilometers analyzed by the gradient and STRAT-2D algorithms and the wind profiler by the range-corrected SNR method. The results compared to the COSMO-2 regional model. The results showed good agreement found between the heights derived by the ceilometer and wind profiler during the daytime cloud-free conditions ($R^2=0.81$). However, in most cases, the model underestimated the PBL height. The researchers presumed the grid resolution, parametrization schemes, and the surface type did not match the real topography. The comparison between a single measurement point and a grid point is not straight forward.

- The parcel method achieved better results in Ramat David, Tel Aviv, Bet Dagan, and Weizmann. In the elevated site of Jerusalem, the correlation of COSMO-ri was the highest (R=0.7).

- The COSMO model produced better results in the shoreline and plain regions (Ramat David, Tel Aviv, Bet Dagan) except for Weizmann (60 m a.s.l, 11.5 km from the coastline), where IFS-pm obtained the highest correlation (R=0.85).

- IFS model based on the bulk Richardson method overestimated the PBL heights ( ~ 420 m) in the plain sites of Bet Dagan, Tel Aviv, Weizmann, and Ramat David. The bulk Richardson evaluation (See Sect. 4.1) includes the horizontal wind speed profiles that are less accurate and may contribute to the discrepancies. Collaud et al. (2014) referred to the limitations of the bulk Richardson method of the COSMO-2 regional model (2.2 km resolution), which overestimated the convective boundary layer by 500–1000m. They explained the Richardson method is sensitive to the surface temperature, and errors and uncertainties in the model's temperature and relative humidity profiles could explain the significant bias. Also, the occurrence of clouds, which may be missing in the model, can lead to lower PBL heights.

## 5.2 COSMO PBL height correction

A correction formula for the models' PBL height employing ceilometers is given below:

$$dH_{st} = \alpha h_{st} + \beta d_{st} + \gamma \tag{6}$$

Where $dH_{st}$ is the PBL height difference between the ceilometer and the model, the altitude ($h_{st}$), and distance from the shoreline ($d_{st}$) for each measurement site ($st$). The formula runs

simultaneously for all ceilometer sites to derive the dependent variables α, β, and the constant

γ. The formula is suitable for both models

A case-study demonstrates the correction formula on August 14, 2015, from the COSMO

model based on the parcel method (COSMO-pm). COSMO-pm is the model and method that

derived good results in Sect. 5.1. The formula runs for each hour between 9-14 UTC for the

daytime PBL height (See Sect. 5). Results are portrayed for each hour by a 2-D plot of the

height correction within the area of ceilometers' deployment. Along with an east-west cross-

section plot, corresponding to the location of the ceilometers. Cross-validation tests for Bet

Dagan and Jerusalem show the effectivity of the correction formula. Main findings for each

344    hour are as follows:

9 UTC (Fig. 8): Along the coast, the correction tool lowers the PBL height by 70 m to 670 m

and increases by 90 m in the inner strip of Israel to ~ 890 m a.s.l. Cross-validation for Bet

Dagan (CV-BD) shows good results, whereas, in Jerusalem (CV-JRM), the correction tool

reduced the height by 600 m.

10 UTC (Fig. 9): The correction tool distinguishes between the coastal sites of Tel Aviv and

Hadera, and the inland locations of Bet Dagan and Weizmann, only ~ 10 km apart from Tel

Aviv. While the correction tool increased the height of the coastal stations, a slight height

decreased was performed in the inner sites. In the arid southern Hazerim, the correction tool

lowered the  PBL height by 400 m. In the desert south of Nevatim, the correction tool decreased

the PBL height by 200 m. Cross-validation of Jerusalem (CV-JRM) underestimates the PBL

height in Jerusalem by 400 m.

11 UTC (Fig. 10): A distinction between the shoreline and the inner sites is more evident, as

the PBL height of Tel Aviv and Hadera is increased by ~100 m to ~700 m a.s.l, whereas, Bet

Dagan and Weizmann remained ~ 800 m a.s.l. This finding corresponds to Uzan et al. (2016)

analysis of the mean diurnal-cycle of the PBL height from July to August 2014, based on

ceilometer measurements. A pronounced correction is visible in the elevated southern site of

Hazerim by 550 m down to 1120 m a.s.l.  This gap is not unexpected since NWP models have

difficulty assessing the meteorological conditions over complex terrain. Here, Jerusalem cross-

validation (CV-JRM) underestimates the PBL height by a comparatively lower range of 200

364    m.

12 UTC (Fig. 11): The correction tool increased the PBL height in the coast and inland stations,

but in fact, the height is lower than an hour before. The PBL height in Hazerim is decreased by

300 m. Jerusalem cross-validation (CV-JRM) underestimates the PBL height in Jerusalem by
600 m.

13 UTC (Fig. 12): The correction tool increased the PBL heights. A substantial increase of 380 m in Jerusalem generates a height of ~1750 m a.s.l. Jerusalem cross-validation (CV-JRM) underestimates the PBL height by 550 m.

14 UTC (Fig. 13): Similar to an hour before, the correction increases the PBL height in all sites, but in fact, the PBL heights are lower than an hour earlier, except a mild increase in the coastal locations of Tel Aviv and Hadera. Jerusalem cross-validation (CV-JRM) underestimates the PBL height by ~300 m.

## 6. Summary and Conclusions

The primary purpose of this study was to improve the performance of air pollution dispersion models by providing applicable data of PBL heights from NWP models employing ceilometers. A correction tool using ceilometer measurements was established to validate the models' PBL height assessments. The study focused on the summer PBL heights (July-September 2015, June-August 2016) during the day hours (9-14 UTC). At this period, the highest air pollution events occur in Israel from tall stacks (Dayan et al., 1988, Uzan et al., 2012).

The study contained eight ceilometers, a radiosonde, two models - IFS and COSMO, and three PBL height analysis methods. The bulk Richardson method, the parcel method for the models and radiosonde, and the WCT method for the ceilometers. In Bet Dagan radiosonde launching site, results revealed good agreement between the ceilometer's PBL heights and the radiosonde (N = 91 days, ME = 4 m, RMSE=143 m, R=0.83). In Ramat David, Tel Aviv, Weizmann, Jerusalem, and Nevatim, the models were compared to the ceilometers. The COSMO model performed better in the plain areas of Tel Aviv (10 m a.s.l), Bet Dagan (33 m a.s.l), and Ramat David (50 m a.s.l) and the mountainous Jerusalem (830 m a.s.l). The IFS model showed good agreement with the ceilometer in Weizmann (60 m a.s.l, N=55 days, ME = 67 m, RMSE = 162 m, R=0.85). In the arid southern site of Nevatim (400 m a.s.l), overall correlations were poor. The IFS-pm produced better in Bet Dagan, Ramat David, Tel Aviv, and Weizmann (four out of five sites except for Nevatim). The COSMO-pm produced better results in Bet Dagan and Ramat David, while in Tel Aviv the results generated by both methods were similar (N = 123

days, COSMO-pm: ME = 17 m, RMSE = 183 m, R=0.74, COSMO-ri: ME = 18 m, RMSE = 180 m, R=0.80).

The PBL height correction tool for the NWP models is based on the altitude and the distance from the shoreline of the ceilometers' measurement sites. A case-study demonstrated the tool's feasibility on August 14, 2015. Moving from 9 to 14 UTC, the correction decreased the PBL height in flat terrain (Tel Aviv, Hadera, Bet Dagan, and Ramat David). This finding corresponds with Uzan et al., 2016, analyzing the diurnal PBL height of Bet Dagan and Tel Aviv in the summer of 2014. Similar results produced in Hadera describe the summer PBL height between 1997-1999 and 2002-2005 based on measurements from a wind profiler (Uzan et al., 2012). Koch and Dayan (1992) revealed air pollution episodes of sulfur dioxide increased in shallow PBL heights in the coastal plain of Israel. Uzan et al. (2012) showed an average decrease of ~ 100 m in the coastal PBL height resulted in an average increase of ~200 air pollution episodes of sulfur dioxide.

The tool increased the PBL height in the elevated site of Jerusalem (830 m a.s.l) by ~380 m. In the arid south in Hazerim (200 m a.s.l), the tool lowered the PBL height by ~ 550 m. The significant height corrections in the elevated sites are attributed to the models' difficulty to imitate local meteorological processes in complex terrain (e.g., Alpert et al., 1984). Dayan et al. (1988) presumed the diurnal cycle and the prevailing synoptic systems govern the temporal behavior of the Israeli summer PBL height. The strength of the sea breeze determines significant variations in the inner PBL heights.

Cross-validation for Bet Dagan produced excellent results. Bet Dagan is located in flat terrain 11 km north to the Weizmann site and 12 km southeast to Tel Aviv site. Without the single measurement site in Jerusalem (830 m a.s.l), the correction tool failed to generate Jerusalem's PBL height and produced lower values up to a 600 m difference. This finding shows the process of cross-validation can assist in defining the required ceilometers' deployment in the future.

In summary, our results offer a preview of the great potential of ceilometers as a correction tool for PBL heights derived from NWP models. This tool demonstrates the benefit of deploying ceilometers, specifically in complex terrain. Future research should include a larger dataset to create a systematic correction process and produce sufficient input data for mandatory air pollution dispersion assessments.

**Data availability**

Weather reports - Israeli Meteorological Service weather reports (in Hebrew): http://www.ims.gov.il/IMS/CLIMATE/ClimateSummary.

Radiosonde profiles - Israeli Meteorological Service provided by request.

Ceilometer profiles - the data is owned by several institutions and provided by request.

**Author contribution**

Leenes Uzan carried out the research and prepared the manuscript under the careful guidance of Pinhas Alpert and Smadar Egert alongside a fruitful collaboration with Yoav Levi, Pavel Khain, and Elyakom Vladislavsky. The authors declare that they have no conflict of interest.

**Acknowledgments**

We wish to thank the Israeli Meteorological Service, the Israeli Air Force, the Association of Towns for Environmental Protection (Sharon-Carmel), and Rafat Qubaj from the Department of Earth and Planetary Sciences at the Weizmann Institute of Science for their ceilometer data. We thank Noam Halfon from the IMS for the topography map.

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

Table 1. Location and type of ceilometers

| Location | Terrain | Lat/Lon | Distance from MD[c] shoreline (km) | Height (m a.s.l) | Ceilometer type (resolution, max range[a]) |
|---|---|---|---|---|---|
| Ramat David (RD) | Plain | 32.7 °/35.2 ° | 24 | 50 | CL31 (10 m,16 s, up to 7.7 km) |
| Hadera (HD) | Coast | 32.5 °/34.9 ° | 3.5 | 10 | CL31 (10 m,16 s, up to 7.7 km) |
| Tel Aviv (TLV) | Coast | 32.1 °/34.8 ° | 0.05 | 5 | CL31 (10 m,16 s, up to 7.7 km) |
| Bet Dagan (BD)[b] | Plain | 32.0 °/34.8 ° | 7.5 | 33 | CL31 (10 m,15 s, up to 7.7 km) |
| Weizmann (WZ) | Plain | 31.9 °/34.8 ° | 11.5 | 60 | CL51 (10 m,16 s, up to 15.4 km) |
| Jerusalem (JRM) | Mount. | 31.8 °/35.2 ° | 53 | 830 | CL31 (10 m,16 s, up to 7.7 km) |
| Hazerim (HZ) | Arid | 31.2 °/34.6 ° | 44 | 200 | CL31 (10 m,16 s, up to 7.7 km) |
| Nevatim (NV) | Arid | 31.2 °/35.0 ° | 70 | 400 | CL31 (10 m,16 s, up to 7.7 km) |

[a]The maximum height decreases as the atmospheric optical density increases.
[b]Adjacent to the radiosonde launch site.
[c] Mediterranean

Table 2. Description of the NWP models

| Model | IFS | COSMO |
|---|---|---|
| Operation center | ECMWF | IMS |
| Global/regional | Global | Regional, boundary conditions from IFS |
| Horizontal grid resolution | 0.125$^o$ in 2015 (~13km) <br> 0.1$^o$ in 2016 (~9 km) | 0.025$^o$ (~2.5 km) |
| Vertical grid resolution | 137 layers up to ~79 km <br> 23 lie within the first 3 km | 60 layers up to 23.5 km <br> 20 lie within the first 3 km |
| Temporal resolution of the output | Hourly profiles | 15 min profiles |
| Convection parametrization | Mass flux Tiedke-Bechthold (Bechthold, 2008) | Deep convection resolved. Parametrization of mass flux shallow convection. (Tiedtke, 1989) |

Table 3. Statistical analysis of Bet Dagan PBL heights (N=91, Fig. 2a)

| PBL detection | IFS-pm | COSMO-pm | IFS-ri | COSMO-ri | Ceilometer | RS |
|---|---|---|---|---|---|---|
| Mean Error (m) | 346 | -52 | 366 | 57 | 4 | - |
| RMSE (m) | 494 | 146 | 579 | 193 | 143 | - |
| R | 0.14 | 0.84 | -0.13 | 0.7 | 0.83 | - |
| Mean (m a.s.l) | 1236 | 838 | 1255 | 947 | 894 | 890 |
| STD (m) | 290 | 237 | 346 | 232 | 239 | 245 |

Table 4. Statistical analysis of Tel Aviv PBL heights (N=122, Fig. 3a)

| PBL detection | IFS-pm | COSMO-pm | IFS-ri | COSMO-ri | Ceilometer |
|---|---|---|---|---|---|
| Mean Error (m) | 14 | 17 | 436 | 18 | - |
| RMSE (m) | 256 | 183 | 616 | 180 | - |
| R | 0.47 | 0.74 | -0.03 | 0.73 | - |
| Mean (m a.s.l) | 702 | 706 | 1124 | 707 | 674 |
| STD (m) | 224 | 238 | 337 | 211 | 258 |

Table 5. Statistical analysis of Ramat David PBL heights (N=123, Fig. 4a)

| PBL detection | IFS-pm | COSMO-pm | IFS-ri | COSMO-ri | Ceilometer |
|---|---|---|---|---|---|
| Mean Error (m) | 4 | 40 | 446 | 123 | - |
| RMSE (m) | 347 | 245 | 745 | 313 | - |
| R | 0.14 | 0.55 | -0.08 | 0.39 | - |
| Mean (m a.s.l) | 995 | 1031 | 1437 | 1114 | 991 |
| STD (m) | 276 | 256 | 521 | 268 | 253 |

Table 6. Statistical analysis of Weizmann PBL heights (N=55, Fig. 5a)

| PBL detection | IFS-pm | COSMO-pm | IFS-ri | COSMO-ri | Ceilometer |
|---|---|---|---|---|---|
| Mean Error (m) | 67 | -106 | 430 | 21 | - |
| RMSE (m) | 162 | 207 | 604 | 192 | - |
| R | 0.85 | 0.76 | -0.01 | 0.72 | - |
| Mean (m a.s.l) | 892 | 719 | 1256 | 846 | 825 |
| STD (m) | 186 | 193 | 322 | 219 | 271 |

Table 7. Statistical analysis of Jerusalem PBL heights (N=53, Fig. 6a)

| PBL detection | IFS-pm | COSMO-pm | IFS-ri | COSMO-ri | Ceilometer |
|---|---|---|---|---|---|
| Mean Error (m) | 366 | -129 | 117 | -44 | - |
| RMSE (m) | 498 | 252 | 257 | 239 | - |
| R | 0.18 | 0.63 | 0.59 | 0.70 | - |
| Mean (m a.s.l) | 2239 | 1744 | 1991 | 1830 | 1874 |
| STD (m) | 276 | 253 | 258 | 328 | 250 |

Table 8. Statistical analysis of Nevatim PBL heights (N=72, Fig. 7a)

| PBL detection | IFS-pm | COSMO-pm | IFS-ri | COSMO-ri | Ceilometer |
|---|---|---|---|---|---|
| Mean Error (m) | 149 | 186 | 214 | 264 | - |
| RMSE (m) | 423 | 436 | 369 | 488 | - |
| R | 0.1 | 0.15 | 0.30 | 0.23 | - |
| Mean PBL (m a.s.l) | 1728 | 1756 | 1792 | 1843 | 1579 |
| STD PBL (m) | 341 | 352 | 268 | 394 | 237 |

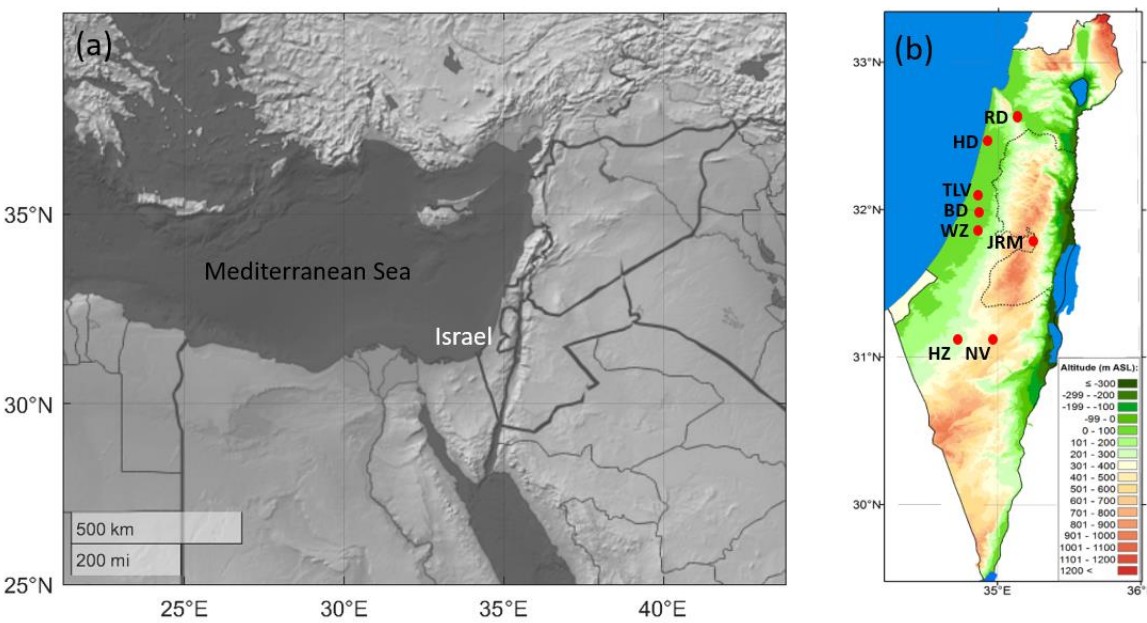

Fig. 1 Maps of the East Mediterranean (a), and the study region in Israel (b), with indications

of the ceilometers' measurement sites (red circles, details given in Table 1) on a topography

map, adapted from © Israeli meteorological service.

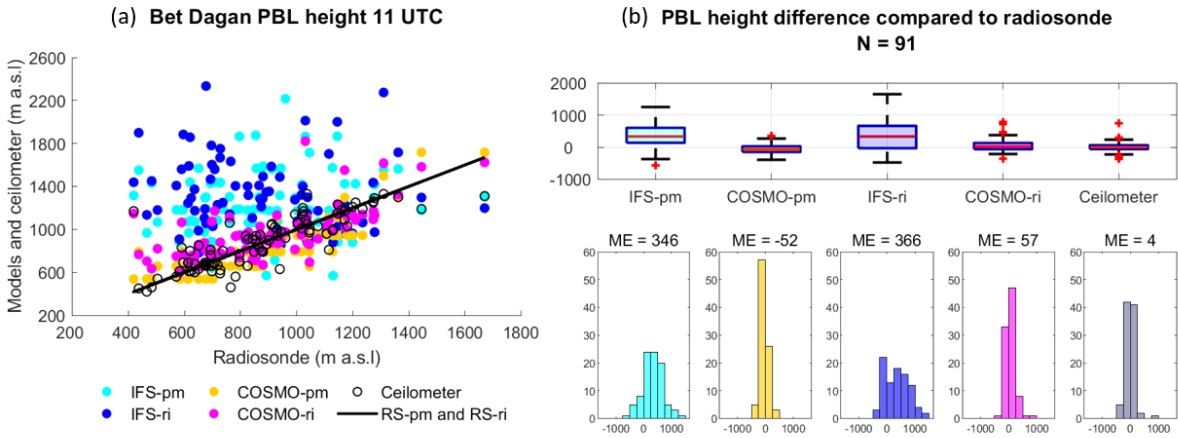

Fig. 2 PBL height from Bet Dagan site at 11 UTC on 91 days for periods of July-September 2015 and June-September 2016. Ceilometer profiles analyzed by the WCT method. The IFS, COSMO, and radiosonde profiles analyzed by the bulk Richardson method (RS-ri, IFS-ri, COSMO-ri) and the parcel method (RS-pm, IFS-pm, COSMO-pm). The results compared to the radiosonde (RS-ri and RS-pm produced the same heights). Statistical analysis of the scatter plot (a) is given in Table 3. PBL height difference presented by boxplots and histograms (b). The edges of the boxplot are the 25th and 75th percentiles (q1 and q3), the whiskers enclose all data points not considered outliers (red crosses). A central red line indicates the median. Each boxplot is described by a histogram beneath.

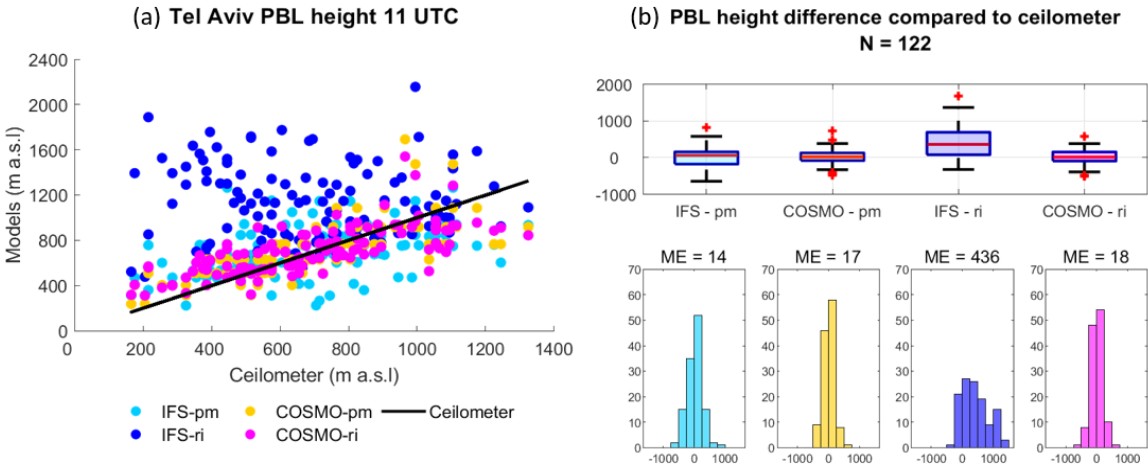

Fig. 3 Same as Fig. 2 but for Tel Aviv on 122 days. The models were compared to the
ceilometer.

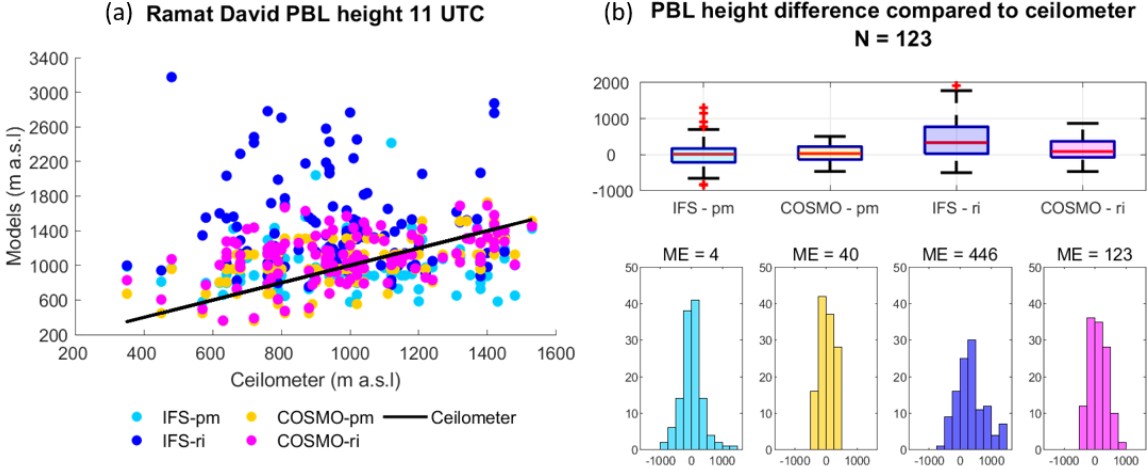

Fig. 4 Same as Fig. 2 but for Ramat David on 123 days. The models were compared to the
ceilometer.

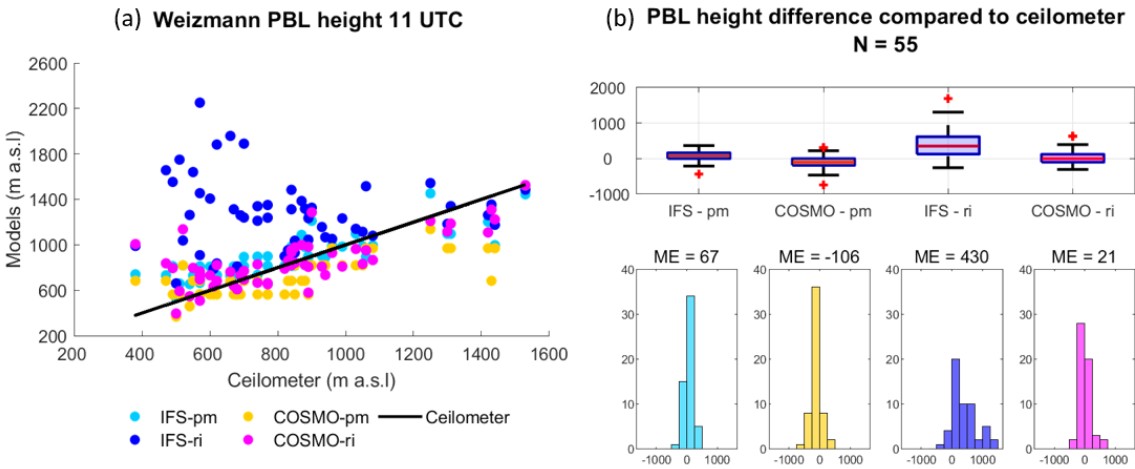

Fig. 5 Same as Fig. 2 but for Weizmann on 55 days. The models were compared to the ceilometer.

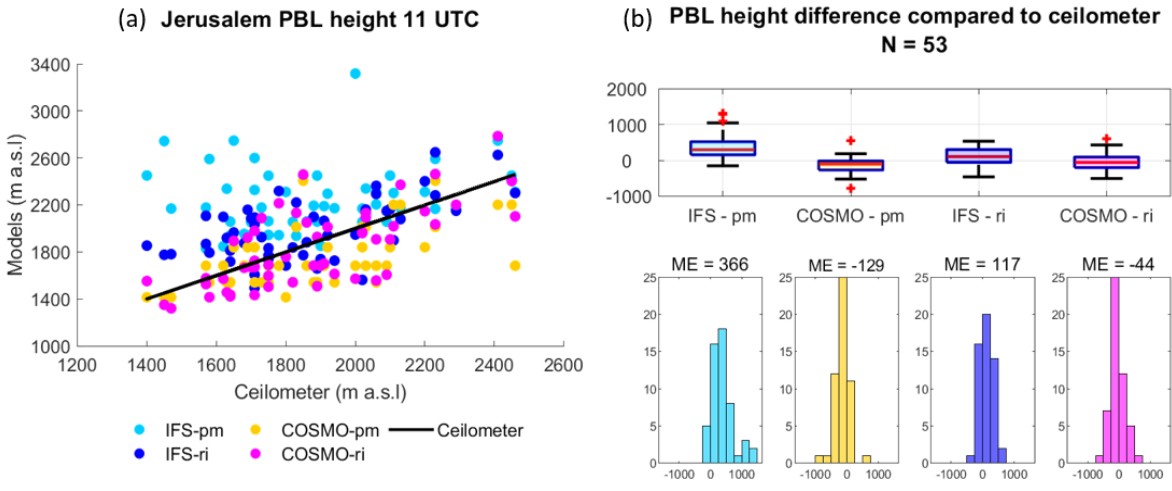

Fig. 6 Same as Fig. 2 but for Jerusalem on 53 days. The models were compared to the ceilometer.

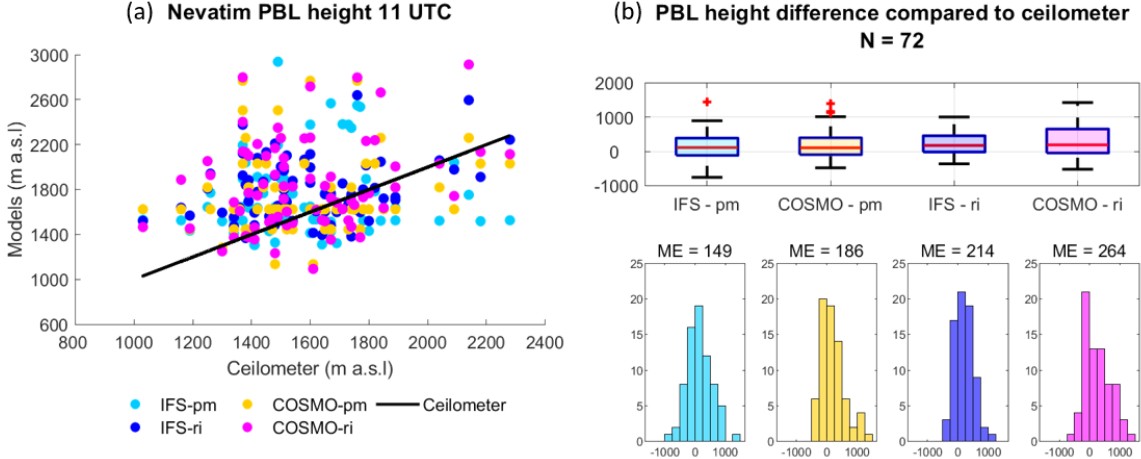

Fig. 7 Same as Fig. 2 but for Nevatim on 72 days. The models were compared to the ceilometer.

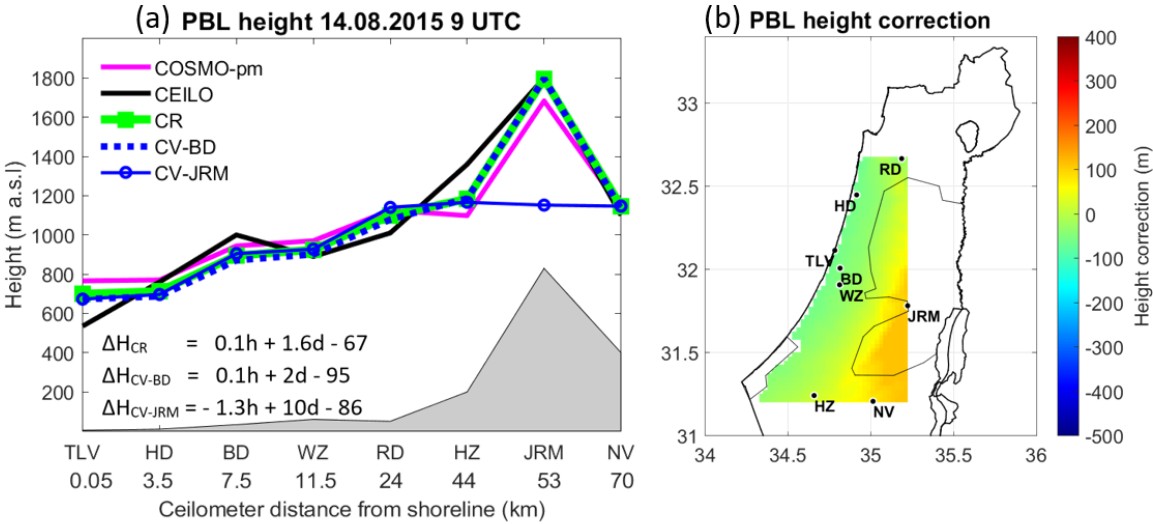

Fig. 8 PBL heights on August 14, 2015, at 9 UTC. The left panel (a) presents an east-west cross-section map, according to the ceilometers' distance from the Mediterranean shoreline. The PBL heights were derived from COSMO-pm (pink line), the ceilometers (black line), the correction tool for COSMO-pm (CR, green line), cross-validation for Bet Dagan (CV-BD, dashed blue line), and cross-validation for Jerusalem (CV-JRM, blue circles). The right panel (b) shows a 2-D map (b) of the height correction range, corresponding to figure (a).

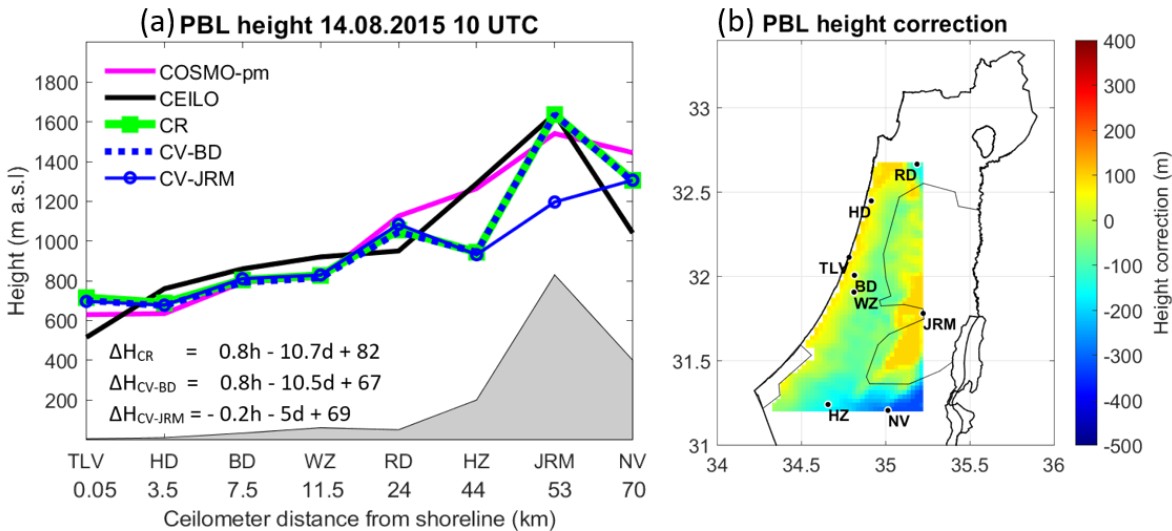

Fig. 9 Same as Fig. 8 but for 10 UTC.

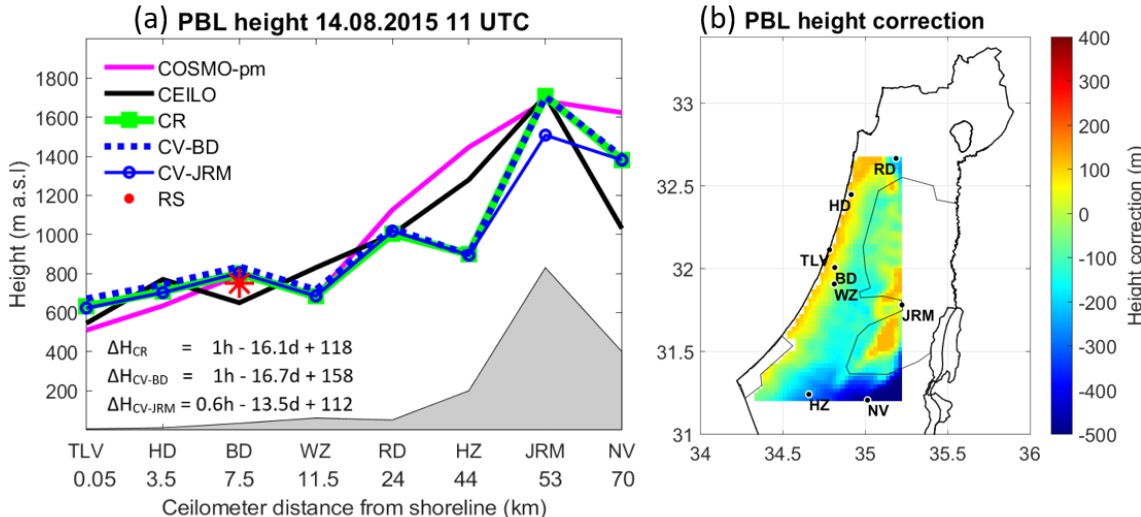

Fig. 10 Same as Fig. 8 but for 11 UTC and including the PBL height estimation from the

radiosonde (red star).

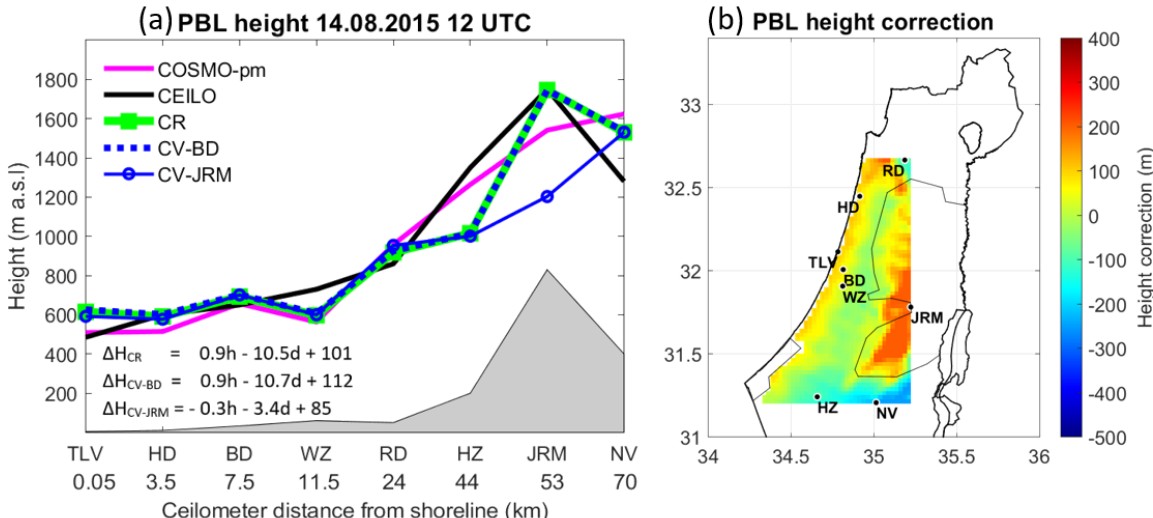

Fig. 11 Same as Fig. 8 but for 12 UTC.

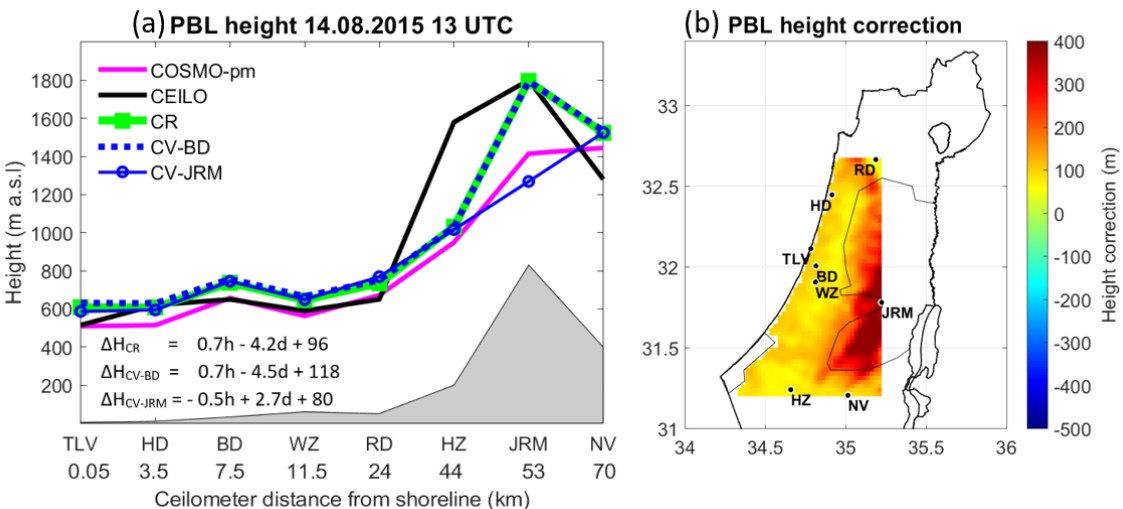

Fig. 12 Same as Fig. 8 but for 13 UTC.

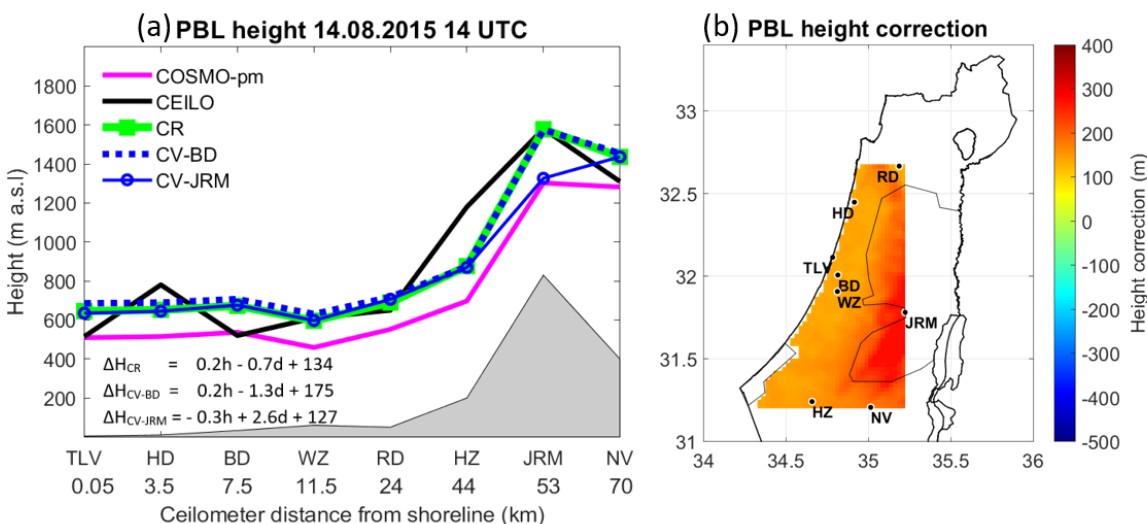

750

Fig. 13 Same as Fig. 8 but for 14 UTC.

752

754