# Peer review of "Ceilometers as planetary boundary layer height detectors and a corrective tool for COSMO and IFS models"

_Atmospheric Chemistry and Physics, 2019_

## Referee Comment (RC1) · Anonymous Referee #1 · 30 Nov 2019

The manuscript deals with the detection of the planetary boundary layer (PBL) height with ceilometers. The measurements are compared with results of atmospheric modelling and with the daily 11 UTC PBL height derived from radiosonde profiles of meteorological parameters.

The main topic of the paper is not new, but some new aspects are given in this paper and may justify publication.

The study is based on ceilometer observations in Israel. The PBL diurnal cycle is strongly influenced by sea breeze effects which makes the analysis quite complicated. Surprizingly, this aspect is not explicitly mentioned, e.g., in the abstract. The se-

lected case studies, however, clearly show the impact of sea breeze effects and, all in all, leave behind a rather confusing impression of the findings, . . .after reading the manuscript (see my comments below). These sea breeze effects are obviously (partly) not considered in the weather prediction models. There is no discussion on this. The PBL heights derived from the ceilometer and radiosonde profiles are, to my opinion, mostly wrong, and are in contradiction to the traditional definition of the PBL as the lowest well-mixed layer of the troposphere.

In conclusion, the manuscript is not acceptable in the present form.

Major revisions are necessary.

Here my comments and questions in more detail:

Introduction:

P2, L46: The mentioned advantage of ceilometers over lidars must be specified! Regarding what? . . .. is the question! If I would have to select, I would take a sophisticated lidar because such a system is much more powerful concerning emitted pulse energies and the list of aerosol products is long compared to quite 'simple' and 'weak' ceilometers. So, please specify what you definitely mean, . . . with advantage! Probably low costs, robust observations, no complex adjustments and calibrations.

However, the clear disadvantage of ceilometers, operated at water vapour absorption lines around 910 nm, is that the only product you can trust is the range-corrected signal, nothing else!

Research area:

P4, L92: Please provide longitude, latitude and height above sea level for Beit Dagan already here, and where is it located (including distance) with respect to Tel Aviv and Jerusalem.

P4, L109: Please provide frequently, what UTC means in local time. Local time is

needed to better follow discussion on PBL evolution and diurnal cycle.

P4, L110-120: There is no general PBL diurnal cycle in Israel, I speculate. But you provide such an impression! The occurrence, onset, strength and impact of the sea breeze circulation depends on given meteorological conditions (marine westerly versus continental easterly air flows, low and high wind speeds, clear or cloudy conditions). The sea breeze event strongly influences the PBL diurnal cycle. All this must be carefully mentioned in the text. And what about the impact of dense desert dust layers (in the PBL and especially in the free troposphere)? Is there any PBL development when there is a dust outbreak event? So all in all, many factors seem to control the sea breeze events and the PBL cycle in Israel. Thus, please provide more details on this.

Instruments:

P6, L161, Why should single-wavelength lidars not allow the retrieval of mass concentration profiles . . . from proper profiles of particle optical properties? Sure, they can be used for this. Ok, this is not the topic of the paper. But the statement is wrong and should be removed.

The ceilometer on the other hand side cannot be used to derive proper optical and microphysical properties. That is true! A ceilometer can only be used to detect aerosol layers as a function of height. This is not much, but sufficient for PBL studies. That should be clearly mentioned.

P7, L185: Please state again where Beit Dagan is located.

P8, L184-187: It should be clearly emphasized that the radiosonde provides ONE value for the PBL height, no diurnal cycle, . . . nothing! Only a snapshot of the PBL height, a few minutes after launch, is provided by the sonde! In contrast, models can produce the diurnal cycle, and ceilometers can measure it. But all this is not shown and discussed!

Methods:

This chapter is much too long. Text book knowledge is presented in unnecessary detail.

For each method, please provide the equation, the explanation of the equation, the link to PBL height, and a proper reference. More is not needed. A short and compact section on methods is desirable.

P9, L247: This is confusing: A ceilometer is made to detect the base of the water cloud, but not to detect the cloud top height. In most cases of low level (liquid-water) clouds there is no chance to detect the cloud top! This needs to be clearly stated.

The maximum signal you measure cannot be interpreted as cloud top. This is a very erroneous statement! The maximum backscatter signal is somewhere between cloud base and cloud top. The maximum signal is at that height where the attenuation effect becomes so strong that the signal immediately drops to the sky background level. This needs to be clearly stated. The height of the maximum signal maybe 100, 300, or 1000 m below cloud top. Nobody knows!

P10 L268: Therefore also the following statement is wrong: Our algorithm denotes the PBL height as the top of the shallow cloud. As just mentioned, you are unable to see the cloud top with ceilometer, only exceptional, in cases with optically rather thin clouds. Please improve your statements. The discussion is unacceptable in the present form.

Results:

P10, L286, and Figure 3: This is the worst case you can select in a comparison paper. There is the PBL development, there is the sea breeze effect, and there is cloud evolution! As a consequence, the PBL depth is more or less undefined at these complex atmospheric conditions. Fortunately, the radiosonde temperature profile indicates the PBL height at about 800m because for this height range (from 50 – 800m) the layer is well mixed indicated by the almost height-independent virt. pot. temperature. Then the pot. Temperature strongly increases with height and prohibits vertical mixing higher up. However, in Fig.3, the PBL heights obtained by the authors (from radiosonde, ceilometer, COSMO and IFS model) are between 1000 and 2200m? This is confusing! The

PBL height is clearly not at 1000m, 1400m, 1700m, or even 2000m. So, the ceilometer result of 1700m is totally wrong to my opinion. The reason is obviously that the range-corrected signal (and the wavelet analysis) cannot be used at these cloudy conditions to detect the true PBL height. What you see is some arbitrary height where the range-corrected signal takes its maximum. . .

If the radiosonde observations of temperature, relative humidity, wind speed, and wind direction would be shown, we would have the chance to see what is going on here. But all this is not presented. Height resolved trajectory analysis would be helpful as well in the discussion of the complex meteorological conditions. Please provide at least the wind and RH profiles of the radiosnde in the figures. The reader may want to know more about the meteorological situation.

This case study is rather confusing and not helpful. Unambiguous, cloud free conditions would be desirable to check the different approaches of PBL height retrieval.

P11, L308: Again, Figure 5 shows a rather difficult case (PBL evolution plus sea breeze effect). There is obviously a marine boundary layer (with top at 600m, clearly seen by the radiosonde) and, on top, the upper part of continental PBL up to about 1500m (also visible in the radiosonde profile). But, per definition, the lower PBL counts (the lowest well mixed layer above the surface is the boundary layer, as defined by Stull 1988). And that is the marine boundary layer, indicated by the potential temperature profile and the ceilometer data. But the PBL height obtained from the ceilometer profile analysis is again around 1700 m. This is an error of more than 100%!

Please show RH and wind profiles (direction and speed) so that more information about the complex PBL develepment at sea breeze conditions is available. Again, the selected case and the discussion are rather confusing. The results are at all not convincing, and not understandable. What is then the message of the study?

Obviously the IFS model does not simulate the impact of the sea breeze impact correctly or even ignores sea breeze effects so that the continental pot. temperature profile

is obtained with this model. The IFS PBL heights seem to be in contradiction with the IFS pot. temp. profile.

The COSMO pot. temp. profile is in good agreement with the radiosonde profile and shows the PBL height at 600 m. Very stable conditions higher up are simulated with COSMO so that not vertical mixing is possible above 600 m height. Surprisingly, the COSMO PBL height is at 1700 to 2100 m. This is totally confusing! This seems to be simply a mistake! Please clarify!

P12: Is section 6.3 needed? It is a very specific regression approach, just applicable to Israel.

P12-13 The conclusions must be rewritten after clarifying all the contradictions.

As a general, summarizing remark: Both case studies are not well selected. They indicate very complex meteorological conditions. The authors do not provide sufficient meteorological information. Additional trajectory analysis would be helpful. The results are at all not convincing. What will the reader learn from such a confusing study? . . .. except that the PBL diurnal cycle is not easy to predict in areas with sea breeze effects and cloud formation over the day.

How can we then trust the findings presented in Figs. 2, 4, 6, and 7 when we have such confusing results in Figs. 3 and 5? In the present state, the paper cannot be accepted.

---

## Referee Comment (RC2) · Anonymous Referee #2 · 5 Dec 2019

Review for Atmospheric Chemistry and Physics Discussions

Ceilometers as planetary boundary layer detectors and a corrective tool for ECMWF and COSMO NWP models, *by  L. Uzan et al.*

**General comments**

This paper addresses the daytime summer PBL height throughout Israel, by combining measurements from eight ceilometers,  radiosonde profiles (at one location), along with simulations from the global IFS and the regional COSMO models. In particular, it focuses on the analysis of three PBL height evaluation methods: the bulk Richardson method, the parcel method, and the wavelet covariance transform method.

Although there is no doubt that determining the mixing height is important,  the scientific community has done extensive research and progress so far on the daily boundary layer. However, there are still significant problems as well as gaps in the night boundary layer (stable conditions) and in the transitional periods. These periods cannot be omitted in a study when referring to the importance of mixing height in the formulation of concentrations and even more when one of the main initiatives is to designate ceilometers as a correction for NWP. The statement on line 218 is not appropriate for the exclusion of the nighttime period. Also, the methodology applied by the authors for the reliability of PBL estimation from ceilometers data, raises many reservations. I personally could not find the value of this research effort. Concluding, I believe that the whole processing of the subject is rather limited, covers a very short period and it is of local interest only. Therefore, I do not agree that this study is published to the ACP Journal.

**More specific comments**

- I wonder if we could perform a similar exercise for an area with restricted characteristics, thus no general applicability. This is the case here, where local flows are developed but there are not taken into consideration. In particular, both sea breeze and/or anabatic winds are expected to develop in this area during the summer period. For this reason, I am not sure what ceilometer is measuring.  For example, at the station of DB just 7km away from the shore, the PBL depth is measured at 1km. To my knowledge, this is an unrealistic value (too high) under the presence of sea breeze (or IBL). Thus, I wonder if this instrument finally shows the off-shore current of the sea breeze flow.
- The PBL depth is a non-specific parameter, the definition and estimation of which is not straightforward. The simulated PBL depths are mainly determined, based on the definition that each PBL scheme applies (*in this study, no information is provided regarding the PBL parameterization schemes considered by the two models*). This also applies between measurements from different instruments (ceilometer and radiosonde) as they do not have the same operating principles.
  The ceilometer measurements mainly present the mixed PBL that does not always coincide with the simulated PBL depth.

On the other hand, has it been taken into account that the radiosonde moves with the flow? As it ascends the measurements do not correspond to the vertical position above the launch point. This is another reason of possible discrepancy between the radiosonde and ceilometer measurements.

- Therefore, the same criteria should be used for the estimation of both measured and simulated PBL depth. In particular, the same criteria should be applied to the profiles of certain atmospheric parameters, such as temperature, wind and mixing ratio profiles that depict the atmospheric boundary layer structure. These criteria should not necessarily be the same for all the atmospheric conditions. For example, the gradient of potential temperature profile is inadequate to provide the turbulent ABL depth. Therefore, for the comparison with ceilometer, it would be more appropriate to consider the eddy-viscosity simulated profiles or even better the aerosol layering from chemistry transport model simulation.

- In particular, under convective conditions, the mixing height determined by ceilometer is strongly related to the aerosol stratification.

- How much value does the global model have in such a small analysis to take part in the comparison, especially in a strongly heterogeneous area?

Also, there are several arbitrary statements on the text, without any justification (no measurements of wind speed and direction are provided) or any reference. For example: line 105, As a result, the average PBL height is comparatively low (∼ 1000 m a.g.l)

line 116, Through the day, the sea breeze circulation steers clockwise and the wind speed is enhanced by the west-north-west synoptic winds

line 119, Due to the large distance (∼30-50 km inland), the SBF reaches the eastern elevated complex terrain only in the afternoon (∼ 11-12 UTC)

line 170, However, the PBL detection algorithm utilized here (see Sect. 5.3) is based on a significant signal slope, therefore can be determined from uncalibrated ceilometers.

Line 203, Does the bulk Richardson refer to a certain height or a layer?

Lines 265-end of this paragraph. I am confused.

---

## Author Comment (AC1) · 16 Jan 2020

Author's Response to referee #1: We wish to thank referee #1 for the comprehensive and constructive comments providing the opportunity to improve our manuscript. The comments led to a major revision of the manuscript. For convenience, our response is given by order of appearance following the structure of the manuscript.

1.INTRODUCTION

Referee's comment: P2, L46: The mentioned advantage of ceilometers over lidars must be specified! Regarding what? .... is the question! If I would have to select,

[Figure]

I would take a sophisticated lidar because such a system is much more powerful concerning emitted pulse energies and the list of aerosol products is long compared to quite 'simple' and 'weak' ceilometers. So, please specify what you definitely mean, ... with advantage! Probably low costs, robust observations, no complex adjustments, and calibrations. However, the clear disadvantage of ceilometers, operated at water vapour absorption around 910 nm, is that the only product you can trust is the range-corrected signal, nothing else!

Author's response: Thank you for your remark. An explanation was added to the introduction section as well as to the section describing the instrument (Sect.4.1).

Author's changes in manuscript: Additional text in Sect.1 (Introduction): " Applicable evaluation of PBL heights can be derived either by actual measurements or estimations based on numerical weather prediction (NWP) models. On the one hand, NWP models, such as regional models, provide high temporal and spatial data resolution beyond the capability of actual measurements. On the other, they are based on mathematical equations with initial assumptions and boundary conditioned set beforehand. Hence, the models' products require a systematic validation tool based on actual measurements. There are two main PBL height measurement methods: in-situ radiosonde launches and remote sensings such as lidars and profilers. Unfortunately, radiosonde launches are costly as successive measurements. Profilers and sophisticated lidars produce high temporal resolution profiles but are limited in space. Moreover, certain meteorological conditions may reduce their performance, such as precipitation for radio acoustic sounding systems and dust storms for Raman lidars. These limitations have led several research groups to successfully utilized ceilometers, single wavelength cloud base height detectors, as a means to recognize and determine the PBL height (Eresmaa et al., 2006, Haeffelin and Angelini, 2012, Wiegner et al., 2014). Ubiquitous in airports and meteorological service centers worldwide, ceilometers obtain a large spatial resolution per lidar (for further information see TOPROF of COST Action ES1303 and E-PROFILE of the EUMETNET Profiling Program). They produce

high temporal resolution profiles about every 15 s and every 10 m, up to several km, retrieved as attenuated backscatter signals. The ceilometers are low cost, easy to maintain, and operate continuously unattended under diverse meteorological conditions (Kotthaus and Grimmond, 2018). These qualities reflect their advantages over high-cost, multi-wavelength sophisticated lidars, which require surveillance, calibration procedures, and careful maintenance. Hence, they are limited in amount and operational time (Mamouri et al., 2016) and cannot produce the spatial and temporal measurements coverage essential to validate the PBL heights generated by NWP models.". Additional text in Sect.4.1 (Ceilometers): "The PBL height detection is based on a pronounced change of the attenuated backscatter profile. This change is attributed to variations in the aerosol content providing indications for both clouds and atmospheric layers. Therefore, the limitation of a single wavelength within the spectral range of water vapor absorption does affect the PBL height detection. Nevertheless, Weigner et al., (2014) succeeded to properly derive the backscatter coefficient from ceilometers, providing signal calibrations and corrections for water vapor (Wiegner and Gasteiger, 2015)".

2.RESEARCH AREA

Referee's comment: P4, L92: Please provide longitude, latitude and height above sea level for Beit Dagan already here, and where is it located (including distance) with respect to Tel Aviv and Jerusalem.

Author's response: The location and topography of Beit Dagan were given in Fig. 1 and Table 1. Following the referee's remark, the radiosonde parameters were added to the text given in Sect. 4.2. Author's changes in manuscript: Text in Sect. 4.2 (Radiosonde): "The Israeli Meteorological Service (IMS) obtains systematic radiosonde atmospheric observations twice daily, at 23 UTC and 11 UTC, adjacent to a ceilometer. Launching is performed in Beit Dagan (32.0 ° long, 34.8 ° lat, 33 m a.s.l), situated 7.5 km east from the shoreline, 11 km southeast to Tel Aviv, 45 km northwest to Jerusalem (Fig.1 and Table 1)". the title of Table 1 was changed: "Location of measurement sites and

ceilometer types". An affiliation was added to the Beit-Dagan stating it is the location of the radiosonde launch site and ceilometer measurements. The caption of Fig. 1 was changed to:". . . The Radiosonde launch site is situated in Beit Dagan, adjacent to the ceilometer ".

Referee's comment: P4, L109: Please provide frequently, what UTC means in local time. Local time is needed to better follow the discussion on PBL evolution and the diurnal cycle.

Author's response: Comment accepted. Author's changes in manuscript: UTC was corrected to LST winter time (corresponding to UTC+2) in the paragraph describing the Israeli summer PBL evolution (Sect.2 Research area).

Referee's comment: P4, L110-120: There is no general PBL diurnal cycle in Israel, I speculate. But you provide such an impression! The occurrence, onset, strength, and impact of the sea breeze circulation depend on given meteorological conditions (marine westerly versus continental easterly air flows, low and high wind speeds, clear or cloudy conditions). The sea breeze event strongly influences the PBL diurnal cycle. All this must be carefully mentioned in the text. And what about the impact of dense desert dust layers (in the PBL and especially in the free troposphere)? Is there any PBL development when there is a dust outbreak event? So all in all, many factors seem to control the sea breeze events and the PBL cycle in Israel. Thus, please provide more details on this.

Author's response: The description of the PBL diurnal cycle refers solely to the Israeli summer as stated in the text (line 105): "Comprehensive research of the Israeli summer PBL. . .". In the summer, the east Mediterranean is dominated by rather persistent synoptic systems explained in lines 104-107: " . . . a persistent Persian Trough (either deep, shallow or medium) followed by a Subtropical High aloft (Alpert et al., 2004)", combined with the sea breeze circulation. These conditions generate the PBL height diurnal cycle described in the manuscript and presented in Fig.1 and Fig.2 from Levy

et. al., (2011) and Uzan et. al., (2012), respectively. In both figures, the diurnal PBL was obtained signal to noise measurements and virtual temperature profiles from an acoustic radar. The radar was stationed in flat terrain, 3.5 km inland from the shoreline, 51 km north to Beit Dagan. In Uzan et al, (2012) the profiles were classified by the three dominant summer synoptic systems at the time of research (Jun-Oct, 1997-1999, 2002-2005). Levi et al produced the average diurnal evolution for the month of July between 1997-1999. Concerning dust outbreak events, Alpert et al. (2002) investigated dust forcing over the eastern Mediterranean. They concluded: "Summer outbreaks of dust over the Eastern Mediterranean are relatively rare. This area gets frequent intrusions of dust in spring (Alpert and Ziv 1989; Alpert et al. 2000; Moulin et al. 1997) with a secondary maximum in the autumn (Ganor 1994). The dynamical system that transports the dust is primarily the Sharav cyclone, which is also called the Saharan depression, generated in the lee of the Atlas Mountains (Egger et al. 1995) and moving along the North African coast eastward (Alpert et al. 1990b). The Sharav cyclone is clearly not the associated synoptic system in summer". Moreover, dust layers that were evident over Israel in the summer were located in high altitudes.

Author's changes in manuscript: Following the referee's remark, we rephrased the text to emphasize the description of the PBL diurnal cycle refers only to the Israeli summer season: " Previous research describes the formation and evolution of the Israeli summer PBL height as a function of the synoptic and mesoscale conditions, as well as the distance from the shoreline, and the topography. Overall, the diurnal PBL height in the summer season may be portrayed in the following manner.." Details about the occurrence of dust events in the summer were added to the text: " The Israeli summer season (June-September) is characterized by dry weather (no precipitation), high relative humidity (RH, up to 80% in midday in the shoreline, IMS weather reports) and sporadic shallow cumulus clouds. On the synoptic scale, the summer is defined by a persistent Persian Trough (either deep, shallow or medium) followed by a Subtropical High aloft (Felix Y., 1994, Dayan et al., 2002, Alpert et al., 2004). Combined with the sea breeze, the average PBL height is found to be quite low. For example, the PBL

height in Beit Dagan (33 m a.s.l and 7.5 km east from the shoreline) reaches ∼900 m a.g.l after sunrise, and before the entrance of the sea breeze front (Felix Y.,1994, Dayan and Rodinzki, 1999, Uzan et al., 2016, Yuval et al., 2019). At this height level dust plumes do not exist (Alpert et al., 2002) as summer dust outbreaks in the eastern Mediterranean are quite rare (Alpert and Ziv 1989, Alpert et al., 2000)".

(3.IFS AND COSMO MODELS- no comments)

4.INSTRUMENTS

Referee's comment: P6, L161: Why should single-wavelength lidars not allow the retrieval of mass concentration profiles ... from proper profiles of particle optical properties? Sure, they can be used for this. Ok, this is not the topic of the paper. But the statement is wrong and should be removed. The ceilometer on the other hand side cannot be used to derive proper optical and microphysical properties. That is true! A ceilometer can only be used to detect aerosol layers as a function of height. This is not much, but sufficient for PBL studies. That should be clearly mentioned.

Author's response: In order to differentiate and define the composition of atmospheric aerosols, various wavelengths corresponding to different characteristics are necessary. Weigner et al., (2014) further explains: "Whereas the detection of aerosol layers and their vertical extent requires only simple single-wavelength backscatter lidars, the derivation of extinction coefiňĄcient profiles and a series of intensive aerosol properties requires advanced lidar concepts such as high-spectral resolution lidars (HSRL, Shipley et al., 1983) or Raman lidars (Ansmann et al., 1992)". Nonetheless, Weigner succeeded to produce satisfactory estimations of the attenuated coefficient based on signal calibrations and corrections for water vapor absorption (Weigner and Gasteiger, 2015).

Author's changes in manuscript: "The PBL height detection is based on a pronounced change of the attenuated backscatter profile. This change is attributed to variations in the aerosol content providing indications for both clouds and atmospheric layers.
Therefore, the limitation of a single wavelength within the spectral range of water vapor absorption does affect the PBL height detection. Nevertheless, Weigner et al., (2014) succeeded to properly derive the backscatter coefficient from ceilometers, providing signal calibrations and corrections for water vapor (Wiegner and Gasteiger, 2015).

Referee's comment: P7, L185: Please state again where Beit Dagan is located. P8, L184-187: It should be clearly emphasized that the radiosonde provides ONE value for the PBL height, no diurnal cycle, ... nothing! Only a snapshot of the PBL height, a few minutes after launch is provided by the sonde! In contrast, models can produce the diurnal cycle, and ceilometers can measure it. But all this is not shown and discussed!

Author's response: Lines 184-185 state:" Radiosonde (RS) type....is launched twice daily at 23 UTC and 11 UTC by the IMS in the Beit Dagan site, adjacent to the ceilometer". The time differences between the models and the ceilometers were mentioned in the text as follows: P 5, lines 146-147: "IFS profiles were limited to hourly resolution, while COSMO generated profiles every 15 minutes. To compare COSMO's PBL heights, a series of trials were performed to find the correct representation of hourly values as the last 15 minutes within an hour". P 6, lines 179-181: "To compare the hourly results of the models (Sect. 3), the ceilometers' 15 seconds profiles were averaged to half-hour ones, whereas the second half-hour profile within each hour was chosen". Nonetheless, the relevant sections were rephrased to create a clearer explanation.

Author's changes in manuscript: Sect 4.2 (Radiosonde) was rephrased with additional information: "The Israeli Meteorological Service (IMS) obtains systematic radiosonde atmospheric observations twice daily, at 23 UTC and 11 UTC, adjacent to a ceilometer. Launching is performed in Beit Dagan (32.0 ° long, 34.8 ° lat, 33 m a.s.l), situated 7.5 km east from the shoreline, 11 km southeast to Tel Aviv, 45 km northwest to Jerusalem (Fig.1 and Table 1). The radiosonde, type Vaisala RS41-SG, produces profiles of RH, temperature, pressure, wind speed and wind direction as it ascends. Measurements are retrieved every 10 seconds, corresponding to about every 45 m, reaching 2 km in about 8 minutes. The horizontal displacement of the radiosonde depends on the inten-
sity of the ambient wind speed. In this study, we analyzed the PBL height of midday summer profiles (11 UTC). The average wind speed along these profiles is about 5 m/s (Uzan et al., 2012). Therefore, the horizontal displacement of the radiosonde from its launch position is fairly low and is estimated at about 2.5 km. Moreover, the radiosonde position resolution is defined as 0.01°. As aforementioned, the PBL height in Beit Dagan for midday summer is estimated below 1 km (Dayan and Rodinzki, 1999, Uzan et al., 2016, Yuval et al., 2019). Hence, within an ascending height of 1 km, there could only be a change of 0.01° in the radiosonde position. This spatial error is in the order magnitude of the models' grid resolution. Thus, we assert the radiosonde profiles represent the Beit Dagan site and the displacement error of the ascending radiosonde can be neglected". A text was added to Sect. 6.1 (Comparison to in-situ radiosonde profiles): "Statistical analysis of the Beit Dagan PBL heights mean error (ME), root mean square error (RMSE), and correlation (R) is presented in Fig. 2 and Table 3 for 11 UTC. The analysis was based on the comparison between radiosonde measurements at 11 UTC, to Beit Dagan ceilometer average profiles between 10:30-11:00 UTC, IFS estimations for 11 UTC and COSMO results for 10:45 UTC".

5.METHODS

Referee's comment: This chapter is much too long. Textbook knowledge is presented in unnecessary detail. For each method, please provide the equation, the explanation of the equation, the link to PBL height, and a proper reference. More is not needed. A short and compact section on methods is desirable.

Author's response: Comment accepted.

Author's changes in manuscript: The method section was edited in a concise manner.

Referee's comments: P9, L247: This is confusing: A ceilometer is made to detect the base of the water cloud, but not to detect the cloud top height. In most cases of low level (liquid-water) clouds, there is no chance to detect the cloud top! This needs to be clearly stated. The maximum signal you measure cannot be interpreted as a

cloud top. This is a very erroneous statement! The maximum backscatter signal is somewhere between the cloud base and cloud top. The maximum signal is at that height where the attenuation effect becomes so strong that the signal immediately drops to the sky background level. This needs to be clearly stated. The height of the maximum signal maybe 100, 300, or 1000 m below the cloud top. Nobody knows! P10 L268: ...Therefore, also the following statement is wrong: Our algorithm denotes the PBL height as the top of the shallow cloud. As just mentioned, you are unable to see the cloud top with ceilometer, only exceptional, in cases with optically rather thin clouds. Please improve your statements. The discussion is unacceptable in the present form.

Author's response: Thank you for this important remark. In this research, we employed the wavelet covariance transform (WCT) method on the ceilometers' backscatter profiles. The principle of this method is to calculate the derivatives between measuring points along the length of the backscatter profile. The highest derivative implies a profound difference in the atmospheric aerosol content. On clear days, this difference occurs as the transmitted light exits the well-mixed layer and enters the stable layer above. In the presence of clouds, the highest values are retrieved at cloud base height which is considered as the mixed layer height. The cloud top denotes the bottom height of the free atmosphere (Fig.3 from Stull, 1988). Therefore, in order to generate a consistent definition of the PBL height by the WCT method, our algorithm seeks the height of the transition zone in the presence of clouds as well. This height is defined here as the highest measuring point of a cloud above the cloud base height. Even though the summer clouds are relatively shallow ($\sim$ 500 m thickness based on observations, see example in Fig.4 and Fig.5), there is no guarantee the algorithm detects the actual cloud top. Therefore, to prevent misinterpretations, the phrase "cloud top" was omitted and clarified as the highest measurement point of a cloud above a cloud base height.

Author's changes in manuscript: " When clouds are present (mainly summer shallow cumulous), the algorithm defines the highest measurement point of a cloud (above

the cloud base height) as the height where the signal counts decrease to the amount retrieved by background values. This signifies the ceilometer's identification of the entrainment zone (Stull, 1988)".

6.RESULTS

Referee's comment: P10, L286, and Figure 3: This is the worst case you can select in a comparison paper. There is the PBL development, there is the sea breeze effect, and there is cloud evolution! As a consequence, the PBL depth is more or less undefined at these complex atmospheric conditions. . . This case study is rather confusing and not helpful. Unambiguous, cloud-free conditions would be desirable to check the different approaches of PBL height retrieval.

Author's response: We analyzed a total of 33 cases and received good results for the majority of the data (cases of either cloud-free or sporadic shallow cumulus clouds). The largest gaps between the models' estimations and the radiosonde measurements were found on August 17, 2016, presenting an uncommon multi-layer summer cloud. As the referee correctly discerned, this complex meteorology explains the large gaps between the models and the instruments. We agree with the referee for the necessity to present a case reflecting the ability of the method. Therefore, we generated a new figure demonstrating a typical event to explain the method rather than the extraordinary results of Aug 17, 2016.

Author's changes in manuscript: Figures 3 and 5 were removed and a new figure from August 15, 2015, was added representing a typical event (given here as Fig 6).

Referee's comment: P10, L286, and Figure 3: Fortunately, the radiosonde temperature profile indicates the PBL height at about 800m because for this height range (from 50 – 800m) the layer is well mixed indicated by the almost height-independent virt. pot. temperature. Then the pot. Temperature strongly increases with height and prohibits vertical mixing higher up. However, in Fig.3, the PBL heights obtained by the authors (from radiosonde, ceilometer, COSMO and IFS model) are between 1000 and 2200m?

This is confusing! The PBL height is clearly not at 1000m, 1400m, 1700m, or even 2000m. So, the ceilometer result of 1700m is totally wrong to my opinion. The reason is obviously that the range-corrected signal (and the wavelet analysis) cannot be used at these cloudy conditions to detect the true PBL height. What you see is some arbitrary height where the range-corrected signal takes its maximum...

Author's response: The referee indicated the PBL height as the highest point aloft before the virt. pot. temperature increases. Following Stull (1988, Chapter 5, paragraph 5.5, see attached Fig.7), and the parcel method (Holzworth 1964, Seidel et al., 2010) we indicated the PBL height as the height where the virtual potential temperature reaches the value that of the surface level. By this method, the PBL height is indicated as the height where the passage from the unstable layer to the stable layer above occurs. The unstable layer is defined by the mixed layer and the entrainment zone above. This definition corresponds with the height point at which an abrupt change is measured by the ceilometers, at the transition zone between the well-mixed layer and the free atmosphere above.

Author's changes in manuscript: No changes were made in the manuscript.

Referee's comment: P10, L286, and Figure 3: If the radiosonde observations of temperature, relative humidity, wind speed, and wind direction would be shown, we would have the chance to see what is going on here. But all this is not presented. Height resolved trajectory analysis would be helpful as well in the discussion of the complex meteorological conditions. Please provide at least the wind and RH profiles of the radiosonde in the figures. The reader may want to know more about the meteorological situation.

Author's response: Comment accepted. Profiles of temperature, RH wind speed and wind direction from the adjacent radiosonde launch site are given here in Fig.8.

Author's changes in manuscript: Additional plots presenting radiosonde profiles of wind speed, wind direction, relative humidity, and temperature were added to the typical

case on Aug 15, 2015.

Referee's comment: P11, L308: Again, Figure 5 shows a rather difficult case (PBL evolution plus sea breeze effect). There is obviously a marine boundary layer (with the top at 600m, clearly seen by the radiosonde) and, on top, the upper part of continental PBL up to about 1500m (also visible in the radiosonde profile). But, per definition, the lower PBL counts (the lowest well-mixed layer above the surface is the boundary layer, as defined by Stull 1988). And that is the marine boundary layer, indicated by the potential temperature profile and the ceilometer data. But the PBL height obtained from the ceilometer profile analysis is again around 1700 m. This is an error of more than 100%! Please show RH and wind profiles (direction and speed) so that more information about the complex PBL development at sea breeze conditions is available. Again, the selected case and the discussion are rather confusing. The results are at all not convincing, and not understandable. What is then the message of the study? Obviously, the IFS model does not simulate the impact of the sea breeze impact correctly or even ignores sea breeze effects so that the continental pot. temperature profile is obtained with this model. The IFS PBL heights seem to be in contradiction with the IFS pot. temp. profile. The COSMO pot. temp. profile is in good agreement with the radiosonde profile and shows the PBL height at 600 m. Very stable conditions higher up are simulated with COSMO so that not vertical mixing is possible above 600 m height. Surprisingly, the COSMO PBL height is at 1700 to 2100 m. This is totally confusing! This seems to be simply a mistake! Please clarify!

Author's response: We deeply apologize for this clerical error. The referee is correct. Fig. 5 contains a grave mistake. Unfortunately, the data of PBL heights of Aug 17, 2016, were mistakenly presented also for Aug 10, 2015 inevitably causing a disagreement between the virt. pot. temperature profiles, and the PBL height indicated upon the ceilometer figure. A correct figure is given in (Fig. 9) including meteorological profiles from the adjacent radiosonde (Fig.10).

Author's changes in manuscript: The corrected figure including the meteorological conditions for each study case are given in the point to point response, but not in the manuscript. Following the referee's suggestion, they were replaced by a representative case of the method on August 15, 2015.

Referee's comment: P12: Is section 6.3 needed? It is a very specific regression approach, just applicable to Israel.

Author's response: Sect. 6.3 suggests a new approach to correct COSMO PBL height estimations by ceilometers. Actually, that is the goal of the research. The method proved as an applicable tool to validate and even correct the model's estimations. In regions with scarce profiling, there are no other alternatives to validate the model's results. Considering the simplicity of the method, it can be easily adapted in similar topographical areas by adjusting the correction factors (Eq. 6).

Author's changes in manuscript: The paragraph was rephrased to emphasize the advantage and importance of the suggested method.

Referee's comment: P12-13 The conclusions must be rewritten after clarifying all the contradictions.

Author's response: Comment accepted.

Author's changes in manuscript: The Conclusions paragraph was rephrased accordingly.
* * *
Author's comment: In the process of responding to the referees' comments, we repeatedly examined our datasets and evaluations of the equations of each method. We found that the virtual temperature and the virtual potential temperature employed values of $Rd/Cp = 287/1004$ ($\sim = 0.28586$) and surface pressure of $Po = 1000$ mb for the radiosonde data. On the other hand, in both models, these factors were defined as $Rd/Cp = 0.263$, $Po = 1013.15$ mb. Therefore, we decided to modify the factors assimilated on the models to the same values given for the radiosonde data ($Rd/Cp =$

287/1004, P0 = 1000 mb). Essentially, the updated values did not change the correction method (which was based on the bulk Richardson method) or the conclusions of the research, but it altered the models' results based on the parcel method as presented below (Tables 3-5 and Fig. 11 ):
* * ** * *
[Figure]

Fig.1   The average diurnal evolution of the boundary layer height during Julys of 1997–99 as defined by the height of SNRmax value (filled circles) together with the upper and lower 95% confidence limits. The inversion in the corrected virtual temperature $T_v$ measured by the RASS is indicated by open squares.

(Source: Levi et al., 2011)

**Fig. 1.** The diurnal summer MLH

[Figure]

Fig.2 Lap-3000 profiler results of the average PBL height for the three
main synoptic systems, Persian trough weak (PT-W, blue line, an average
of 347 days), Persian trough medium (PT-M, green line, an average of
232 days) and High to the west (H-W, red line, an average of 198 days),
during June-October 1997-1999,2002-2005. Also indicated are times of
sunrise and sunset, maximum solar radiation and SBF entrance.
(Source: Uzan et al., 2012)

**Fig. 2.** The diurnal summer MLH

[Figure]

Fig.3   The troposphere can be divided into two parts: a boundary layer (shaded) near the surface and the free atmosphere above it.

(Source: Stull, 1988)

**Fig. 3.** PBL illustration

[Figure]

Fig.4 IMS photograph of the sky over Beit Dagan site on August 2, 2019, at 8 UTC
presenting typical shallow cumulus clouds.

**Fig. 4.** Cumulus clouds - sky vision

[Figure]

Fig.5 Terra-MODIS 250 m resolution picture over Israel on August 2, 2019, at 8 UTC.
Beit Dagan site is indicated by a red dot. Adapted from @NOAA- EARTHDATA.

**Fig. 5.** Cumulus clouds - Terra Modis

[Figure]

Fig.6 Meteorological measurements from Beit Dagan site on August 15, 2015: Virtual potential temperature profiles at 11 UTC generated from radiosonde measurements, IFS and COSMO models (a), ceilometer signal counts plot including indications of the PBL heights at 11 UTC from the models (IFS$_R$, IFS$_P$, COSMO$_R$, COSMO$_P$), radiosonde (RS$_R$, RS$_P$) and ceilometer (b). The bottom panel presents radiosonde profiles of temperature, RH, wind speed and wind direction at 11 UTC (c).

**Fig. 6.** Analysis of Aug,15,2015

[Figure]

Fig. 7. Static stability as a function of the $\overline{\theta}_v$ profile. Dotted lines denote parcel movement.

**Fig. 7.** Virt.Pot.Temp profiles

[Figure]

[Figure]

Fig.8 Ceilometer signal counts plot on August 17, 2016, including indications of the PBL heights at 11 UTC from the models (IFS$_R$, IFS$_P$, COSMO$_R$, COSMO$_P$), radiosonde (RS$_R$, RS$_P$) and ceilometer (a). The bottom panel presents the radiosonde profiles retrieved at 11 UTC on the same day (b).

**Fig. 8.** Analysis of Aug,17,2016

[Figure]

**Fig. 9.** Correction of the analysis on Aug,10,2015

[Figure]

Fig.10 Ceilometer signal counts plot on August 10, 2015, including indications of the PBL heights at 11 UTC from the models (IFS$_R$, IFS$_P$, COSMO$_R$, COSMO$_P$), radiosonde (RS$_R$, RS$_P$) and ceilometer (a). The bottom panel presents the radiosonde profiles retrieved at 11 UTC on the same day (b).

**Fig. 10.** Analysis of Aug,10,2015

Table 3. Statistical analysis of the Beit Dagan PBL heights on 33 summer days (13 days in August 2015 and 20 days in August 2016) from IFS and COSMO models by the bulk Richardson method (IFS$_R$, COSMO$_R$), the parcel method (IFS$_P$, COSMO$_P$) and the WCT method for the adjacent ceilometer. The PBL heights were compared to those derived from Beit Dagan radiosonde by either the parcel or bulk Richardson methods (see Fig 2).

| PBL detection | IFS$_R$ | IFS$_P$ | COSMO$_R$ | COSMO$_P$ | Ceilometer |
|---|---|---|---|---|---|
| Mean Error (m) | 274 | 249 (271) | -3 | -17 (-106) | 12 |
| RMSE (m) | 432 | 409 (411) | 152 | 179 (176) | 97 |
| R | 0.18 | 0.18 (0.21) | 0.83 | 0.73 (0.83) | 0.93 |
| Mean PBL (m a.s.l) | 1250 | 1225 (1247) | 973 | 959 (869) | 989 |
| Std PBL (m) | 274 | 256 (245) | 273 | 229 (222) | 259 |

*New results are given in brackets.

Table 4. Root mean square errors of PBL heights from five sites on 13 summer days (Fig. 4), derived by IFS and COSMO models by the bulk Richardson method (IFS$_R$, COSMO$_R$) and the parcel method (IFS$_P$, COSMO$_P$). The PBL heights were compared to the heights measured by the Beit Dagan ceilometer.

| Site | IFS$_R$ | IFS$_P$ | COSMO$_R$ | COSMO$_P$ |
|---|---|---|---|---|
| Ramat David | 173 m | 191 (180) m | 247 m | 241 (232) m |
| Tel Aviv | 276 m | 465 (498) m | 203 m | 183 (182 ) m |
| Beit Dagan | 405 m | 569 (569) m | 235 m | 234 (171) m |
| Weizmann | 214 m | 274 (339) m | 175 m | 145 (209) m |
| Jerusalem | 351 m | 368 (285) m | 251 m | 273 (179) m |

*New results are given in brackets.

Table 5. Same as in Table 3 but for mean errors.

| Site | IFS$_R$ | IFS$_P$ | COSMO$_R$ | COSMO$_P$ |
|---|---|---|---|---|
| Ramat David | -31 m | 30 (0) m | -26 m | 0 (-12) m |
| Tel Aviv | 234 m | 376 (422) m | 19 m | -35 (-35) m |
| Beit Dagan | 332 m | 497 (497) m | 12 m | -9 (-55) m |
| Weizmann | 114 m | 218 (280) m | 16 m | -42 (-42) m |
| Jerusalem | 298 m | 327 (243) m | -6 m | 29 ( -1) m |

*New results are given in brackets.

**Fig. 11.** Tables 3-5

[Figure]

[Figure]

[Figure]

Fig 11. PBL heights over Beit Dagan site on 33 summer days (13 days on August 2015 and 20 days on August 2016), generated by the bulk Richardson method for IFS model (IFS$_R$, blue solid circles), COSMO model (COSMO$_R$, pink solid circles), and Beit Dagan radiosonde profiles (RS$_R$, black line). PBL heights generated by the parcel method for the IFS model (IFS$_P$, open blue circles), COSMO model (COSMO$_P$, open pink circles), and Beit Dagan radiosonde profiles (RS$_P$, same black line as RS$_R$, the results are identical). PBL heights derived from the Beit Dagan ceilometer produced by the WCT method (green circles). Results for August 17, 2016, are indicated by a circle.

**Fig. 12.** Analysis of 33 days

---

## Author Response (AR1)

We wish to thank referee #1 for the comprehensive and constructive comments providing the opportunity to improve our manuscript. The comments led to a major revision of the manuscript. For convenience, our response is given by order of appearance following the structure of the manuscript.

**1.Introduction**

Referee's comment:
P2, L46: The mentioned advantage of ceilometers over lidars must be specified! Regarding what? .... is the question! If I would have to select, I would take a sophisticated lidar because such a system is much more powerful concerning emitted pulse energies and the list of aerosol products is long compared to quite 'simple' and 'weak' ceilometers. So, please specify what you definitely mean, ... with advantage! Probably low costs, robust observations, no complex adjustments, and calibrations. However, the clear disadvantage of ceilometers, operated at water vapour absorption around 910 nm, is that the only product you can trust is the range-corrected signal, nothing else!
Author's response:
Comment accepted.
Author's changes in manuscript:
Additional text in Sect.1 (Introduction):
" Applicable evaluation of PBL heights can be derived either by actual measurements or estimations based on numerical weather prediction (NWP) models. On the one hand, NWP models, such as regional models, provide high temporal and spatial data resolution beyond the capability of actual measurements. On the other, they are based on mathematical equations with initial assumptions and boundary conditioned set beforehand. Hence, the models' products require a systematic validation tool based on actual measurements.

There are two main PBL height measurement methods: in-situ radiosonde launches and remote sensing such as lidars and profilers. Unfortunately, radiosonde launches are costly as successive measurements. Profilers and sophisticated lidars produce high temporal resolution profiles but are limited in space. Moreover, certain meteorological conditions may reduce their performance, such as precipitation for radio acoustic sounding system profilers (Uzan et al., 2012) and dust storms for Raman lidars (Mamouri et al., 2016).

These limitations have led several research groups to successfully utilized ceilometers - single wavelength cloud base height detectors, as a means to recognize and determine the PBL height (Eresmaa et al., 2006, Haeffelin and Angelini, 2012, Wiegner et al., 2014). Ubiquitous in airports and meteorological service centers worldwide, ceilometers obtain a wide spatial resolution per lidar (for further information see TOPROF of COST Action ES1303 and E-PROFILE of the EUMETNET Profiling Program). They produce high temporal resolution profiles about every 15 s and every 10 m, up to several km, retrieved as attenuated backscatter signals. The ceilometers are low cost, easy to maintain, and operate continuously unattended under diverse meteorological conditions (Kotthaus and Grimmond, 2018). These qualities reflect their advantages over high-cost, multi-wavelength sophisticated lidars, that require

surveillance, calibration procedures, and careful maintenance. Hence, they are limited in space and operational time (Mamouri et al., 2016) and cannot achieve the spatial and temporal measurements coverage essential to validate the PBL heights generated by NWP models.".

**2.Research area**

Referee's comment:
P4, L92: Please provide longitude, latitude and height above sea level for Beit Dagan already here, and where is it located (including distance) with respect to Tel Aviv and Jerusalem.
Author's response:
The location and topography of Beit Dagan were given in Fig. 1 and Table 1. Following the referee's remark, the radiosonde parameters were added to the text given in Sect. 4.2.
Author's changes in manuscript:
(1) Text in Sect. 4.2 (Radiosonde): "The Israeli Meteorological Service (IMS) obtains systematic radiosonde atmospheric observations twice daily, at 23 UTC and 11 UTC, adjacent to a ceilometer. Launching is performed in Beit Dagan (32.0 ° long, 34.8 ° lat, 33 m a.s.l), situated 7.5 km east from the shoreline, 11 km southeast to Tel Aviv, 45 km northwest to Jerusalem (Fig.1 and Table 1)".
(2) Changes in Table 1:
   a. Title: "Location of measurement sites and ceilometer types".
   b. Affiliation: Beit Dagan (BD)$_b$- [b]The location of ceilometer Beit Dagan and the radiosonde launch site.
(3) Changes in the caption of Fig. 1:" … The Radiosonde launch site is situated in Beit Dagan, adjacent to the ceilometer ".

Referee's comment:
P4, L109: Please provide frequently, what UTC means in local time. Local time is needed to better follow the discussion on PBL evolution and the diurnal cycle.
Author's response:
Comment accepted.
Author's changes in manuscript:
UTC was corrected to LST winter time (corresponding to UTC+2) in the paragraph describing the Israeli summer PBL evolution (Sect.2 Research area).

Referee's comment:
P4, L110-120: There is no general PBL diurnal cycle in Israel, I speculate. But you provide such an impression! The occurrence, onset, strength, and impact of the sea breeze circulation depend on given meteorological conditions (marine westerly versus continental easterly air flows, low and high wind speeds, clear or cloudy conditions). The sea breeze event strongly influences the PBL diurnal cycle. All this must be carefully mentioned in the text. And what about the impact of dense desert dust layers (in the PBL and especially in the free troposphere)? Is there any PBL development when there is a dust outbreak event? So all in all, many factors

seem to control the sea breeze events and the PBL cycle in Israel. Thus, please provide more details on this.

Author's response:

The description of the PBL diurnal cycle refers solely to the Israeli summer as stated in the text (line 105): "Comprehensive research of the Israeli summer PBL…".

In the summer, the east Mediterranean is dominated by rather persistent synoptic systems explained in lines 104-107: " … a persistent Persian Trough (either deep, shallow or medium) followed by a Subtropical High aloft (Alpert et al., 2004)", combined with the sea breeze circulation. These conditions generate the PBL height diurnal cycle described in the manuscript and presented in Fig. 1 and Fig.2 from Levy et. al., (2011) and Uzan et. al., (2012), respectively. In these figures, the diurnal PBL was obtained by signal to noise measurements and virtual temperature profiles from an acoustic radar. The radar was stationed in flat terrain, 3.5 km inland from the shoreline, 51 km north to Beit Dagan. In Uzan et al, (2012) the profiles were classified by the three dominant summer synoptic systems at the time of research (Jun-Oct, 1997-1999, 2002-2005). Levi et al produced the average diurnal evolution for the month of July between 1997-1999".

[Figure]

Fig.1    The average diurnal evolution of the boundary layer height during Julys of 1997–99 as defined by the height of SNRmax value (filled circles) together with the upper and lower 95% confidence limits. The inversion in the corrected virtual temperature $T_v$ measured by the RASS is indicated by open squares.

(Source: Levi et al., 2011)

[Figure]

Fig.2 Lap-3000 profiler results of the average PBL height for the three main synoptic systems, Persian trough weak (PT-W, blue line, an average of 347 days), Persian trough medium (PT-M, green line, an average of 232 days) and High to the west (H-W, red line, an average of 198 days), during June-October 1997-1999,2002-2005. Also indicated are times of sunrise and sunset, maximum solar radiation and SBF entrance. (Source: Uzan et al., 2012)

Concerning dust outbreak events, Alpert et al. (2002) investigated dust forcing over the eastern Mediterranean. They concluded: "Summer outbreaks of dust over the Eastern Mediterranean are relatively rare. This area gets frequent intrusions of dust in spring (Alpert and Ziv 1989; Alpert et al. 2000; Moulin et al. 1997) with a secondary maximum in the autumn (Ganor 1994). The dynamical system that transports the dust is primarily the Sharav cyclone, which is also called the Saharan depression, generated in the lee of the Atlas Mountains (Egger et al. 1995) and moving along the North African coast eastward (Alpert et al. 1990b). The Sharav cyclone is clearly not the associated synoptic system in summer". Moreover, dust layers that were evident over Israel in the summer were located in high altitudes.

Author's changes in manuscript:

(1) Following the referee's remark, we rephrased the text to emphasize the description of the PBL diurnal cycle refers only to the Israeli summer season: " Previous research describes the formation and evolution of the Israeli summer PBL height as a function of the synoptic and mesoscale conditions, as well as the distance from the shoreline, and the topography. Overall, the diurnal PBL height in the summer season may be portrayed in the following manner.."

(2) Details about the occurrence of dust events in the summer were added to the text:
" The Israeli summer season (June-September) is characterized by dry weather (no precipitation), high relative humidity (RH) - up to 80% in midday in the shoreline (Israeli Meteorological Service -IMS weather reports) and sporadic shallow cumulus clouds. On the synoptic scale, the summer is defined by a persistent Persian Trough (either deep, shallow or medium) followed by a Subtropical High aloft (Felix Y., 1994,

Dayan et al., 2002, Alpert et al., 2004). Combined with the sea breeze, the average PBL height is found to be quite low. For example, the average summer PBL height in Beit Dagan (33 m a.s.l and 7.5 km east from the shoreline) reaches ~900 m a.g.l after sunrise, and before the entrance of the sea breeze front (Felix Y.,1994, Dayan and Rodinzki, 1999, Uzan et al., 2016, Yuval et al., 2019).  Summer dust outbreaks in the eastern Mediterranean are quite rare (Alpert and Ziv 1989, Alpert et al., 2000) therefore, they were not addressed here, especially in the height levels below 1 km (Alpert et al., 2002)".

**(3.IFS and COSMO Models- no comments)**

**4.Instruments**

Referee's comment:

P6, L161: Why should single-wavelength lidars not allow the retrieval of mass concentration profiles ... from proper profiles of particle optical properties? Sure, they can be used for this. Ok, this is not the topic of the paper. But the statement is wrong and should be removed. The ceilometer on the other hand side cannot be used to derive proper optical and microphysical properties. That is true! A ceilometer can only be used to detect aerosol layers as a function of height. This is not much, but sufficient for PBL studies. That should be clearly mentioned.

Author's response:

In order to differentiate and define the composition of atmospheric aerosols, various wavelengths corresponding to different characteristics are necessary. Weigner et al., (2014) further explains: "Whereas the detection of aerosol layers and their vertical extent requires only simple single-wavelength backscatter lidars, the derivation of extinction coefficient profiles and a series of intensive aerosol properties requires advanced lidar concepts such as high-spectral resolution lidars (HSRL, Shipley et al., 1983) or Raman lidars (Ansmann et al., 1992)". Nonetheless, Weigner succeeded to produce satisfactory estimations of the attenuated coefficient based on signal calibrations and corrections for water vapor absorption (Weigner and Gasteiger, 2015).

Author's changes in manuscript:

" One drawback is that calibration procedures were nonexistent in all sites, and in most cases, maintenance procedures (cleaning of the ceilometer window) were not regularly carried out, with the exception of the IMS Beit Dagan ceilometer. Nevertheless, the PBL height detection is based on a pronounced change of the attenuated backscatter profile. This change is attributed to variations in the aerosol content providing indications for both clouds and atmospheric layers. Therefore, the limitation of a single wavelength within the spectral range of water vapor absorption does not affect this type of detection. In order to derive the backscatter coefficient from ceilometer measurements, signal calibrations and water vapor corrections are necessary (Weigner et al., 2014, Wiegner and Gasteiger, 2015)".

Referee's comment:

P7, L185: Please state again where Beit Dagan is located.

P8, L184-187: It should be clearly emphasized that the radiosonde provides ONE value for the PBL height, no diurnal cycle, ... nothing! Only a snapshot of the PBL height, a few minutes after launch is provided by the sonde! In contrast, models can produce the diurnal cycle, and ceilometers can measure it. But all this is not shown and discussed!

Author's response:

(1) Lines 184-185 state:" Radiosonde (RS) type….is launched twice daily at 23 UTC and 11 UTC by the IMS in the Beit Dagan site, adjacent to the ceilometer".

(2) The time differences between the models and the ceilometers were mentioned in the text as follows:

    a) P 5, lines 146-147: "IFS profiles were limited to hourly resolution, while COSMO generated profiles every 15 minutes. To compare COSMO's PBL heights, a series of trials were performed to find the correct representation of hourly values as the last 15 minutes within an hour".

    b) P 6, lines 179-181: "To compare the hourly results of the models (Sect. 3), the ceilometers' 15 seconds profiles were averaged to half-hour ones, whereas the second half-hour profile within each hour was chosen".

Nonetheless, the relevant sections were rephrased to create a clearer explanation.

Author's changes in manuscript:

(1) Sect 4.2 (Radiosonde) was rephrased with additional information:

"The IMS obtains systematic radiosonde atmospheric observations twice daily, at 23 UTC and 11 UTC, adjacent to a ceilometer. Launching is performed in Beit Dagan (32.0 ° long, 34.8 ° lat, 33 m a.s.l), situated 7.5 km east from the shoreline, 11 km southeast to Tel Aviv, 45 km northwest to Jerusalem (Fig.1 and Table 1). The radiosonde, type Vaisala RS41-SG, produces profiles of RH, temperature, pressure, wind speed and wind direction as it ascends. Measurements are retrieved every 10 seconds, corresponding to about every 45 m, reaching 2 km in about 8 minutes. The horizontal displacement of the radiosonde depends on the intensity of the ambient wind speed. The average wind speed along the 11 UTC summer profiles is about 5 m/s (Uzan et al., 2012). Therefore, the horizontal displacement of the radiosonde from its launch position is fairly low and is estimated at about 2.5 km. Moreover, the radiosonde position resolution is defined as 0.01°. As aforementioned, the PBL height in Beit Dagan for midday summer is estimated below 1 km (Dayan and Rodinzki, 1999, Uzan et al., 2016, Yuval et al., 2019). Hence, within an ascending height of 1 km, the change in the radiosonde's horizontal position is under 0.01° which is an order of magnitude from the models' grid resolution. Thus, we assert the radiosonde profiles represent the Beit Dagan site and the displacement error of the ascending radiosonde can be neglected".

(2) A text was added to Sect. 6.1 (Comparison to in-situ radiosonde profiles):

"Statistical analysis of the Beit Dagan PBL heights mean error (ME), root mean square error (RMSE), and correlation (R) is presented in Fig. 2 and Table 3 for 11 UTC. The analysis was based on the comparison between radiosonde measurements at 11 UTC, to Beit Dagan ceilometer average profiles between 10:30-11:00 UTC, IFS estimations for 11 UTC and COSMO results for 10:45 UTC".

**5.Methods**

Referee's comment:
This chapter is much too long. Textbook knowledge is presented in unnecessary detail. For each method, please provide the equation, the explanation of the equation, the link to PBL height, and a proper reference. More is not needed. A short and compact section on methods is desirable.

Author's response:
Comment accepted.

Author's changes in manuscript:
The method section was edited in a concise manner.

Referee's comments:
P9, L247: This is confusing: A ceilometer is made to detect the base of the water cloud, but not to detect the cloud top height. In most cases of low level (liquid-water) clouds, there is no chance to detect the cloud top! This needs to be clearly stated. The maximum signal you measure cannot be interpreted as a cloud top. This is a very erroneous statement! The maximum backscatter signal is somewhere between the cloud base and cloud top. The maximum signal is at that height where the attenuation effect becomes so strong that the signal immediately drops to the sky background level. This needs to be clearly stated. The height of the maximum signal maybe 100, 300, or 1000 m below the cloud top. Nobody knows!

P10 L268: ...Therefore, also the following statement is wrong: Our algorithm denotes the PBL height as the top of the shallow cloud. As just mentioned, you are unable to see the cloud top with ceilometer, only exceptional, in cases with optically rather thin clouds. Please improve your statements. The discussion is unacceptable in the present form.

Author's response:
Thank you for this important remark.

In this research, we employed the wavelet covariance transform (WCT) method on the ceilometers' backscatter profiles. The principle of this method is to calculate the derivatives between measuring points along the length of the backscatter profile. The highest derivative implies a profound difference in the atmospheric aerosol content. On clear days, this difference occurs as the transmitted light exits the well-mixed layer and enters the stable layer above. In the presence of clouds, the highest values are retrieved at cloud base height which is considered as the mixed layer height. The cloud top denotes the bottom height of the free atmosphere (Fig.3 from Stull, 1988).

[Figure]

Fig.3 The troposphere can be divided into two parts: a boundary layer (shaded) near the surface and the free atmosphere above it.

(Source: Stull, 1988)

Therefore, in order to generate a consistent definition of the PBL height by the WCT method, our algorithm seeks the height of the transition zone in the presence of clouds as well. This height is defined here as the highest measuring point of a cloud above the cloud base height. Even though the summer clouds are relatively shallow (~ 500 m thickness based on observations, see example in Fig.4 and Fig.5), there is no guarantee the algorithm detects the actual cloud top. Therefore, to prevent misinterpretations, the phrase "cloud top" was omitted and clarified as the highest measurement point of a cloud above a cloud base height.

[Figure]

Fig.4 IMS photograph of the sky over Beit Dagan site on August 2, 2019, at 8 UTC presenting typical shallow cumulus clouds.

[Figure]

Fig.5  Terra-MODIS 250 m resolution picture over Israel on August 2, 2019, at 8 UTC.
Beit Dagan site is indicated by a red dot.  Adapted from @NOAA- EARTHDATA.

Author's changes in manuscript:
"When clouds are present (mainly summer shallow cumulous), the algorithm defines the highest measurement point of a cloud (above the cloud base height) as the height where the signal counts decrease to the amount retrieved by background values. This signifies the ceilometer's identification of the entrainment zone (Stull, 1988)".

**6.Results**

Referee's comment:
P10, L286, and Figure 3: This is the worst case you can select in a comparison paper. There is the PBL development, there is the sea breeze effect, and there is cloud evolution! As a consequence, the PBL depth is more or less undefined at these complex atmospheric conditions… This case study is rather confusing and not helpful. Unambiguous, cloud-free conditions would be desirable to check the different approaches of PBL height retrieval.
Author's response:
We analyzed a total of 33 cases and received good results for the majority of the data (cases of either cloud-free or sporadic shallow cumulus clouds). The largest gaps between the models' estimations and the radiosonde measurements were found on August 17, 2016, presenting an uncommon multi-layer summer cloud. As the referee correctly discerned, this complex meteorology explains the large gaps between the models and the instruments. We agree with the referee for the necessity to present a case reflecting the ability of the method. Therefore, we generated a new figure demonstrating a typical event to explain the method rather than the extraordinary results of Aug 17, 2016.
Author's changes in manuscript: Figures 3 and 5 were replaced by a typical event on August 15, 2015 (Fig. 6).

[Figure]

Fig.6 Meteorological measurements from Beit Dagan site on August 15, 2015: Virtual potential temperature profiles at 11 UTC generated from radiosonde measurements, IFS and COSMO models (a), ceilometer signal counts plot including indications of the PBL heights at 11 UTC from the models (IFS$_R$, IFS$_P$, COSMO$_R$, COSMO$_P$), radiosonde (RS$_R$, RS$_P$) and ceilometer (b). The bottom panel presents radiosonde profiles of temperature, RH, wind speed and wind direction at 11 UTC (c).

**Referee's comment:**

P10, L286, and Figure 3: Fortunately, the radiosonde temperature profile indicates the PBL height at about 800m because for this height range (from 50 – 800m) the layer is well mixed indicated by the almost height-independent virt. pot. temperature. Then the pot. Temperature strongly increases with height and prohibits vertical mixing higher up. However, in Fig.3, the PBL heights obtained by the authors (from radiosonde, ceilometer, COSMO and IFS model) are between 1000 and 2200m? This is confusing! The PBL height is clearly not at 1000m, 1400m, 1700m, or even 2000m. So, the ceilometer result of 1700m is totally wrong to my opinion. The reason is obviously that the range-corrected signal (and the wavelet analysis) cannot be used at these cloudy conditions to detect the true PBL height. What you see is some arbitrary height where the range-corrected signal takes its maximum...

**Author's response:**

The referee indicated the PBL height as the highest point aloft before the virt. pot. temperature increases. Following Stull (Fig.8 from Stull, 1988, Chapter 5, paragraph 5.5), and the parcel method (Holzworth 1964, Seidel et al., 2010) we indicated the PBL height as the height where the virtual potential temperature reaches the value that of the surface level. By this method, the PBL height is indicated as the height where the passage from the unstable layer to the stable layer above occurs. The unstable layer is defined by the mixed layer and the entrainment zone above. This definition corresponds with the height point at which an abrupt change is measured by the ceilometers, at the transition zone between the well-mixed layer and the free atmosphere above.

[Figure]

Fig. 7 Static stability as a function of the $\overline{\theta_v}$ profile. Dotted lines denote parcel movement.

We refereed to figure (k) and (I) corresponding to daytime summer static stability of the eastern Mediterranean.

Author's changes in manuscript:
No changes were made in the manuscript.

Referee's comment:
P10, L286, and Figure 3: If the radiosonde observations of temperature, relative humidity, wind speed, and wind direction would be shown, we would have the chance to see what is going on here. But all this is not presented. Height resolved trajectory analysis would be helpful as well in the discussion of the complex meteorological conditions. Please provide at least the wind and RH profiles of the radiosonde in the figures. The reader may want to know more about the meteorological situation.

Author's response:
Comment accepted. Profiles of temperature, RH wind speed and wind direction from the adjacent radiosonde launch site are given in Fig. 8 below.

[Figure]

Fig.8 Ceilometer signal counts plot on August 17, 2016, including indications of the PBL heights at 11 UTC from the models (IFS$_R$, IFS$_P$, COSMO$_R$, COSMO$_P$), radiosonde (RS$_R$, RS$_P$) and ceilometer (a). The bottom panel presents the radiosonde profiles retrieved at 11 UTC on the same day (b).

Author's changes in manuscript:
Additional plots presenting radiosonde profiles of wind speed, wind direction, relative humidity, and temperature were added to the typical case on Aug 15, 2015.

Referee's comment:

P11, L308: Again, Figure 5 shows a rather difficult case (PBL evolution plus sea breeze effect). There is obviously a marine boundary layer (with the top at 600m, clearly seen by the radiosonde) and, on top, the upper part of continental PBL up to about 1500m (also visible in the radiosonde profile). But, per definition, the lower PBL counts (the lowest well-mixed layer above the surface is the boundary layer, as defined by Stull 1988). And that is the marine boundary layer, indicated by the potential temperature profile and the ceilometer data. But the PBL height obtained from the ceilometer profile analysis is again around 1700 m. This is an error of more than 100%! Please show RH and wind profiles (direction and speed) so that more information about the complex PBL development at sea breeze conditions is available. Again, the selected case and the discussion are rather confusing. The results are at all not convincing, and not understandable. What is then the message of the study? Obviously, the IFS model does not simulate the impact of the sea breeze impact correctly or even ignores sea breeze effects so that the continental pot. temperature profile is obtained with this model. The IFS PBL heights seem to be in contradiction with the IFS pot. temp. profile. The COSMO pot. temp. profile is in good agreement with the radiosonde profile and shows the PBL height at 600 m. Very stable conditions higher up are simulated with COSMO so that not vertical mixing is possible above 600 m height. Surprisingly, the COSMO PBL height is from 1700 to 2100 m. This is totally confusing! This seems to be simply a mistake! Please clarify!

Author's response:

We deeply apologize for this clerical error. The referee is correct. The figure contains a grave mistake. Unfortunately, the data of PBL heights of Aug 17, 2016, were mistakenly presented for Aug 10, 2015, as well. A correct figure including meteorological profiles from the adjacent radiosonde are given in Fig. 9 and Fig 10 below:

[Figure]

Fig.9 Before (left panel) and after (right panel) the correction of the figure describing Aug 10, 2015.

[Figure]

Fig.10 Ceilometer signal counts plot on August 10, 2015, including indications of the PBL heights at 11 UTC from the models (IFS$_R$, IFS$_P$, COSMO$_R$, COSMO$_P$), radiosonde (RS$_R$, RS$_P$) and ceilometer (a). The bottom panel presents the radiosonde profiles retrieved at 11 UTC on the same day (b).

Author's changes in manuscript:
The corrected figure including the meteorological conditions for each study case is given in the point to point response, but not in the manuscript. Following the referee's suggestion, they were replaced by a representative case of the method on August 15, 2015.

Referee's comment:
P12: Is section 6.3 needed? It is a very specific regression approach, just applicable to Israel.
Author's response:
Sect. 6.3 suggests a new approach to correct COSMO PBL height estimations by ceilometers. Actually, that is the goal of the research. The method proved as an applicable tool to validate and even correct the model's estimations. In regions with scarce profiling, there are no other alternatives to validate the model's results. Considering the simplicity of the method, it can be easily adapted in similar topographical areas by adjusting the correction factors (Eq. 6).
Author's changes in manuscript:
The paragraph was rephrased to emphasize the advantage and importance of the suggested method.

Referee's comment:
P12-13 The conclusions must be rewritten after clarifying all the contradictions.
Author's response:
Comment accepted.
Author's changes in manuscript:
The Conclusions paragraph was rephrased accordingly.

We wish to thank referee #2 for the constructive comments. Although the referee suggested the article should not be published in its current form, the referee took the time and effort to present a list of comments. The manuscript was intensely reexamined and has gone over a major revision. We thank both the referee and the editor for the opportunity to reply and improve the paper. Our point to point response is given by order of appearance.

Referee's comment:

Although there is no doubt that determining the mixing height is important, the scientific community has done extensive research and progress so far on the daily boundary layer. However, there are still significant problems as well as gaps in the night boundary layer (stable conditions) and in the transitional periods. These periods cannot be omitted in a study when referring to the importance of mixing height in the formulation of concentrations and even more when one of the main initiatives is to designate ceilometers as a correction for NWP. The statement on line 218 is not appropriate for the exclusion of the nighttime period. Also, the methodology applied by the authors for the reliability of PBL estimation from ceilometers data raises many reservations. I personally could not find the value of this research effort. Concluding, I believe that the whole processing of the subject is rather limited, covers a very short period and is of local interest only. Therefore, I do not agree that this study is published in the ACP Journal.

Author's response:

The analysis of the PBL heights from NWP models over diverse terrain and the ability of the regression tool (Eq. 6) to produce adequate corrections presents an interesting study case and a preview of the great potential of ceilometers as a validation and correction tool to discern PBL heights derived by NWP models.

The distribution of ceilometers in Israel is at its first stages. Data for the summer season from as many ceilometers as possible over a heterogeneous area concluded with a time span of two months (August between 2015-2016). Initially, we analyzed the diurnal evolution of the summer PBL height. The models' PBL scheme is based on the bulk Richardson method. Thus, the models estimated the nocturnal surface boundary layer (SBL) as the first model level for all dates examined. Moreover, the ceilometers' detection of the SBL height in ground-level sites was found mainly within the first range gates. At these range gates, a perturbation exists due to the overlap of the emitted laser beam and the receiver's field of view. This constrained our ability to determine the low SBL height of the summer season. Consequently, the research focused on convective daytime hours (09-14 UTC).

Author's changes in manuscript:

The manuscript has gone over a major revision to address the referee's reservations.

Referee's comment:

I wonder if we could perform a similar exercise for an area with restricted characteristics, thus no general applicability. This is the case here, where local flows are developed but there are not taken into consideration. In particular, both sea breeze and/or anabatic winds are expected to develop in this area during the summer period. (1) For this reason, I am not sure what the ceilometer is measuring. (2) For example, at the station of DB just 7 km away from the shore, the PBL depth is measured at 1km. To my knowledge, this is an unrealistic value (too high) under the presence of sea breeze (or IBL). Thus, I wonder if this instrument finally shows the off-shore current of the sea breeze flow.

Author's response:

(1) Local flows are taken into consideration by the models and the ceilometers. While the models simulate the physical parameters generating them (for example, see Fig.1), the ceilometers measure the results of these flows expressed as backscatter signals

[Figure]

Fig 1. COSMO model maps of RH, wind speed and wind direction over Israel in August 2019 between the hours 07 Z-15 Z (LST=Z-2). Source: Dr. Pavel Khain, Israeli Meteorological Service.

Uzan et al, (2012) studied the ability of the wavelet covariance transform (WCT) method to delineate the evolution of the summer mixed layer height (not the PBL height) based on ceilometers' profile. The results are presented in Fig.2.

[Figure]

Fig.2 Hourly averaged mixed layer height (MLH) for the east Mediterranean summer season (July – August) 2014. Dashed lines indicated ceilometer measurements in Beit Dagan (BD) and solid lines indicate the measurements in Tel Aviv (TLV) for July (blue and light blue) and August (red and pink). TLV and BD plots are based on 19, 31 days in July and 24, 30 days in August, respectively. Indications of the time of sunrise, sunset and the SBF entrance time are given.

This figure demonstrates the diurnal summer mixed layer height between July-August in 2014. The analysis was carried out by two ceilometer sites: Tel Aviv (50 m from the shoreline, 5 m a.s.l) and Beit Dagan (7.5 km from shoreline, 11 km southeast to Tel Aviv,33 m a.s.l). The ceilometers' measurements succeeded to capture the inflation of the mixed layer height after sunrise followed by subsidence as the sea breeze front prevails. A height difference of 200 m was measured between the two sites at midday. This difference is attributed to the greater distance of Beit Dagan from the shoreline (7.5 km) enabling the convective thermals to develop and inflate the mixed layer height. Tel Aviv site, on the other hand, is practically on the shoreline, therefore the sea breeze promptly surmounts the convective thermals preventing from the mixed layer to inflate. The apparent height difference of the mixed layer height in Beit Dagan in July (dashed blue line) compared to August (dashed pink line) was ascribed to the fact that August was cloudier than July after sunset.

(2) The assertion Beit Dagan PBL height reaches 1000 m a.s.l is based on the following studies:
   a) Felix Y, 1994 stated: " The daily inversions over the coast of Israel have been studied by Shaia and Jaffe (1976). Their analysis was based on 10 years of observations of temperature profiles measured by the afternoon radiosonde (1200 UTC) at Bet Dagan (7 km inland from the central coast of Israel). According to their statistics, in 81% of the summer days (June-August), inversions occurred. The base height of most of these inversions was between 500 and 1000 m and their mean thickness was about 400 m.
   b) Yuval et al., (2019) evaluated monthly median values of the PBL height (denoted as CBL for midday PBL) as evaluated by the midday Beit Dagan radiosonde profiles, based on the W&W method. Fig. 3 presents the median value of the PBL heights (green line) in August reach 900 m a.s.l.

[Figure]

Fig 3. Monthly median values of the PBL height (continuous lines) and the lapse rate Γ (dashed lines) under CBL conditions. The PBL heights were estimated using the W&W method. The lapse rate was calculated between the temperature at the 12th radiosonde reporting level data (24 s from launch, on average at 108 m) and the second level (two seconds, on average 6.2 m).

c) The sea breeze effect is evident by the ceilometers' attenuated backscatter profiles as shown by the figure below depicted from Uzan et al., 2016. Note the Beit Dagan PBL height reaches 1000 m a.g.l.

[Figure]

Fig. 4. Mixed layer height (MLH, solid black line) on the 20.08.2014 for Beit Dagan (BD) and Tel Aviv (TLV). The MLH line is laid upon half hourly averaged attenuated backscatter profiles (units 10-6 m-1 sr-1). The BD plot is shifted 2 hours to coincide with UTC time. The plot includes indications of the sunrise (yellow bar) and the sea breeze front (SBF) entrance time (pink bar), MLH evaluation by radiosonde profiles at 0 and 12 UTC (red plus) and calculation of the MLH subsidence rate due to SBF entrance.

Author's changes in manuscript:

The text was rephrased in Sect. 2 (Research area): " On the synoptic scale, the summer is defined by a persistent Persian Trough (either deep, shallow or medium) followed by a

Subtropical High aloft (Felix Y., 1994, Dayan et al., 2002, Alpert et al., 2004). Combined with the sea breeze effect, the average PBL height is found to be quite low. For example, the PBL height in Beit Dagan (33 m a.s.l and 7.5 km east from the shoreline) reaches ~900 m a.s.l after sunrise, and before the entrance of the sea breeze front (Felix Y.,1994, Dayan and Rodinzki, 1999, Uzan et al., 2016, Yuval et al., 2019).".

Referee's comment:

(1) The PBL depth is a non-specific parameter, the definition and estimation of which is not straightforward. The simulated PBL depths are mainly determined, based on the definition that each PBL scheme applies (in this study, no information is provided regarding the PBL parameterization schemes considered by the two models). (2) This also applies between measurements from different instruments (ceilometer and radiosonde) as they do not have the same operating principles. The ceilometer measurements mainly present the mixed PBL that does not always coincide with the simulated PBL depth. (3) On the other hand, has it been taken into account that the radiosonde moves with the flow? As it ascends the measurements do not correspond to the vertical position above the launch point. This is another reason for a possible discrepancy between the radiosonde and the ceilometer measurements.

Author's response:

(1) Comment accepted: Descriptions of the models' PBL parameterization schemes were added to Sect. 3 (IFS and COSMO Models).

(2) We addressed the same methods on the models and radiosonde measurements (the bulk Richardson method and the parcel method). These methods cannot be imposed on the ceilometers' attenuated backscatter profiles, therefore we generated a specific method based on the WCT method, and compared the results to the heights generated by the radiosonde. Results for 33 days (presented in the manuscript in Fig.2 and Table 1) revealed a high correlation between the two instruments (0.93) and low RMSE (97 m).

(3) Radiosonde profiles are retrieved every 10 seconds, corresponding to about every 45 m, reaching 2 km in about 8 minutes. The horizontal displacement of the radiosonde depends on the intensity of the ambient wind speed. In this study, we analyzed the PBL height of midday summer profiles (11 UTC). The average wind speed along these profiles is about 5 m/s (Uzan et al., 2012). Therefore, the horizontal displacement of the radiosonde from its launch position is fairly low and is estimated at about 2.5 km. Moreover, the radiosonde position resolution is defined as 0.01°. The PBL height in Beit Dagan for midday summer is estimated below 1 km (Felix Y.,1994, Dayan and Rodinzki, 1999, Uzan et al., 2016, Yuval et al., 2019). Hence, within an ascending height of 1 km, the change in the radiosonde position will be below 0.01°. This spatial error is in the order of magnitude of the models' grid resolution. Thus, we assert the radiosonde profiles represent the Beit Dagan site and the displacement error of the ascending radiosonde can be neglected

Author's changes in manuscript:

(1) Concise descriptions of the models' schemes were added to Sect. 3 (IFS and COSMO models):

a) IFS PBL parameterization scheme: "The turbulent diffusion scheme represents the vertical exchange of heat, momentum, and moisture through sub-grid scale

turbulence. In the surface layer, the turbulence fluxes are computed using a first-order K-diffusion closure based on the Monin-Obukhov (MO) similarity theory. Above the surface layer, a K-diffusion turbulence closure is used everywhere, except for unstable boundary layers where an Eddy-Diffusivity Mass-Flux (EDMF) framework is applied, to represent the non-local boundary layer eddy fluxes (Koehler et al. 2011)".

b) The COSMO turbulent scheme: "The turbulence scheme, based on Mellor and Yamada (1982) at Level 2.5, uses a reduced second-order closure with a prognostic equation for the turbulent kinetic energy. The transport and local time tendency terms in all the other second-order momentum equations are neglected and the vertical turbulent fluxes are derived diagnostically (Cerenzia I., 2017)".

(2) Sect. 6.1 (Comparison to in-situ radiosonde profiles) was rephrased: " In order to evaluate the daytime PBL heights produced by the models and the ceilometers, the results were compared to the radiosonde's evaluations. Consequently, the investigation was held in Beit Dagan at the time of the midday launch (11 UTC). For this comparison, the ceilometer's 15 s profiles were averaged as half-hour profiles between 10:30-11:00 UTC. COSMO's results referred to the profiles of 10:45 UTC, and IFS estimations were given at 11 UTC. The analysis was carried out for 33 summer days, 13 days from August 2015, and 20 days from Aug 2016. The PBL heights were produced by the same methods: the parcel method (denoted by subscript P) and the bulk Richardson method (denoted by subscript R). These methods require meteorological parameters such as temperature and pressure profiles generated by the models and the radiosonde. Ceilometers, on the other hand, produce only backscatter signals. Therefore, they were analyzed by the WCT method. The results were statistically analyzed by mean error (ME), root mean square error (RMSE), and correlation (R) presented in Fig. 2 and Table 3. Good agreement was found between the ceilometer and the radiosonde (ME = 12 m, RMSE = 97 m, and R = 0.93), although they produced the PBL heights by different methods. ".

(3) Sect. 4.2 (Radiosonde) was rephrased: "The IMS obtains systematic radiosonde atmospheric observations twice daily, at 23 UTC and 11 UTC, adjacent to a ceilometer. Launching is performed in Beit Dagan (32.0 ° long, 34.8 ° lat, 33 m a.s.l), situated 7.5 km east from the shoreline, 11 km southeast to Tel Aviv, 45 km northwest to Jerusalem (Fig.1 and Table 1). The radiosonde, type Vaisala RS41-SG, produces profiles of RH, temperature, pressure, wind speed and wind direction as it ascends. Measurements are retrieved every 10 seconds, corresponding to about every 45 m, reaching 2 km in about 8 minutes. The horizontal displacement of the radiosonde depends on the intensity of the ambient wind speed. The average wind speed along the 11 UTC summer profiles is about 5 m/s (Uzan et al., 2012). Therefore, the horizontal displacement of the radiosonde from its launch position is fairly low and is estimated at about 2.5 km. Moreover, the radiosonde position resolution is defined as 0.01°. As aforementioned, the PBL height in Beit Dagan for midday summer is estimated below 1 km (Dayan and Rodinzki, 1999, Uzan et al., 2016, Yuval et al., 2019).  Hence, within an ascending height of 1 km, the change in the radiosonde's horizontal position is under 0.01° which is an order of magnitude from the models' grid resolution. Thus, we assert the radiosonde profiles represent the Beit Dagan site and the displacement error of the ascending radiosonde can be neglected".

Referee's comment:
Therefore, the same criteria should be used for the estimation of both measured and simulated PBL depth. In particular, the same criteria should be applied to the profiles of certain atmospheric parameters, such as temperature, wind and mixing ratio profiles that depict the atmospheric boundary structure. These criteria should not necessarily be the same for all atmospheric conditions. For example, the gradient of potential temperature profile is inadequate to provide the turbulent ABL depth. Therefore, for the comparison with ceilometer, it would be more appropriate to consider the eddy-viscosity simulated profiles or even better the aerosol layering from chemistry transport model simulation.

In particular, under convective conditions, the mixing height determined by ceilometer is strongly related to the aerosol stratification.

Author's response:
We employed the parcel method to evaluate the transition zone between the mixed layer and the free atmosphere, as presented in Fig. 5 from Stull (1988). In this method, the virtual potential temperature at ground level is crucial while the models' lowest grid point is above the surface layer (IFS begins at 10 m a.g.l. and COSMO at 20 m a.g.l.). Therefore, the virtual potential temperature at ground level height (2 m a.g.l) was evaluated by the temperature and dew point temperature (or RH) derived by the models based on the similarity theory.

As explained in the previous comment, we addressed the same methods on the models and radiosonde measurements (the bulk Richardson method and the parcel method). These methods cannot be imposed on the ceilometers' attenuated backscatter profiles, therefore we generated a specific method for the ceilometers' PBL heights evaluations based on the WCT method. To ensure the WCT method addressees the same PBL heights generated by the other methods, we compared the ceilometer's evaluations to the radiosonde's heights. Results for 33 days (presented in the manuscript in Fig.2 and Table 1) revealed a high correlation between the two instruments (0.93) and low RMSE (97 m).

[Figure]

(source: Stull, 1988)

Author's changes in manuscript:
No changes were made in the manuscript.

Referee's comment:

How much value does the global model have in such a small analysis to take part in the comparison, especially in a strongly heterogeneous area?

Author's response:

The main goal of the study was to utilize ceilometers as a correction tool for NWP models. Therefore, two types of models were tested, global and regional. The limited ability of the global models to correctly simulate complex terrain was taken into consideration. Therefore, we did not anticipate the significantly large overestimations of IFS over flat grid points under fairly "simple" meteorological conditions characterizing the summer in the East Mediterranean. This disclosed the advantages of the regional model as well as the limitations of the global model in regard to PBL height estimations.

Author's changes in manuscript:

No changes were made in the manuscript.

Referee's comment:

(1) Also, there are several arbitrary statements on the text, without any justification (no measurements of wind speed and direction are provided) or any reference.

(2) For example:

Line 105: "As a result, the average PBL height is comparatively low (~1000 m a.g.l)".

Line 116: "Through the day, the sea breeze circulation steers clockwise and the wind speed is enhanced by the west-north-west synoptic winds".

Line 119: "Due to the large distance (~30-50 km inland), the SBF reaches the eastern elevate complex terrain only in the afternoon (~11-12 UTC).

Line 170:" However, the PBL detection algorithm utilized here (see Sect. 5.3) is based on a significant signal slope, therefore can be determined from uncalibrated ceilometers".

Author's response:

(1) Comment accepted. Moreover, following the comments from referee # 1, the study cases of August 10, 2015, and August 17, 2016, were removed and replaced with the description of a typical case on August 15, 2015, provided with radiosonde profiles of wind, temperature, and relative humidity.

(2) Comment accepted. The whole paragraph was rephrased accordingly and the references were inserted within the text rather than the list given in the previous form.

Author's changes in manuscript:

(1) The study case of August 15, 2015, was provided with radiosonde profiles of temperature, relative humidity, wind speed, and wind direction.

(2) Sect. 2 (Research area) was rephrased in the following manner: "Previous research describes the formation and evolution of the Israeli summer PBL height as a function of the synoptic and mesoscale conditions, as well as the distance from the shoreline, and the topography. Overall, the diurnal PBL height in the summer season may be portrayed in the following manner: After sunrise (~4-5 LST, where LST=UTC+2) clouds initially formed over the Mediterranean Sea are advected eastward to the shoreline. As the ground warms up, the nocturnal surface boundary layer (SBL) dissipates and buoyancy induced convective updrafts to instigate the formation of the sea breeze circulation (Stull, 1988). The entrance of the sea breeze front (SBF) is estimated between 7-9 LST (Felix Y., 1993,

Alpert and Rabinovich-Hadar, 2003, Uzan and Alpert, 2012), depending on the time of sunrise and the different synoptic modes (weak, medium and deep) of the prevailing system – the Persian trough (Alpert et al., 2004). Cool and humid marine air hinder the convective updrafts, thus clouds dissolve and the height of the shoreline convective boundary layer (CBL) lowers by ~250 m (Felix Y., 1993, Felix Y., 1994, Levi et al., 2011, Uzan and Alpert, 2012). Further inland, the convective thermals continue to inflate the CBL (Hashmonay et al., 1991, Felix, 1993, Lieman, R. and Alpert, 1993) while the sea breeze circulation steers clockwise and wind speed is enhanced by the west-north-west synoptic winds (Neumann, 1952, Neumann, 1977, Uzan and Alpert, 2012). By noontime (~11-13 LST), the sea breeze and the synoptic wind merge and produce maximum wind speeds which suppress the CBL (Uzan and Alpert, 2012). In the afternoon (~13-14 LST), the SBF reaches ~30-50 km inland to the eastern elevated complex terrain (Hashmonay et al., 1991, Lieman, R. and Alpert, 1993). At sunset (~18-19 LST), as the insolation diminishes, the potential energy of the convective updrafts weakens and the CBL height drops (Dayan and Rodnizki, 1999). After sunset, the CBL finally collapses and a residual layer (RL) is formed above the SBL (Stull, 1988). As the ground cools down, the high humidity and low RL create low condensation levels which produce shallow evening clouds".

Referee's comment:
Line 203-Does the bulk Richardson refers to a certain height or layer?
Author's response:
The bulk Richardson method refers to a certain layer in the models and to a specific height in the radiosonde profiles.
Author's changes in manuscript:
Sect 5.1 (The bulk Richardson number method) was rephrased as follows:" The IFS model defines the PBL height as the lowest height level at which the Rb (Eq. 1) reaches a critical threshold of 0.25 (ECMWF-IFS documentation – Cy43r3, Part IV: Physical Processes, July 2017). The PBL height is distinguished by scanning the bulk Richardson results from the surface upwards. When the PBL height is found between two levels of the model, it is determined by linear interpolation. Radiosonde's profiles were analyzed in the same manner by a $R_b$ threshold of 0.25 to detect a specific height rather than a certain layer.
COSMO estimates the $R_b$ based on the dynamic conditions of the first four levels (10, 34.2, 67.9, 112.3 m a.g.l.) signified by a threshold of 0.33 for stable conditions and 0.22 for unstable ones. If no level is found, then a missing value is assigned for the PBL height".

Referee's comment:
Lines 265-end of this paragraph. I am confused.
Author's response:
The end of the paragraph states: "However, as previously mentioned, our algorithm denotes the PBL height as the top of the shallow cloud (Stull, 1988)". We assume the confusion regards the term "cloud top". We agree with the referee this definition is confusing within the context

it was used and apologize for the misunderstanding we have caused. The term was changed given the explanation as follows:

In this research, we employed the wavelet covariance transform (WCT) method on the ceilometers' backscatter profiles. The principle of this method is to calculate the derivatives between measuring points along with the backscatter profile. The highest derivative implies a profound difference in the atmospheric aerosol content. On clear days, this difference occurs as the transmitted light exits the well-mixed layer and enters the stable layer above. In the presence of clouds, the highest values are retrieved at cloud base height which is considered as the mixed layer height. The cloud top denotes the bottom height of the free atmosphere (see Fig.6 from Stull, 1988).

[Figure]

Fig.6    The troposphere can be divided into two parts: a boundary layer (shaded) near the surface and the free atmosphere above it.

(Source: Stull, 1988)

Therefore, in order to generate a consistent definition of the PBL height by the WCT method, our algorithm seeks the height of the transition zone in the presence of clouds as well. This height is defined here as the highest measuring point of a cloud above the cloud base height. Even though the summer clouds are relatively shallow (~ 500 m thickness based on observations, see example in Fig.5 and Fig.6 below), there is no guarantee the algorithm detects the actual cloud top. Therefore, to prevent misinterpretations, the phrase "cloud top" was omitted and clarified as the highest measurement point of a cloud above a cloud base height.

[Figure]

Fig 7.  IMS photograph of the sky over Beit Dagan site on August 2, 2019, at 8 UTC presenting typical shallow cumulus clouds.

[Figure]

Fig 8.  Terra-MODIS 250 m resolution picture over Israel on August 2, 2019, at 8 UTC.
Beit Dagan site is indicated by a red dot.  Adapted from @NOAA- EARTHDATA.

Author's changes in manuscript:

" When clouds are present (mainly summer shallow cumulous), the algorithm defines the
highest measurement point of a cloud (above the cloud base height) as the height where the
signal counts decrease to the amount retrieved by background values. This signifies the
ceilometer's identification of the entrainment zone (Stull, 1988)".

* * *
**Author's comment:**

Following the referees' comments, we repeatedly examined our datasets and the methods. We found that the equations of the virtual temperature (Eq.1) and the virtual potential temperature (Eq.2) employed values of Rd/Cp = 287/1004 (~ = 0.28586), and surface pressure of $P_0$ =1000 mb for the radiosonde data, while in the models, these factors were defined as Rd/Cp = 0.263, $P_0$ = 1013.15 mb. Therefore, we decided to transform all factors to the same values of Rd/Cp = 287/1004, $P_0$ = 1000 mb. Essentially, this altered the models' results based on the parcel method (see changes in Tables 3-5 and Fig. 2 below). It did not change the results of the correction equation or the conclusions of the research.

$$(1)\ T_v = \frac{T}{1 - \frac{e}{P}(1-\varepsilon)} \qquad (2)\ \theta_v = T_v \left(\frac{P_0}{P}\right)^{\frac{Rd}{Cp}}$$

Before                                        After

[Figure]

Fig 2. PBL heights over Beit Dagan site on 33 summer days (13 days on August 2015 and 20 days on August 2016), generated by the bulk Richardson method for IFS model (IFS$_R$, blue solid circles), COSMO model (COSMO$_R$, pink solid circles), and Beit Dagan radiosonde profiles (RS$_R$, black line). PBL heights generated by the parcel method for the IFS model (IFS$_P$, open blue circles), COSMO model (COSMO$_P$, open pink circles), and Beit Dagan radiosonde profiles (RS$_P$, same black line as RS$_R$, the results are identical). PBL heights derived from the Beit Dagan ceilometer produced by the WCT method (green circles). Results for August 17, 2016, are indicated by a circle.

Table 3. Statistical analysis of the Beit Dagan PBL heights on 33 summer days (13 days in August 2015 and 20 days in August 2016) from IFS and COSMO models by the bulk Richardson method (IFS$_R$, COSMO$_R$), the parcel method (IFS$_P$, COSMO$_P$) and the WCT method for the adjacent ceilometer. The PBL heights were compared to those derived from Beit Dagan radiosonde by either the parcel or bulk Richardson methods (see Fig 2).

| PBL detection | IFS$_R$ | IFS$_P$ | COSMO$_R$ | COSMO$_P$ | Ceilometer |
|---|---|---|---|---|---|
| Mean Error (m) | 274 | 249 (271) | -3 | -17 (-106) | 12 |
| RMSE (m) | 432 | 409 (411) | 152 | 179 (176) | 97 |
| R | 0.18 | 0.18 (0.21) | 0.83 | 0.73 (0.83) | 0.93 |
| Mean PBL (m a.s.l) | 1250 | 1225 (1247) | 973 | 959 (869) | 989 |
| Std PBL (m) | 274 | 256 (245) | 273 | 229 (222) | 259 |

*New results are given in brackets.

Table 4. Root mean square errors of PBL heights from five sites on 13 summer days (Fig. 4), derived by IFS and COSMO models by the bulk Richardson method (IFS$_R$, COSMO$_R$) and the parcel method (IFS$_P$, COSMO$_P$). The PBL heights were compared to the heights measured by the Beit Dagan ceilometer.

| Site | IFS$_R$ | IFS$_P$ | COSMO$_R$ | COSMO$_P$ |
|---|---|---|---|---|
| Ramat David | 173 m | 191 (180) m | 247 m | 241 (232) m |
| Tel Aviv | 276 m | 465 (498) m | 203 m | 183 (182 ) m |
| Beit Dagan | 405 m | 569 (569) m | 235 m | 234 (171) m |
| Weizmann | 214 m | 274 (339) m | 175 m | 145 (209) m |
| Jerusalem | 351 m | 368 (285) m | 251 m | 273 (179) m |

*New results are given in brackets.

Table 5. Same as in Table 3 but for mean errors.

| Site | IFS$_R$ | IFS$_P$ | COSMO$_R$ | COSMO$_P$ |
|---|---|---|---|---|
| Ramat David | -31 m | 30 (0) m | -26 m | 0 (-12) m |
| Tel Aviv | 234 m | 376 (422) m | 19 m | -35 (-35) m |
| Beit Dagan | 332 m | 497 (497) m | 12 m | -9 (-55) m |
| Weizmann | 114 m | 218 (280) m | 16 m | -42 (-42) m |
| Jerusalem | 298 m | 327 (243) m | -6 m | 29 ( -1) m |

*New results are given in brackets.
* * *
**A list of relevant changes made in the manuscript:**

1. Abstract- slightly changed to emphasize the goal and advantage of the research.
2. Introduction – additional text on the benefits of ceilometers as a research tool.
3. Research area- the description was elaborated and rephrased according to the relevant references.
4. IFS and COSMO models- the parameterization schemes were added.
5. Instruments- additional information regarding the radiosonde and ceilometers.
6. Methods- the section was rephrased in a concise manner.
7. Results- the study cases were changed according to the referees' comments.
8. Summary and conclusions - rephrased.

[revised manuscript text omitted]

---

## Referee Report (RR1)

Review of "Ceilometers as planetary boundary layer height detectors and a corrective tool for ECMWF and COSMO NWP models" by Uzan et al.

Uzan et al. combine observations by ceilometers at various sites and radiosoundings at one location to evaluate the planetary boundary layer height in two numerical models, namely the global IFS and the regional COSMO, in a geographically varying region (Israel). Only daytime planetary boundary layer (PBL) height in summer is considered. A good agreement between the PBL height retrieved from ceilometer measurements and soundings is shown, in line with previous studies, and indicates that the ceilometer is a suitable tool for evaluating model performance. The comparison of the PBL height between the models and observations show that the COSMO model generally performs better than the IFS. Two methods to estimate the PBL height from the models are used, neither one showing superior performance for both IFS and COSMO. The study implies that the COSMO PBL height has a bias depending on the distance from the shoreline and the topography, and a correction using these two parameters is presented. As the main finding, the manuscript claims to demonstrate that the PBL height retrieved by ceilometers can be used to improve the PBL height estimate from the COSMO model.

The region of study seems meteorologically interesting, and previous work using various observational techniques have demonstrated the influence of synoptic conditions, topography, and sea breeze on the PBL height in this region (Dayan et al, 1988, 2002; Uzan et al. 2016). Evaluating model performance regarding PBL height in this environment is a worthwhile effort. The methodology is given consideration: two techniques for calculating PBL height from the models and soundings are used, and most of the limitations of the measurements are fairly presented. The methods applied to estimate the PBL height (the bulk Richardson and the parcel method for model and radiosonde data, and the wavelet covariance transformation method for ceilometer) are standard and have been used in many previous studies. However, the manuscript falls seriously short in its analysis of the data, and the presentation and discussion of the results. The main conclusion, that the ceilometers can be used to improve the COSMO PBL height, is not sufficiently backed up by the results presented. A considerable drawback of this work is the small amount of data, which is surprising given that the main strengths of the ceilometer compared to other observational techniques available are the robust performance and low cost that allow continuous observations at multiple sites simultaneously. I strongly encourage the authors to obtain more measurements, if possible. Considering the lack of novelty in the methods and the small sample size, the authors should make considerably more effort in the careful evaluation and interpretation of their results.

**Major comments**

**1.** Motivation
The motivation, strengths and the central question of the paper could be made clearer in the introduction of the paper. As explained in Sect. 2, the region studied is interesting and different aspects are impacting the PBL height. The interesting aspects of the spatial variability of the studied region could be included in the introduction. In the light of the spatial variability, evaluating model performance on a single site would have limited value. One of the strengths of the study is the use of a network of ceilometers that can estimate the temporal development of the PBL at various locations simultaneously. This aspect deserves to be mentioned in the introduction.

Secondly, the introduction does not provide enough information to motivate the development of a post-processing tool for the modeled PBL height. The the goal to use ceilometer detected PBL height to correct for modeled PBL height could be simplified to "use A to correct B". Currently, it is demonstrated that to "use A" is possible, e.g. PBL height can, with some limitations, be retrieved

from the ceilometer measurements. However, "to correct B" is neglected in the introduction. The introduction only states the need for accurate PBL estimate, but no literature on identified shortcomings, methods found for improvement or anything else that would have been done previously to evaluate or improve PBL height estimates in NWP models is presented. Do previous studies suggest that it is more feasible to correct the end product (e.g. the PBL height) than to improve model parametrizations in order to obtain a better result from the model? Do the authors envision a use for the corrected PBL height? The authors could also consider whether their main aim should be on developing a correction, or rather a rigorous evaluation of model performance in the complex region. The latter could be helpful for understanding model shortcomings and would be a more general result than a location and time specific correction.

**2.** Amount and selection of data
One of the confusing aspects of this paper is the small number of days analyzed. The strength of the ceilometer is that data acquisition is cheap (see Sect. 1), however the small dataset is undermining this specific strength. The conclusions drawn are seriously undermined by the small sample size. For example, Sect. 6.2 seems to describe statistical results obtained from 13 data points. If possible, the authors should obtain more data. Alternatively, the study could be shifted to focus on case studies evaluating the shortcomings of the models in more detail.

Although the reasons for focusing on daytime PBL only in summer are given, further selection seems to have taken place. Why are only 13 days included from August 2015, and 20 days from August 2016 (L. 292-293) in Sect. 6.1? Why does Sect. 6.2 only include 5 ceilometer sites, when Sect. 6.3 includes 8 ceilometer sites (L. 319-321 and 345-346)? Why do Sections 6.2 and 6.3 only include data from August 2015, and not from August 2016? Are the 13 days used in Sect. 6.2 a subset of the 33 days in Sect 6.1? The authors should provide an explanation for the small number of days analyzed and why certain days and sites were selected at different stages of the study.

**3.** Significance
Related to the comment above about the amount of data, the authors should consider the statistical significance of the presented results. Specifically, wherever R-values are given (L. 298, Table 3, and elsewhere), the corresponding p-value should also be presented. Other techniques to analyze the statistical significance of the results are also welcomed, and the results should be discussed from the point of view of statistical significance.

**4.** Spatial variability (Sect. 6.2)
Section 6.2 could provide possibly the most interesting results for considering model performance in terms of PBL height in complex environments. If model under- or overestimation could be connected to certain processes (e.g. the sea breeze), the results would be more generally interesting. Mountainous coastlines are not unique to Israel, and many people inhabit such areas. This section deserves a proper evaluation, and the analysis and discussion should be extended.

Specifically, this section is hard to understand for someone not familiar with the geography of Israel. I would advice the authors to consider the presentation of their results. For example, the mean error at each site for each model and method could be presented with a symbol on a map having the color indicating the value. This would make any spatial structures in the mean, mean error (ME) or root mean square error (RMSE) more apparent. The authors could also plot the ME and/or RMSE as a function of the distance of the site to the shoreline and altitude above sea level (these are the two variables used for the correction in the next section).

From the authors description of the situation, it seems that the sea breeze has a clear influence on the PBL height. Is it to be understood, that the model does not correctly produce the sea breeze circulation, or is the model lacking in terms of the effect of the sea breeze on PBL height? It would

be interesting if the authors could evaluate the discrepancy between ceilometer and model PBL height in terms of the strength, and spatial and temporal development of the sea breeze circulation during the day. Furthermore, in Sect. 6.3 data for 9-14 UTC are used, and I suggest the authors consider including the temporal development of the PBL height in their analysis in Sect. 6.2 as well.

**5.** The rationale of the correction presented in Sect. 6.3

Before a correction is developed and presented, it should be made clear that a corrections is needed and that there is a systematic bias that can be corrected for. Table 3 (and Section 6.1) show that the mean error of COSMO$_R$ compared to radiosondes is -3 m, which does not leave much room for improvement. Also Table 5 shows that at different sites the mean error of COSMO$_R$ is within a few tens of meters at most. (However, I would be cautious to draw conclusions from statistics comprising of 13 data points, and the authors should obtain a larger sample size if possible. See comments 2 and 3). For a 1 km deep PBL, an error of 30 m is 3%. For which application is this not good enough, and how good should the model performance be? Furthermore, considering that the definition of the planetary boundary layer is slightly ambiguous, can a perfect agreement between different methods be expected? The authors should explain why they think the model performance is not good enough and requires improvement. Furthermore, the authors could consider if the correction they presented would actually be more useful for the IFS model that shows clearly worse performance than the COSMO in terms of PBL height prediction.

Sect. 6.2 should demonstrate the basis of the correction presented in Sect. 6.3. The fact that the mean error in Tel Aviv, Beit Dagan and Weizmann are so similar suggests a spatial consistency that is more clear for COSMO$_R$ than COSMO$_P$. (Table 5). Is this the reason COSMO$_R$ was used for the correction in Sect. 6.3 instead of COSMO$_P$? The fact that there seems to be some spatial structure in the mean error is promising for developing a correction. The RMSE does not seem so spatially consistent.

To justify the correction method presented in Sect. 6.3, it should be established that a bias exist in the models' PBL height estimation that depends on altitude and distance from shoreline, that could consequently be corrected for. The authors should evaluate how the discrepancy between ceilometer and model PBL height depends on the topography and distance from shoreline. Furthermore, this could be done for different hours of the day, as the correction procedure is also applied for each hour separately.

**6.** Conclusion not supported by data

Perhaps the most serious shortcoming of the manuscript is that it is not demonstrated that the model result is better after correction. The authors should include a quantitative evaluation of the improvement of the model PBL after the correction. For example, the radiosondes at Beit Dagan could be used as an independent reference for the model PBL height. Another approach would be to estimate the correction parameters using only some of the available ceilometer stations, and using the remaining stations as a references to estimate the improvement in PBL height achieved by the correction. Varying the number of stations and the locations of the stations included for fitting the the correction parameters also gives an indicator for how many ceilometers needs to be included, or how they need to be located, for achieving a significant improvement for the COSMO$_R$ PBL height. If the authors aim is to show that the ceilometer is a useful tool to improve the modeled PBL height, the strength of their paper relies on the extent and rigor that this kind of analysis is carried out.

**7.** Presenting the research area

More attention should be paid to make the reasoning understandable for readers that are not so familiar with the specific geography and climatology of the region. Firstly, the studied region and its interesting aspects could be mentioned in the introduction. The first time the the location is given is the very end of the introduction, on line 97. This should be included already in the previous

paragraph that outlines the purpose of the study, as well as in the abstract. Secondly, a topography map should be included. Global topography data is available (for example from NOAA https://doi.org/10.7289/V5C8276M) and a map can be drawn using openly available tools (such as python). Depending on the weight the authors want to give to the humidity (mentioned on lines 103-104) and the prevailing synoptic conditions (line 125), they could also include a map of mean precipitation and pressure in August to help the reader to follow their argumentation.

**Minor comments**

**8.** Lines 1-2.
The authors should reconsider the title of the manuscript. The current title is somewhat misleading because it implies that the correction for PBL height was considered for both models, when in the manuscript only the COSMO PBL height was corrected. Furthermore, the journal guidelines recommend avoiding the use of abbreviations in the title, so the authors might want to avoid the use of "NWP" in the title.

**9.** L. 23-25.
Here results are given for flat and elevated terrain. Consulting Tables 4 and 5 it seems that flat terrain refers to Tel Aviv, and elevated terrain to Jerusalem. The authors should consider mentioning the sites for which the numbers refer to to avoid ambiguity, or at least mention that the values presented are from single stations.

**10.** Abstract.
The abstract does not mention Israel or give any other indication over the geographic locations apart from "heterogeneous area" and mention of the Beit Dagan radiosonde launch site. Location should be given.

**11.** L. 33-40.
Considering that this paragraph states the broad motivation and importance of this study, some references would be appropriate.

**12.** L. 56-57.
"ceilometers obtain a wide spatial resolution per lidar" - I'm afraid I do not understand the meaning of this phrase. Perhaps the authors mean that the a wider spatial resolution is achieved by ceilometers than lidars?

**13.** L. 53-65.
This paragraph seems to suggest that ceilometers are better than lidars in every aspect. It would be fair to mention a shortcoming of the ceilometer compared to a lidar.

**14.** L. 89-91.
It is not obvious here why the summer season is more appropriate for a approach that is limited by precipitation. It is later explained that this season has low precipitation. This should also be mentioned here to help the readers not familiar with local climatology.

**15.** L. 92-97.
It would be possible to help the reader further by outlining the structure of Sect. 6, either here or at the beginning of Sect. 6.

**16.** L. 85-86.
The introduction demonstrates the strengths of ceilometers compared to other available

observational techniques to estimate PBL height, but only states that ceilometers have not been used often for evaluating model performance. However, other observational techniques have, and this should be mentioned. Specifically, have other observational tools been used for evaluating PBL height in NWP models in Israel, or other mountainous coastlines?

**17.** Introduction.
I find the extent of presenting the literature for the use of ceilometer to detect PBL height satisfactory. However, no mention of previous work using ceilometer to derive PBL hight in Isreal is presented. The authors should site at least Uzan et al. (2016) and any other studies employing the measurement technique in their region of study.

**18.** L. 106.
"IMS weather reports" - The authors should provide a more specific reference, if possible.

**19.** L. 100-103.
Here could cite Fig. 1.

**20.** L. 111.
PBL height detection becomes increasingly difficult with increasing range (because of the decrease in the signal-to-noise ratio), and because of the low power of the ceilometer deep boundary layers are hard to detect. The moderate PBL height means that it is less of an issue in this study, and the authors could mention this to support their choice of instrumentation.

**21.** L. 112-115.
"Summer dust outbreaks in the eastern Mediterranean are quite rare (Alpert and Ziv 1989, Alpert et al., 2000) therefore, they were not addressed here, especially in the height levels below 1 km (Alpert et al., 2002)." - The sentence structure is unclear. Do the authors mean that especially dust outbreaks below 1 km were not addressed, or perhaps that the dust outbreaks below 1 km were especially rare and therefore not addressed? Should be clarified.

**22.** L. 119.
The abbreviation LST is not defined.

**23.** L. 116-138.
This is a paragraph about PBL structure and development in the studied region based on literature. It is useful and informative, even though it is concise and provides a lot of information for someone not familiar with the region. This paragraph is crucial for understanding the results, and the authors should not be afraid to extend if necessary to better understand the results. They should also refer back to this section at later parts of the manuscript when the concepts described are discussed. Furthermore, Fig. 3b could also be referred to as an example to aid the description of the diurnal cycle.

**24.** L. 116-138.
The use of abbreviations seems excessive: SBF  and RL are only used once after being introduced, and could therefore omitted. Also CBL and SBL are only used 1-2 times after this paragraph and the need for the abbreviations is questionable and does not aid readability of the manuscript.

**25.** L. 136-138.
Please provide reference(s) for nocturnal PBL in Israel, if available.

**26.** Sect. 4.1
The placement of ceilometers in the heterogeneous research area should be described. Do the

ceilometer sites adequately represent the variability of the region? Are the different regions mentioned in the text (humid, arid, coastal, complex terrain) covered by the measurements?

**27.** Sect 5.3
The ceilometer backscatter profile is related to the aerosol loading, and therefore the layer that is detected is actually a aerosol layer. Implicit in the method described is the assumption that the PBL height corresponds to the height of the aerosol layer directly above ground. This assumption should be stated, and potential consequences to the results discussed. It is especially a limitation for detecting internal boundary layers which might develop due to the sea breeze circulation or katabatic winds.

**28.** L. 143 & Tables 1 and 2.
Table 2 is mentioned before Table 1 in text, the order of the tables should be swapped.

**29.** L. 156.
The authors could consider using the word "increased" rather than "improved" because it is more neutral. Although the model performance might have improved in important aspects due to increase in resolution, the computational cost likely did not.

**30.** L. 163-164.
"The spatial resolution of the models affects their ability to refer to the actual topography rather than a smoothed grid point." Is this the reason that the ceilometer site is used as a parameter for the correction? If so, it should be clarified.

**31.** L. 164-165.
"the models' results were corrected by the actual ground base heights for each measurement site" - Unfortunately I cannot follow here. Presumably the correction meant here is not the correction presented in Sect. 6.3. Perhaps the authors mean that the model levels were adjusted based on the precise altitude of each ceilometer station? Clarification would be appreciated.

**32.** L. 144-162.
Considering that IFS provides boundary conditions for COSMO, and that the description of the COSMO model refers to IFS model parameterizations, the authors could consider switching the order of introducing the two models. e.g. move lines 156-165 before line 144.

**33.** L. 157.
It seems that the IFS has more vertical levels, but does it have better vertical resolution in the boundary layer? Information on vertical resolution should be added in Table 2.

**34.** L. 188-189.
"In order to derive the backscatter coefficient from ceilometer measurements, signal calibrations and water vapor corrections are necessary" - It is not clear if the corrections were done (presumably not), and should be clarified.

**35.** L. 193-194.
It could be mentioned that averaging multiple profiles improves the signal-to-noise ratio and thereby is likely to also improve the detection of the PBL height.

**36.** L. 197.
The overlap effect is a well known issue for lidar systems, however, the authors could provide a reference.

**37.** L. 215-217.

"the radiosonde's horizontal position is under 0.01° which is an order of magnitude from the models' grid resolution" - This is true for IFS but not for COSMO, which has a resolution of 0.025°. The authors should be more specific to avoid a misleading statement.

**38.** L. 239-241.

The method used for COSMO, why two different thresholds are needed, and how it differentiates from that used in for IFS or the radiosondes is not clear. What is the reason for applying a different criteria for COSMO than the IFS and soundings?

**39.** L. 282-283.

"This height indicates the entrainment zone rather than the actual cloud top." - For anything than the most optically thin clouds, the ceilometer signal attenuates before reaching the cloud top. Therefore, the ceilometer is very unlikely to be detecting cloud top.

**40.** L. 292-293.

Considering the change in IFS resolution between 2015 and 2016, is it appropriate to evaluate the IFS data together, or should data from 2015 be considered separately from 2016?

**41.** L. 302.

In the introduction it is mentioned that Ketterer et al. (2014) found poor correlation between ceilometer PBL height and the PBL height from COSMO. Why is their result so different from that found here?

**42.** L. 310 – 314.

As far as I can see in Fig. 2, the gap between $IFS_P$ and RS is even larger for the data point indicated by the red rectangle in the figure below. I appreciate that the authors give an explanation to the anomalous PBL height on the 17 Aug 2016, but I'm concerned that this paragraph is slightly misleading. I'm not convinced that the difference between the $IFS_P$ and RS is the largest on 17 Aug 2016.

[Figure]

I suggest the authors re-formulate this paragraph with the emphasis on giving an explanation for the anomalous PBL on 17 Aug 2016, rather than claiming this is the day with largest discrepancies, or

alternative provide an objective measure for a "largest gap" and an explanation why the large discrepancy in IFS$_R$ is worth considering but the even larger discrepancy in IFS$_P$ on another day is omitted. Based on the next section, I could guess that these data points indicated by the red box are from 10 Aug 2015 (Fig 4b). If so, please include this information in this section of the manuscript.

**43.** Sect 6.1.
No discussion about the differences between bulk Richardson and parcel method is included. From Tables 4 and 5 it seems like IFS results are more sensitive to the choice of method. Perhaps the authors could discus these results.

**44.** Sect 6.1.
As far as I can understand, the main purpose of this chapter is to demonstrate the feasibility of ceilometer measurements to use for model evaluation. The authors could consider using this 33 point data set to compare the model results to the ceilometer to see if the results are similar than those obtained in comparison with the radiosondes to give additional confidence.

**45.** L. 324-330.
If the 13 days evaluated in Sect 6.2. are also included in the analysis of Sect 6.1, this paragraph does not provide any new information. For the clarity of the manuscript, I would advice the authors to include all comparison of radiosonde with other data in Sect 6.1, and focus on the spatial analysis in Sect. 6.2, as indicated by the title.

**46.** L. 331.
"By and large, COSMO$_R$ achieved the best statistical results" - This statement seems overemphasized. In terms of root mean square error, COSMO$_P$ performed better on 4 of the 5 sites presented, and the mean error was better for 2 sites.

**47.** L. 336-349.
"These results emphasize the advantage of high-resolution regional models such as COSMO (~2.5 km resolution) over the IFS global model (resolution of ~13 km in 2015 and ~10 km in 2016) over a diverse area." Although not necessarily surprising, this is one of the few clear results of the paper, and deserves to be discussed and possibly further analyzed. Is the poor performance of the IFS related to lacking representation of the sea breeze circulation or some local scale phenomena?

**48.** Sections 6.1 and 6.2.
Did the authors consider the differences between the bulk Richardson and parcel method, and whether it indicates certain shortcomings in the models description of the boundary layer structure or processes? Comparing the COSMO$_R$ and COSMO$_P$ mean errors presented in Table 5, it seems that the two methods produce more similar results more inland (Ramat David and Jerusalem) than closer to the coast (Tel Aviv, Beit Dagan, Weizmann). This seems to also hold for the IFS. Is this related to the meteorological conditions, or simply a coincidence? Again, a significantly larger data set would be desirable.

**49.** Section 6.2.
Why are only 5 sites included, if ceilometers are available at 8? No station with the description "South" is included in the analysis of spatial variability (Table 1, L. 320), do the included 5 ceilometer sites adequately represent the spatial variability of the studied region?

**50.** L. 342-344.
 "Following the conclusions of previous stages, COSMO$_R$ was chosen as the model and method that achieved the best results." In my opinion, this was not well demonstrated (see also comment 46).

**51.** L. 344.

I'm guessing that the time window chosen is somehow related to the diurnal PBL height cycle that was nicely described in Sect. 2. Please provide explanation for the time chosen.

**52.** Fig. 4 and Sect. 6.2.

How are daily values obtained? Is the procedure the same as in Sect. 6.1, e.g. estimating the PBL height at approximately 11 UTC? If so, it should be mentioned in the text.

**53.** L. 349-357.

I'm not sure I understand the correction procedure. First, the variables α, β and γ are obtained by using the mean error (ME) between model and ceilometer at each station, and the altitude and distance from shoreline as predictor variables. After α, β and γ are obtained, it is possible to estimate ME anywhere in the domain. The corrected PBL height is then the COSMOR PBL height + the ME that is computed using altitude, distance from shoreline and α, β and γ. The same procedure is repeated for each hour, resulting in a time dependent α, β and γ. Is this a correct interpretation? The authors should clarify the description of their method.

**54.** L. 349-357.

Could the authors report the values of α, β and γ? The choice of repeating the correction for each hour of the day suggest some dependence of the correction needed on the diurnal cycle, does that exist? Do α, β and γ vary from hour to hour? What is the role of γ in the equation, and is it really needed? Presenting α and β would show whether altitude (e.g. topography) or distance from the shoreline (e.g. sea breeze circulation?) contributes more to the model discrepancy.

**55.** L. 358

Is the cross-section along a fixed longitude?

**56.** L. 369-370.

"The lowest value was corrected from 09 UTC (11 LST) to 14 UTC (16 LST)" - The way I understand this sentence is that the the lowest value was before the correction at 9 UTC, and after the correction it was at 14 UTC. This seems to contradict Fig. 5, which shows the opposite. Comparing Figures 5 a and b, it seems that the uncorrected data had the lowest PBL height at 14 UTC (independent of longitude). After the correction, at longitudes eastward of 35.1° (where Jerusalem lies) the lowest PBL height is found at 9 UTC. It would be advisable for the authors to clarify their statement.

**57.** L. 403.

"which improved the description of the diurnal PBL heights" - Unfortunately, there is no evidence presented that the model performance would have improved. See comment 6.

**58.** Conclusions.

The authors could discuss how the results obtained for daytime in a summer month might compare to other seasons.

**59.** Table 1.

Height limit is given as 7.7 or 15.4 km, but the footnote states that the data acquisition was limited to 4.5 km. It is not clear what is the vertical extent of the measurement. Although it is not that important for the study, the presentation is confusing and could be clarified.

**60.** Table 1.

The table includes specifications for the sites such as "north", "south", "inland", "mountain", but these do not seem to be defined or used elsewhere in the manuscript. Perhaps the regions could

provisionally be indicated on a map, and used in the discussion of the results.

**61.** Table 3.
For completeness, the table could include the mean and standard deviation also from the radiosonde used as a reference.

**70.** Table 4.
"The PBL heights were compared to the heights measured by the Beit Dagan ceilometer." The text states (lines 321-322) "the models' results were compared to the ceilometers' measurements in each site". These two statements seem to contradict each other, and I would ask the authors to correct one of them, or to clarify why different comparison measurements are considered in the text and in the table.

**71.** Tables 4 and 5.
It would be interesting to also see the mean PBL height of the ceilometer (the reference) at each site.

**72.** Figures 1 and 6.
Considering the political situation in some areas of Western Asia, the authors should carefully consult the journals guidelines regarding maps.

**73.** Fig. 3a.
The figure could contain the PBL height estimated by the two methods. It would be helpful to demonstrate the performance of the two methods.

**74.** Fig 3b.
It does not look like the data has been averaged for 30 min. Is the data presented at original 15 sec resolution? Please clarify in the caption.

**75.** Fig. 3b.
The authors should consider showing the time series of ceilometer and model based PBL height in this figure. It would be interesting to see 1) how the wavelet covariance transformation method is performing on the time series presented, 2) how the models predict the temporal development of the PBL height, and 3) whether the difference between model and ceilometer is random or the two models and two methods are consistently over or underestimating the PBL height during this one day. Although it might seem trivial to the authors, this helps the reader to gain confidence in the methods and helps with the understanding of the diurnal cycle of the PBL that is described in Sect. 2.

**76.** Fig. 3c.
The results presented here are not discussed. A description of the results presented here, and the ways they help to interpret Fig. 3 a and b or other results should be added. Furthermore, the wind direction figure could be improved by shifting the x-axis so that it is centered around North (e.g. scale from 180 to 360/0 to 180 degrees).

**77.** Fig. 4.
Figure 4 is hardly mentioned in the manuscript (it is referred to in the caption of Table 4, and Fig 4b is mentioned on line 326). Consequently, it is not clear what this figure is communicating. What is the additional information provided that is not already presented in Fig. 2? The better performance of COSMO compared to IFS, and the good agreement of ceilometer and radiosonde (Fig. 4b) are already demonstrated in Sect. 6.1.

**78.** Fig. 5

Figure 5 could indicate the locations of the Tel Aviv and Jerusalem ceilometer stations, as well as the mean (and standard deviation) of the PBL height estimated at these sites.

**79.** Figure 5 and 6.

I don't think it is necessary to list the sites and number of days used for the analysis in each figure caption. In my opinion simply a reference to the text for more details would do.

**80.** Fig 6.

Figure 6 could include the information of the mean PBL height at the stations.

**81.** Fig. 6b.

It is not clear what variable is presented in Fig 6b. Is it the ME estimated based on Equation 6, or one of the fitted parameters ($\alpha$, $\beta$, $\gamma$)?

**82.** Citations.

The authors should check their citations and list of references list. For example, Uzan et al. (2012) and Uzan et al (2018) are cited but missing from the the reference list

**83.** Figures.

The authors should pay attention to the quality of figures. The font size could be increased in almost all figures (especially hard to read is Fig. 3), and use of color-blind friendly colors should be considered.

---

## Author Response (AR2)

**Author's response to referee #3:**

Thank you very much for the thorough, explicit, well organized, and practical comments. The review comes as an opportunity to improve the manuscript in various aspects. For convenience, the response is by order of appearance following the structure of the referee's report.

Referee's comment #1a) The motivation, strengths, and the central question of the paper could be made clearer in the introduction of the paper. As explained in Sect. 2, the region studied is interesting and different aspects are impacting the PBL height. The interesting aspects of the spatial variability of the studied region could be included in the introduction. In the light of the spatial variability, evaluating model performance on a single site would have limited value. One of the strengths of the study is the use of a network of ceilometers that can estimate the temporal development of the PBL at various locations simultaneously. This aspect deserves to be mentioned in the introduction.

Author's response: **Comment accepted.**

Author's changes in manuscript: The abstract and introduction paragraphed were changed accordingly.

Referee's comment #1b) The introduction does not provide enough information to motivate the development of a post-processing tool for the modeled PBL height. The goal to use ceilometer detected PBL height to correct for modeled PBL height could be simplified to "use A to correct B". Currently, it is demonstrated that to "use A" is possible, e.g. PBL height can, with some limitations, be retrieved from the ceilometer measurements. However, "to correct B" is neglected in the introduction. The introduction only states the need for accurate PBL estimate, but no literature on identified shortcomings, methods found for improvement or anything else that would have been done previously to evaluate or improve PBL height estimates in NWP models is presented. Do previous studies suggest that it is more feasible to correct the end product (e.g. the PBL height) than to improve model parametrizations in order to obtain a better result from the model? Do the authors envision a use for the corrected PBL height? The authors could also consider whether their main aim should be on developing a correction, or rather a rigorous evaluation of model performance in the complex region. The latter could be helpful for understanding model shortcomings and would be a more general result than a location and time specific correction.

Author's response: **Comment accepted.**

Author's changes in manuscript: The motivation of this study is to provide air pollution dispersion models with reliable input data of PBL heights. Weather models produce a high spatial and temporal resolution of PBL heights, albeit previous research has shown significant differences between the models' estimations and actual measurements. To overcome this obstacle, we established a correction tool for weather models by employing ceilometer measurements.

Referee's comment #2) One of the confusing aspects of this paper is the small number of days analyzed. The strength of the ceilometer is that data acquisition is cheap (see Sect. 1), however the small dataset is undermining this specific strength. The conclusions drawn are seriously undermined by the small sample size. For example, Sect. 6.2 seems to describe statistical results obtained from 13 data points. If possible, the authors should obtain more data. Alternatively, the study could be shifted to focus on case studies evaluating the shortcomings of the models in more detail. Although the reasons for focusing on daytime PBL only in summer are given, further selection seems to have taken place. Why are only 13 days included from August 2015, and 20 days from August 2016 (L. 292-293) in Sect. 6.1? Why does Sect. 6.2 only include 5 ceilometer sites, when Sect. 6.3 includes 8 ceilometer sites (L. 319-321 and 345-346)? Why do Sections 6.2 and 6.3 only include data from August 2015, and not from August 2016? Are the 13 days used in Sect. 6.2 a subset of the 33 days in Sect 6.1? The authors should provide an explanation for the small number of days analyzed and why certain days and sites were selected at different stages of the study.

Author's response: **Comment accepted.** The ceilometer array in Israel is a collection of ceilometers from different institutes. This study was the first attempt to gather data from all institutions. Unfortunately, some output files are missing. In other cases, the ceilometers operated for short periods. The database further narrowed down by removing days with dust storms or partial data. Eventually, we extracted the maximum days available for each ceilometer within six summer months: July-September 2015, and June-August 2016. We produced additional IFS and COSMO model runs to meet the periods available from the ceilometers. As a result, the analysis expanded from 13 specific days for 5 ceilometers to above 50 days for 6 ceilometers:

| Ceilometer | # Days |
|------------|--------|
| Bet Dagan | 91 |
| Tel Aviv | 122 |
| Ramat David | 123 |
| Weizmann | 55 |
| Jerusalem | 53 |
| Nevatim | 72 |

Hence, we combined sections 6.1 and 6.2 in section 5.1. Section 6.3 changed to Sect. 5.2.

Author's changes in manuscript:

The results and conclusions sections were changed considerably, as aforementioned.

Referee's comment #3) Related to the comment above about the amount of data, the authors should consider the statistical significance of the presented results. Specifically, wherever R-values are given (L. 298, Table 3, and elsewhere), the corresponding p-value should also be presented. Other techniques to analyze the statistical significance of the results are also welcomed, and the results should be discussed from the point of view of statistical significance.

Author's response: **Comment accepted.**

Author's changes in manuscript: Statistical analysis of boxplots, histograms, and tables added.

Referee's comment #4) Section 6.2 could provide possibly the most interesting results for considering model performance in terms of PBL height in complex environments. If model under- or overestimation could be connected to certain processes (e.g. the sea breeze), the results would be more generally interesting. Mountainous coastlines are not unique to Israel, and many people inhabit such areas. This section deserves a proper evaluation, and the analysis and discussion should be extended. Specifically, this section is hard to understand for someone not familiar with the geography of Israel. I would advise the authors to consider the presentation of their results. For example, the mean error at each site for each model and method could be presented with a symbol on a map having the color indicating the value. This would make any spatial structures in the mean, mean error (ME) or root mean square error (RMSE) more apparent. The authors could also plot the ME and/or RMSE as a function of the distance of the site to the shoreline and altitude above sea level (these are the two variables used for the correction in the next section). From the authors description of the situation, it seems that the sea breeze has a clear influence on the PBL height. Is it to be understood, that the model does not correctly produce the sea breeze circulation, or is the model lacking in terms of the effect of the sea breeze on PBL height? It would be interesting if the authors could evaluate the discrepancy between ceilometer and model PBL height in terms of the strength, and spatial and temporal development of the sea breeze circulation during the day. Furthermore, in Sect. 6.3 data for 9-14 UTC are used, and I suggest the authors consider including the temporal development of the PBL height in their analysis in Sect. 6.2 as well.

Author's response: **Comment accepted.**

Author's changes in manuscript: As described in response to comment #3, this section was changed dramatically following the referee's suggestions.

Referee's comment #5a) Before a correction is developed and presented, it should be made clear that a correction is needed and that there is a systematic bias that can be corrected for. Table 3 (and Section 6.1) show that the mean error of COSMOR compared to radiosondes is -3 m, which does not leave much room for improvement. Also Table 5 shows that at different sites the mean error of COSMOR is within a few tens of meters at most. (However, I would be cautious to draw conclusions from statistics comprising of 13 data points, and the authors should obtain a larger sample size if possible. See comments 2 and 3). For a 1 km deep PBL, an error of 30 m is 3%. For which application is this not good enough, and how good should the model performance be? Furthermore, considering that the definition of the planetary boundary layer is slightly ambiguous, can a perfect agreement between different methods be expected? The authors should explain why they think the model performance is not good enough and requires improvement. Furthermore, the authors could consider if the correction they presented would actually be more useful for the IFS model that shows clearly worse performance than the COSMO in terms of PBL height prediction.

Author's response: **Comment accepted.**

Author's changes in manuscript: With great effort, we obtained a larger sample size. Now, the necessity to improve the models' PBL heights is evident from the statistical analysis. The primary purpose of this study was to improve the performance of air pollution dispersion models by providing reliable PBL heights from NWP models. In some cases (Uzan et al., 2012), a height difference of 100 m between the actual PBL height and the models' assessments affect ground-level air pollution concentrations significantly. Therefore, the correction tool is useful for both regional and global models.

Uzan, L. and Alpert, P.: The coastal boundary layer and air pollution - a high temporal resolution analysis in the East Mediterranean Coast, The Open Atmospheric Science Journal, 6, 9–18, 2012.

Referee's comment #5b) Sect. 6.2 should demonstrate the basis of the correction presented in Sect. 6.3. The fact that the mean error in Tel Aviv, Beit Dagan and Weizmann are so similar suggests a spatial consistency that is more clear for COSMOR than COSMOP. (Table 5). Is this the reason COSMOR was used for the correction in Sect. 6.3 instead of COSMOP? The fact that there seems to be some spatial structure in the mean error is promising for developing a correction. The RMSE does not seem so spatially consistent.

Author's response: **Comment accepted.**

Author's changes in manuscript: The new results section reveals which model and method produced the best results following the ceilometers' locations.

Referee's comment #5c) To justify the correction method presented in Sect. 6.3, it should be established that a bias exist in the models' PBL height estimation that depends on altitude and distance from shoreline, that could consequently, be corrected for. The authors should evaluate how the discrepancy between ceilometer and model PBL height depends on the topography and distance from shoreline. Furthermore, this could be done for different hours of the day, as the correction procedure is also applied for each hour separately.

Author's response: **Comment accepted.**

Author's changes in manuscript: Section 6.3 changed to 6.2, including an elaborate explanation of the correction tool performance for a single day study case between 9-14 UTC. Figures for each hour display the models' estimations, PBL heights after correction, and cross-validation examination for Bet Dagan and Jerusalem.

Referee's comment #6) Perhaps the most serious shortcoming of the manuscript is that it is not demonstrated that the model result is better after correction. The authors should include a quantitative evaluation of the improvement of the model PBL after the correction. For example, the radiosondes at Beit Dagan could be used as an independent reference for the model PBL height. Another approach would be to estimate the correction parameters using only some of the available ceilometer stations, and using the remaining stations as a references to estimate the improvement in PBL height achieved by the correction. Varying the number of stations and the locations of the stations included for fitting the correction parameters also give an indicator for how many ceilometers needs to be included, or how they need to be located, for achieving a significant improvement for the COSMOR PBL height. If the authors aim is to show that the ceilometer is a useful tool to improve the modeled PBL height, the strength of their paper relies on the extent and rigor that this kind of analysis is carried out.

Author's response: **Comment accepted.**

Author's changes in manuscript: Cross-validation analysis demonstrated the efficiency of the correction tool. The improvements were discussed in the conclusions according to the new results section (see responses to comments #2 and #5c).

Referee's comment #7a) More attention should be paid to make the reasoning understandable for readers that are not so familiar with the specific geography and climatology of the region. Firstly, the studied region and its interesting aspects could be mentioned in the introduction. The first time the location is given is the very end of the introduction, on line 97. This should be included already in the previous paragraph that outlines the purpose of the study, as well as in the abstract.

Author's response: **Comment accepted.**

Author's changes in manuscript: The spatial variability and locations were added to the abstract and introduction.

Referee's comment #7b) A topography map should be included. Global topography data is available (for example from NOAA https://doi.org/10.7289/V5C8276M) and a map can be drawn using openly available tools (such as python).

Author's response: **Comment accepted.**

Author's changes in manuscript: A topographical map was added.

Referee's comment #7c) Depending on the weight the authors want to give to the humidity (mentioned on lines 103-104) and the prevailing synoptic conditions (line 125), they could also include a map of mean precipitation and pressure in August to help the reader to follow their argumentation.

Author's response: The manuscript modifications doubled the number of figures. Therefore, we preferred to add references instead of maps.

Author's changes in manuscript: Additional references of previous research in Israel describing the dry summer season.

Referee's comment #8) L.1-2: The authors should reconsider the title of the manuscript. The current title is somewhat misleading because it implies that the correction for PBL height was considered for both models, when in the manuscript only the COSMO PBL height was corrected. Furthermore, the journal guidelines recommend avoiding the use of abbreviations in the title, so the authors might want to avoid the use of "NWP" in the title.

Author's response: **Comment accepted.** The research studies two NWP models and established a correction formula feasible for both models. Thus, we find it appropriate to mention IFS in the title as well.

Author's changes in manuscript: "NWP" was removed from the title.

Referee's comment #9) L.23-25: Here results are given for flat and elevated terrain. Consulting Tables 4 and 5 it seems that flat terrain refers to Tel Aviv, and elevated terrain to Jerusalem. The authors should consider mentioning the sites for which the numbers refer to avoid ambiguity, or at least mention that the values presented are from single stations.

Author's response: **Comment accepted.**

Author's changes in manuscript: The titles in the new results section refer to each ceilometer.

Referee's comment #10) The abstract does not mention Israel or give any other indication over the geographic locations apart from "heterogeneous area" and mention of the Beit Dagan radiosonde launch site. Location should be given.

Author's response: **Comment accepted.**

Author's changes in manuscript: Locations were added to the abstract.

Referee's comment #11) L.33-40: Considering that this paragraph states the broad motivation and importance of this study, some references would be appropriate.

Author's response: **Comment accepted.**

Author's changes in manuscript: References were added.

Referee's comment #12) L.56-57: "ceilometers obtain a wide spatial resolution per lidar" - I'm afraid I do not understand the meaning of this phrase. Perhaps the authors mean that a wider spatial resolution is achieved by ceilometers than lidars?

Author's response: **Comment accepted.**

Author's changes in manuscript: The text was changed accordingly.

Referee's comment #13) L.53-65: This paragraph seems to suggest that ceilometers are better than lidars in every aspect. It would be fair to mention a shortcoming of the ceilometer compared to a lidar.

Author's response: **Comment accepted.**

Author's changes in manuscript: The shortcomings of ceilometers were added to the introduction section.

Referee's comment #14) L.89-91: It is not obvious here why the summer season is more appropriate for an approach that is limited by precipitation. It is later explained that this season has low precipitation. This should also be mentioned here to help the readers not familiar with local climatology.

Author's response: **Comment accepted.**

Author's changes in manuscript: The meteorological conditions were added to the introduction section.

Referee's comment #15) L.92-97: It would be possible to help the reader further by outlining the structure of Sect. 6, either here or at the beginning of Sect. 6.

Author's response: **Comment accepted.**

Author's changes in manuscript:  The outline was elaborated accordingly.

Referee's comment #16) L.85-86: The introduction demonstrates the strengths of ceilometers compared to other available observational techniques to estimate PBL height, but only states that ceilometers have not been used often for evaluating model performance. However, other observational techniques have, and this should be mentioned. Specifically, have other observational tools been used for evaluating PBL height in NWP models in Israel, or other mountainous coastlines?

Author's response: **Comment accepted.**

Author's changes in manuscript: Information regarding the observational tools implemented for COSMO PBL height evaluation was added in the introduction section.

Referee's comment #17) I find the extent of presenting the literature for the use of ceilometer to detect PBL height satisfactory. However, no mention of previous work using ceilometer to derive PBL height in Israel is presented. The authors should site at least Uzan et al. (2016) and any other studies employing the measurement technique in their region of study.

Author's response: **Comment accepted**. As discerned by the referee, we were first to employ ceilometers for PBL height detection in Israel (Uzan el al, 2016). Up until our research, the ceilometers' in Israel were acknowledged merely as ceiling height detectors. Thus, historical data had neither been acquired or saved. The data we received was collected following our specific request. It was the maximum amount of data available. This explains the inevitable situation of low data availability for spatial analysis limit to the summer season.

Author's changes in manuscript: Uzan et al. (2016) was cited in the introduction.

Referee's comment #18) L.106: "IMS weather reports" - The authors should provide a more specific reference, if possible.

Author's response: **Comment accepted.**

Author's changes in manuscript: **"**Israeli Meteorological Service relative humidity climate report 1995-2009, https://ims.gov.il/en/ClimateReports**".**

Referee's comment #19) L.100-103: Here could cite Fig. 1.

Author's response: **Comment accepted.**

Author's changes in manuscript: Fig 1 was cited accordingly.

Referee's comment #20) L.111: PBL height detection becomes increasingly difficult with increasing range (because of the decrease in the signal-to-noise ratio), and because of the low power of the ceilometer deep boundary layers are hard to detect. The moderate PBL height means that it is less of an issue in this study, and the authors could mention this to support their choice of instrumentation.

Author's response: **Comment accepted.**

Author's changes in manuscript: The comment was added to the text. Thank you.

Referee's comment #21) L.112-115: "Summer dust outbreaks in the eastern Mediterranean are quite rare (Alpert and Ziv 1989, Alpert et al., 2000) therefore, they were not addressed here, especially in the height levels below 1 km (Alpert et al., 2002)." - The sentence structure is unclear. Do the authors mean that especially dust outbreaks below 1 km were not addressed, or perhaps that the dust outbreaks below 1 km were especially rare and therefore not addressed? Should be clarified.

Author's response: **Comment accepted.**

Author's changes in manuscript: The sentence was clarified in the text.

Referee's comment #22) L.119: The abbreviation LST is not defined.

Author's response: **Comment accepted.**

Author's changes in manuscript: LST = UTC+2 was added to the text.

Referee's comment #23) L.116-138: This is a paragraph about PBL structure and development in the studied region based on literature. It is useful and informative, even though it is concise and provides a lot of information for someone not familiar with the region. This paragraph is crucial for understanding the results, and the authors should not be afraid to extend if necessary to better understand the results. They should also refer back to this section at later parts of the manuscript when the concepts described are discussed. Furthermore, Fig. 3b could also be referred to as an example to aid the description of the diurnal cycle.

Author's response: **Comment accepted.**

Author's changes in manuscript: Changes were made according to the figures and text of the new results section.

Referee's comment #24) L.116-138: The use of abbreviations seems excessive: SBF and RL are only used once after being introduced, and could therefore omitted. Also CBL and SBL are only used 1-2 times after this paragraph and the need for the abbreviations is questionable and does not aid readability of the manuscript.

Author's response: **Comment accepted.**

Author's changes in manuscript: The abbreviations-SBF, SBL, CBL, and RL were removed.

Referee's comment #25) L.136-138: Please provide reference(s) for nocturnal PBL in Israel, if available.

Author's response: Previous studies of the nocturnal PBL in Israel were conducted in regions not in the scope of our research, therefore, they were not cited.

Author's changes in manuscript: No change was made.

Referee's comment #26) Sect. 4.1: The placement of ceilometers in the heterogeneous research area should be described. Do the ceilometer sites adequately represent the variability of the region? Are the different regions mentioned in the text (humid, arid, coastal, complex terrain) covered by the measurements?

Author's response: **Comment accepted.**

Author's changes in manuscript: The region of the ceilometers was added in the relevant sections.

Referee's comment #27) Sect.5.3: The ceilometer backscatter profile is related to the aerosol loading, and therefore the layer that is detected is actually an aerosol layer. Implicit in the method described is the assumption that the PBL height corresponds to the height of the aerosol layer directly above ground. This assumption should be stated, and potential consequences to the results discussed. It is especially a limitation for detecting internal boundary layers which might develop due to the sea breeze circulation or katabatic winds.

Author's response: **Comment accepted.**

Author's changes in manuscript: An explanation was added to Sect. 1.

Referee's comment #28) L.143: Table 2 is mentioned before Table 1 in text, the order of the tables should be swapped.

Author's response: The explanation of the research area was moved to the introduction section therefore, it wasn't necessary to swap the table numbers.

Author's changes in manuscript: No change made.

Referee's comment #29) L.156: The authors could consider using the word "increased" rather than "improved" because it is more neutral. Although the model performance might have improved in important aspects due to increase in resolution, the computational cost likely did not.

Author's response: **Comment accepted.**

Author's changes in manuscript: The section was rephrased.

Referee's comment #30) L.163-164: "The spatial resolution of the models affects their ability to refer to the actual topography rather than a smoothed grid point." Is this the reason that the ceilometer site is used as a parameter for correction? If so, it should be clarified.

Author's response: **Comment accepted.**

Author's changes in manuscript: An explanation was aded to the new summary and conclusions section.

Referee's comment #31) L.164-165: "the models' results were corrected by the actual ground base heights for each measurement site" - Unfortunately, I cannot follow here. Presumably the correction meant here is not the correction presented in Sect. 6.3. Perhaps the authors mean that the model levels were adjusted based on the precise altitude of each ceilometer station? Clarification would be appreciated.

Author's response: **Comment accepted.**

Author's changes in manuscript: Additional text: "Therefore, the models' levels were adjusted based on the precise altitude of each ceilometer station."

Referee's comment #32) L.144-162: Considering that IFS provides boundary conditions for COSMO, and that the description of the COSMO model refers to IFS model parameterizations, the authors could consider switching the order of introducing the two models. e.g. move lines 156-165 before line 144.

Author's response: **Comment accepted.**

Author's changes in manuscript: The order was changed. IFS was introduced before COSMO.

Referee's comment #33) L.157: It seems that the IFS has more vertical levels, but does it have better vertical resolution in the boundary layer? Information on vertical resolution should be added in Table 2.

Author's response: **Comment accepted.**

Author's changes in manuscript: The information was added to Table 2.

Referee's comment #34) L.188-189: "In order to derive the backscatter coefficient from ceilometer measurements, signal calibrations and water vapor corrections are necessary" - It is not clear if the corrections were done (presumably not), and should be clarified.

Author's response: **Comment accepted.**

Author's changes in manuscript: The sentence was rephrased.

Referee's comment #35) L.193-194: It could be mentioned that averaging multiple profiles improves the signal-to-noise ratio and thereby is likely to also improve the detection of the PBL height.

Author's response: **Comment accepted.**

Author's changes in manuscript: The sentence was rephrased.

Referee's comment #36) L.197: The overlap effect is a well-known issue for lidar systems, however, the authors could provide a reference.

Author's response: **Comment accepted.**

Author's changes in manuscript:  A reference was added: "At these heights, a constant perturbation existed due to the overlap of the emitted laser beam and the receiver's field of view (Weigner et al., 2014)".

Referee's comment #37) L.215-217: "the radiosonde's horizontal position is under 0.01° which is an order of magnitude from the models' grid resolution" - This is true for IFS but not for COSMO, which has a resolution of 0.025°. The authors should be more specific to avoid a misleading statement.

Author's response: **Comment accepted.**

Author's changes in manuscript: The text was rephrased.

Referee's comment #38) L.239-241: The method used for COSMO, why two different thresholds are needed, and how it differentiates from that used in for IFS or the radiosondes is not clear. What is the reason for applying a different criterion for COSMO than the IFS and soundings?

Author's response: **Comment accepted.** IFS adapted a single threshold of 0.25 following the conclusions of (Seidal et al.,2015). The COSMO model refers to 0.33 for stable atmospheric conditions (Wetzel, 1982), and 0.22 for unstable conditions by 0.22 (Vogelezang and Holtslag, 1996).

 Author's changes in manuscript: The information was added to the text.

Referee's comment #39) L.282-283: "This height indicates the entrainment zone rather than the actual cloud top." For anything than the most optically thin clouds, the ceilometer signal attenuates before reaching the cloud top. Therefore, the ceilometer is very unlikely to be detecting cloud top.

Author's response: **Comment accepted.** We must clarify we didn't attempt to claim the ceilometer detects the cloud top. On the contrary.

Author's changes in manuscript:  The sentence was rephrased to avoid the misunderstanding.

Referee's comment #40) L.292-293: Considering the change in IFS resolution between 2015 and 2016, is it appropriate to evaluate the IFS data together, or should data from 2015 be considered separately from 2016?

Author's response: In 2015 and 2016 the ceilometers were indicated by the same grid points and horizontal levels. Therefore, we did not find it necessary to separate the results. Furthermore, we ran the analysis separately for 2015 and 2016. The difference between the results was insignificant.

Author's changes in manuscript: No changes made.

Referee's comment #41) L.310-314: In the introduction it is mentioned that Ketterer et al. (2014) found poor correlation between ceilometer PBL height and the PBL height from COSMO. Why is their result so different from that found here?

Author's response: **Comment accepted.** The main difference is the research area. Ketterer et al., (2014) studied complex topography of the Swiss Alps (two sites, 3,580 m a.s.l and 2,061 m a.s.l), whist our stud region was confined between the shoreline to highest point of 830 m a.s.l.

Author's changes in manuscript: To avoid a too-long introduction section, we moved the discussion of previous research (Ketterer et al., 2014 and Collaud et al., 2014) to the results section.

Referee's comment #42) As far as I can see in Fig. 2, the gap between IFSP and RS is even larger for the data point indicated by the red rectangle in the figure below. I appreciate that the authors give an explanation to the anomalous PBL height on the 17 Aug 2016, but I'm concerned that this paragraph is slightly misleading. I'm not convinced that the difference between the IFSP and RS is the largest on 17 Aug 2016. I suggest the authors re-formulate this paragraph with the emphasis on giving an explanation for the anomalous PBL on 17 Aug 2016, rather than claiming this is the day with largest discrepancies, or alternative provide an objective measure for a "largest gap" and an explanation why the large discrepancy in IFSR is worth considering but the even larger discrepancy in IFSP on another day is omitted. Based on the next section, I could guess that these data points indicated by the red box are from 10 Aug 2015 (Fig 4b). If so, please include this information in this section of the manuscript.

Author's response: **Comment accepted.** The new results section consists of new figures according to the referee's comments 2-6.

Author's changes in manuscript: The data of Fig. 2 and new for other ceilometers was analyzed to produce new compelling figures.

Referee's comment #43) Sect 6.1. No discussion about the differences between bulk Richardson and parcel method is included. From Tables 4 and 5 it seems like IFS results are more sensitive to the choice of method. Perhaps the authors could discuss these results.

Author's response: **Comment accepted.**

Author's changes in manuscript: The new results section consists of a discussion on the different methods.

Referee's comment #44) Sect 6.1: As far as I can understand, the main purpose of this chapter is to demonstrate the feasibility of ceilometer measurements to use for model evaluation. The authors could consider using this 33 point data set to compare the model results to the ceilometer to see if the results are similar than those obtained in comparison with the radiosondes to give additional confidence.

Author's response: **Comment accepted.**

Author's changes in manuscript: The results section was changed accordingly.

Referee's comment #45) L.324-330: If the 13 days evaluated in Sect 6.2. are also included in the analysis of Sect 6.1, this paragraph does not provide any new information. For the clarity of the manuscript, I would advise the authors to include all comparison of radiosonde with other data in Sect 6.1, and focus on the spatial analysis in Sect. 6.2, as indicated by the title.

Author's response: **Comment accepted.**

Author's changes in manuscript: The results section was changed accordingly.

Referee's comment #46) L.331: "By and large, COSMOR achieved the best statistical results" - This statement seems overemphasized. In terms of root mean square error, COSMOP performed better on 4 of the 5 sites presented, and the mean error was better for 2 sites.

Author's response: **Comment accepted.**

Author's changes in manuscript: The new results section consists of a discussion on the results of each model by each method.

Referee's comment #47) L.336-349: "These results emphasize the advantage of high-resolution regional models such as COSMO (~2.5 km resolution) over the IFS global model (resolution of ~13 km in 2015 and ~10 km in 2016) over a diverse area." Although not necessarily surprising, this is one of the few clear results of the paper, and deserves to be discussed and possibly further analyzed. Is the poor performance of the IFS related to lacking representation of the sea breeze circulation or some local scale phenomena?

Author's response: **Comment accepted.**

Author's changes in manuscript: An explanation was added to the new summary and conclusions section.

Referee's comment #48) Sect. 6.1 and 6.2: Did the authors consider the differences between the bulk Richardson and parcel method, and whether it indicates certain shortcomings in the models description of the boundary layer structure or processes? Comparing the COSMOR and COSMOP mean errors presented in Table 5, it seems that the two methods produce more similar results more inland (Ramat David and Jerusalem) than closer to the coast (Tel Aviv, Beit Dagan, Weizmann). This seems to also hold for the IFS. Is this related to the meteorological conditions, or simply a coincidence? Again, a significantly larger data set would be desirable.

Author's response: **Comment accepted**.

Author's changes in manuscript: The new result section consists of a discussion on the differences between the models.

Referee's comment #49) Sect. 6.2: Why are only 5 sites included, if ceilometers are available at 8? No station with the description "South" is included in the analysis of spatial variability (Table 1, L. 320), do the included 5 ceilometer sites adequately represent the spatial variability of the studied region?

Author's response: **Comment accepted.**

Author's changes in manuscript: The new results section refers to these comments. See response for comment #2.

Referee's comment #50) L.342-344: "Following the conclusions of previous stages, COSMOR was chosen as the model and method that achieved the best results." In my opinion, this was not well demonstrated (see also comment 46).

Author's response: **Comment accepted.**

Author's changes in manuscript: The new results section includes a discussion of the results by models, methods, and location of the measurement sites.

Referee's comment #51) L.344: I'm guessing that the time window chosen is somehow related to the diurnal PBL height cycle that was nicely described in Sect. 2. Please provide explanation for the time chosen.

Author's response: **Comment accepted.** In the summer season, stable conditions prevail from sunset to an hour after sunrise (Stull, 1988). At this period the models' $R_b$ profiles do not accede the relevant thresholds, and the PBL height is not detected. Subsequently, the analysis fixated on the day time hours, after sunrise and before sunset

Author's changes in manuscript: An explanation was added the results section**.**

Referee's comment #52) Fig. 4: How are daily values obtained? Is the procedure the same as in Sect. 6.1, e.g. estimating the PBL height at approximately 11 UTC? If so, it should be mentioned in the text.

Author's response: **Comment accepted.**

Author's changes in manuscript:  Fig. 4 was replaced by new figures following the referee's recommendations to expand the dataset. See response for comment #2.

Referee's comment #53) L.349-357: I'm not sure I understand the correction procedure. First, the variables α, β and γ are obtained by using the mean error (ME) between model and ceilometer at each station, and the altitude and distance from shoreline as predictor variables. After α, β and γ are obtained, it is possible to estimate ME anywhere in the domain. The corrected PBL height is then the COSMOR PBL height+ the ME that is computed using altitude, distance from shoreline and α, β and γ. The same procedure is repeated for each hour, resulting in a time dependent α, β and γ. Is this a correct interpretation? The authors should clarify the description of their method.

Author's response**: Comment accepted.**

Author's changes in manuscript: The explanation of the correction tool was changed accordingly.

Referee's comment #54) L.349-357: Could the authors report the values of α, β and γ? The choice of repeating the correction for each hour of the day suggest some dependence of the correction needed on the diurnal cycle, does that exist? Do α, β and γ vary from hour to hour? What is the role of γ in the equation, and is it really needed? Presenting α and β would show whether altitude (e.g. topography) or distance from the shoreline (e.g. sea breeze circulation?) contributes more to the model discrepancy.

Author's response: **Comment accepted.**

Author's changes in manuscript: The new results section provides the dependent variables α, β, and the constant γ for each hour (9-14 UTC) for three scenarios: regression by eight ceilometers, regression by seven ceilometers excluding the plain site of Bet Dagan, regression by seven ceilometers excluding the elevated site of Jerusalem.

Referee's comment #55) L.358: Is the cross-section along a fixed longitude?

Author's response: **Comment accepted.**

Author's changes in manuscript: The new results delineate PBL heights from all ceilometers by distance from the shoreline.

Referee's comment #56) L.369-370: "The lowest value was corrected from 09 UTC (11 LST) to 14 UTC (16 LST)" - The way I understand this sentence is that the lowest value was before the correction at 9 UTC, and after the correction it was at 14 UTC. This seems to contradict Fig. 5, which shows the opposite. Comparing Figures 5 a and b, it seems that the uncorrected data had the lowest PBL height at 14 UTC (independent of longitude). After the correction, at longitudes eastward of 35.1º (where Jerusalem lies) the lowest PBL height is found at 9 UTC. It would be advisable for the authors to clarify their statement.

Author's response: **Comment accepted**. New figures in the results section clarify the results of the correction tool, hour by hour, from all ceilometer sites.

Author's changes in manuscript: New figures and explanations.

Referee's comment #57) Line.403: "which improved the description of the diurnal PBL heights" - Unfortunately, there is no evidence presented that the model performance would have improved. See comment 6.

Author's response: **Comment accepted**.

Author's changes in manuscript: The new figures and explanations provide the required evidence.

Referee's comment #58) Conclusions: The authors could discuss how the results obtained for daytime in a summer month might compare to other seasons.

Author's response**: Comment accepted.** The correction tool is relevant for all dates excluding days with precipitation or dust storms.

Author's changes in manuscript: The comment was added in the new discussion and conclusions Sect.

Referee's comment #59) Table 1: Height limit is given as 7.7 or 15.4 km, but the footnote states that the data acquisition was limited to 4.5 km. It is not clear what is the vertical extent of the measurement. Although it is not that important for the study, the presentation is confusing and could be clarified.

Author's response: **Comment accepted.** The explanation referred to the difference between the ceilometer's capabilities (hardware) to measure up to 7.7 or 15.4 km, and the actual height ranges of the database. Data acquisition is obtained by the ceilometer's software, which organizes daily profiles up to a specific height limit defined by the user. In our case, the profile height limit was 4.5 km, except for 7.7 in Bet Dagan site.

Author's changes in manuscript: Table 1 was clarified.

Referee's comment #60) Table 1: The table includes specifications for the sites such as "north", "south", "inland", "mountain", but these do not seem to be defined or used elsewhere in the manuscript. Perhaps the regions could provisionally be indicated on a map, and used in the discussion of the results.

Author's response: **Comment accepted.**

Author's changes in manuscript: A topographical map was added and reference to the regions of each site was included in the results and conclusions sections.

Referee's comment #61) Table 3: For completeness, the table could include the mean and standard deviation also from the radiosonde used as a reference.

Author's response: **Comment accepted.**

Author's changes in manuscript: The new results section included the mean and standard deviation for six ceilometer sites including radiosonde Bet Dagan.

Note: the comments numbering skip from 61 to 70.

Referee's comment #70) Table 4: "The PBL heights were compared to the heights measured by the Beit Dagan ceilometer." The text states (lines 321-322) "the models' results were compared to the ceilometers' measurements in each site". These two statements seem to contradict each other, and I would ask the authors to correct one of them, or to clarify why different comparison measurements are considered in the text and in the table.

Author's response: **Comment accepted.** The clerical error was in the title of the table. Sorry about that.

Author's changes in manuscript: The tables and titles were changed.

Referee's comment #71) Tables 4 and 5: It would be interesting to also see the mean PBL height of the ceilometer (the reference) at each site.

Author's response: **Comment accepted.**

Author's changes in manuscript: The new results section included the mean and standard deviation for 6 ceilometer sites.

Referee's comment #72) Figures 1 and 6: Considering the political situation in some areas of Western Asia, the authors should carefully consult the journals guidelines regarding maps.

Author's response: **Comment accepted.**

Author's changes in manuscript: The maps were adapted accordingly.

Referee's comments:

Comment #73) Fig 3a: The figure could contain the PBL height estimated by the two methods. It would be helpful to demonstrate the performance of the two methods.

Comment #74) Fig 3b: It does not look like the data has been averaged for 30 min. Is the data presented at original 15 sec resolution? Please clarify in the caption.

Comment #75) Fig 3b: The authors should consider showing the time series of ceilometer and model based PBL height in this figure. It would be interesting to see 1) how the wavelet covariance transformation method is performing on the time series presented, 2) how the models predict the temporal development of the PBL height, and 3) whether the difference between model and ceilometer is random or the two models and two methods are consistently over or underestimating the PBL height during this one day. Although it might seem trivial to the authors, this helps the reader to gain confidence in the methods and helps with the understanding of the diurnal cycle of the PBL that is described in Sect.2.

Comment #76) Fig 3c: The results presented here are not discussed. A description of the results presented here, and the ways they help to interpret Fig. 3 a and b or other results should be added. Furthermore, the wind direction figure could be improved by shifting the x-axis so that it is centered around North (e.g. scale from 180 to 360/0 to 180 degrees).

Comment #77) Fig 4: Figure 4 is hardly mentioned in the manuscript (it is referred to in the caption of Table 4, and Fig 4b is mentioned on line 326). Consequently, it is not clear what this figure is communicating. What is the additional information provided that is not already presented in Fig. 2? The better performance of COSMO compared to IFS, and the good agreement of ceilometer and radiosonde (Fig. 4b) are already demonstrated in Sect. 6.1.

Comment #78) Fig. 5: Figure 5 could indicate the locations of the Tel Aviv and Jerusalem ceilometer stations, as well as the mean (and standard deviation) of the PBL height estimated at these sites.

Comment #79) Fig. 5 and 6: I don't think it is necessary to list the sites and number of days used for the analysis in each figure caption. In my opinion simply a reference to the text for more details would do.

Comment #80) Fig. 6: Figure 6 could include the information of the mean PBL height at the stations.

Comment #81) Fig. 6b: It is not clear what variable is presented in Fig 6b. Is it the ME estimated based on Equation 6, or one of the fitted parameters (α, β, γ)?

Author's response:

**Comments accepted.**

Author's changes in manuscript: Fig 1-6 were replaced.

Referee's comment #82) Citations: The authors should check their citations and list of references list. For example, Uzan et al. (2012) and Uzan et al (2018) are cited but missing from the reference list.

Author's response: **Comment accepted.**

Author's changes in manuscript: Previous citations were checked, and new citations added.

Referee's comment #83) Figures: The authors should pay attention to the quality of figures. The font size could be increased in almost all figures (especially hard to read is Fig. 3), and use of color-blind friendly colors should be considered.

Author's response: **Comment accepted.**

Author's changes in manuscript: New figures are provided.

[revised manuscript text omitted]

Fig. 5 COSMO_R mean PBL height cross-section from Tel Aviv to Jerusalem before (a) and
after (b) correction between 9-14 UTC. The analysis was performed on the number of available
days for each site on August 2015 as follows: Jerusalem - 21 days, Nevatim - 13 days, Hazerim
- 20 days, Ramat David - 26 days, Weizmann - 25 days, Beit Dagan - 13 days, Hadera - 16
days, Tel Aviv - 25 days. Indications of the seashore (dashed line) and the topography (brown
area) are given.

[Figure]

Fig. 6 3D maps of COSMO_R mean PBL heights over Israel at 14 UTC before (a), and after (c)
correction. The regression (b) based on Eq. (6), depicts the height difference between the results
from COSMO_R and the ceilometers. The analysis was performed on the number of available
days for each site on August 2015 as follows: Jerusalem - 21 days, Nevatim - 13 days, Hazerim
- 20 days, Ramat David - 26 days, Weizmann - 25 days, Beit Dagan - 13 days, Hadera - 16

[Figure]

Fig. 3 Same as Fig. 2 but for Tel Aviv on 122 days. The models were compared to the
ceilometer.

[Figure]

Fig. 4 Same as Fig. 2 but for Ramat David on 123 days. The models were compared to the
ceilometer.

[Figure]

1092 Fig. 5 Same as Fig. 2 but for Weizmann on 55 days. The models were compared to the ceilometer.

[Figure]

1096 Fig. 6 Same as Fig. 2 but for Jerusalem on 53 days. The models were compared to the ceilometer.

[Figure]

Fig. 7 Same as Fig. 2 but for Nevatim on 72 days. The models were compared to the ceilometer.

[Figure]

Fig. 8 PBL heights on August 14, 2015, at 9 UTC. The left panel (a) presents an east-west cross-section map, according to the ceilometers' distance from the Mediterranean shoreline. The PBL heights were derived from COSMO-pm (pink line), the ceilometers (black line), the correction tool for COSMO-pm (CR, green line), cross-validation for Bet Dagan (CV-BD, dashed blue line), and cross-validation for Jerusalem (CV-JRM, blue circles). The right panel (b) shows a 2-D map (b) of the height correction range, corresponding to figure (a).

[Figure]

Fig. 9 Same as Fig. 8 but for 10 UTC.

[Figure]

Fig. 10 Same as Fig. 8 but for 11 UTC and including the PBL height estimation from the radiosonde (red star).

[Figure]

Fig. 11 Same as Fig. 8 but for 12 UTC.

[Figure]

Fig. 12 Same as Fig. 8 but for 13 UTC.

[Figure]

Fig. 13 Same as Fig. 8 but for 14 UTC.